# Online Learning with Unknown Constraints

**Karthik Sridharan** [1]  **Seung Won Wilson Yoo** [1]

## Abstract

We consider the problem of online learning where the sequence of actions played by the learner must adhere to an unknown safety constraint at every round. The goal is to minimize regret with respect to the best safe action in hindsight while simultaneously satisfying the safety constraint with high probability on each round. We provide a general meta-algorithm that leverages an online regression oracle to estimate the unknown safety constraint, and converts the predictions of an online learning oracle to predictions that adhere to the unknown safety constraint. On the theoretical side, our algorithm's regret can be bounded by the regret of the online regression and online learning oracles, the eluder dimension of the model class containing the unknown safety constraint, and a novel complexity measure that characterizes the difficulty of safe learning. We complement our result with an asymptotic lower bound that shows that the aforementioned complexity measure is necessary. When the constraints are linear, we instantiate our result to provide a concrete algorithm with $\sqrt{T}$ regret using a scaling transformation that balances optimistic exploration with pessimistic constraint satisfaction.

## 1. Introduction

Online learning is a key tool for many sequential decision making paradigms. From a practical view point, it is often the case that either due to safety concerns (Dobbe et al., 2020), to guarantee fairness or privacy properties (Zafar et al., 2019), (Levy et al., 2021), or in many cases, simply due to physical restrictions in the real world (Atawnih et al., 2016), the agent or learner often must pick actions that are not only effective but also strictly adhering to some constraints on every round. Often, the safety constraint

is determined by parameters of the environment that are unknown to the learner. For example individual fairness constraints may be defined by an unknown similarity metric (Gillen et al., 2018), or in robotics applications, safety may hinge on uncertainties such as an unknown payload weight (Brunke et al., 2022). Thus in such situations, the learner must learn the unknown parameters that characterize the safety constraint.

In this work, we study the general problem of online learning with unknown constraints, where the learner only observes noisy feedback of the safety constraints. We consider arbitrary decision spaces and loss functions. Our goal is to design algorithms that can simultaneously minimize regret while strictly adhering to the safety constraint at all time steps. Naturally, regret is measured w.r.t. the best decision in hindsight that also satisfies the constraint on every round. The learner only has access to an initial safe-set of actions/decisions to begin, and must gain more information about the safety constraint.

To solve this problem, we assume access to a general online learning oracle that has low regret (without explicit regard to safety) and a general online regression oracle that provides us with increasingly accurate estimations of the unknown constraint function. The key technical insight in this work is in exploring what complexity or geometry allows us to play actions within guaranteed safe-sets while expanding the safe-sets and simultaneously ensuring regret is small. We introduce a complexity measure that precisely characterizes this inherent per-step tension between regret minimization and information acquisition with respect to the safety constraint (with the key challenge of remaining within the safe set). We complement our results with a lower bound that shows that asymptotically, whenever this complexity measure is large, regret is also large. Our results yield an analysis that non-constructively shows the existence of algorithms for the general setting with arbitrary decision sets, loss functions, and classes of safety constraints. Furthermore, we instantiate these results explicitly for various settings, and give explicit algorithms for unknown linear constraints and online linear optimization - obtaining the first algorithm with $\mathcal{O}(\sqrt{T})$ regret.

[1]Department of Computer Science, Cornell University, Ithaca NY, United States. Correspondence to: Seung Won Wilson Yoo <sy536@cornell.edu>.

*Proceedings of the 42nd International Conference on Machine Learning*, Vancouver, Canada. PMLR 267, 2025. Copyright 2025 by the author(s).

**Key Contributions**

- We introduce a novel complexity measure $\mathcal{V}_\kappa(t)$ defined in eq. 5, that characterizes the difficulty of safe learning. On a high level, it captures the per-timestep trade off between loss minimization and information gain w.r.t. unknown constraint with $\kappa$ serving as the weight placed on information gain.

- We provide a new safe learning algorithm under an unknown constraint (Algorithm 1) that utilizes an online regression oracle w.r.t. $\mathcal{F}$, where $\mathcal{F}$ is the model class to which the unknown safety constraint belongs, and an online learning oracle that guarantees good performance w.r.t. $\Pi$, where $\Pi$ is a benchmark policy class. Notably, our algorithm is able to handle adversarial contexts drawn from $\mathcal{X}$, arbitrary action set $\mathcal{A}$, safety model class $\mathcal{F}$, benchmark class $\Pi$, operates under general modeling assumptions, and enjoys the following regret bound:

  $$\text{Regret}_T \leq$$
  $$\inf_\kappa \left\{ \sum_{t=1}^{T} \mathcal{V}_\kappa(t) + \kappa \inf_\alpha \left\{ \alpha T + \frac{\text{Reg}_{\text{OR}}(T, \delta, \mathcal{F}) \mathcal{E}(\mathcal{F}, \alpha)}{\alpha} \right\} \right\}$$
  $$+ \text{Reg}_{\text{OL}}(T, \delta, \Pi)$$

  where $\text{Reg}_{\text{OR}}(T, \delta, \mathcal{F})$ denotes the regret bound guaranteed by the online regression oracle on $\mathcal{F}$, $\mathcal{E}(\mathcal{F}, \alpha)$ denotes the eluder dimension of $\mathcal{F}$, and $\text{Reg}_{\text{OL}}(T, \delta, \Pi)$ denotes the regret bound guaranteed by the online learning algorithm w.r.t. $\Pi$.

- Via a lower bound, we show that asymptotically, if there exists a $\kappa$ for which

  $$\lim_{T \to \infty} 1/T \sum_{t=1}^{T} \mathcal{V}_\kappa(t) \geq c_0 > 0 ,$$

  no safe algorithm is able to obtain diminishing regret.

- If per-timestep constraint satisfaction is relaxed to *long term* constraint satisfaction, we show a modified version of our main algorithm yields bounds *without* $\mathcal{V}_\kappa(t)$.

- For several settings including finite action spaces, linear & generalized linear, and polytopic settings we give an instantiation of $\kappa = \kappa^*$ that satisfies $\mathcal{V}_\kappa(t) \leq 0$ and,

  $$\text{Regret}_T \leq \kappa^* \inf_\alpha \left\{ \alpha T + \frac{\text{Reg}_{\text{OR}}(T, \delta, \mathcal{F}) \mathcal{E}(\mathcal{F}, \alpha)}{\alpha} \right\}$$
  $$+ \text{Reg}_{\text{OL}}(T, \delta, \Pi)$$

  For linear settings we instantiate this result to give an algorithm with $\mathcal{O}(\sqrt{T})$ regret. We extend our main algorithm to handle vector-valued constraints

## 2. Related Works

**Online Convex Optimization and Long Term Constraints**: (Mahdavi et al., 2012) initiated the problem of online convex optimization with long term constraints, a variant of online convex optimization where the learner is given a set of functional constraints $\{f_i(\cdot) \leq 0\}_{i=1}^{m}$ and is required to ensure that the sum of constraint violations $\sum_{t=1}^{T} \sum_{i=1}^{m} f_i(x_t)$ is bounded rather than ensuring that the constraints are satisfied at every time step. For the case of known and fixed constraints, (Mahdavi et al., 2012) obtain $\mathcal{O}(T^{1/2})$ regret and $\mathcal{O}(T^{3/4})$ constraint violation. This was recently improved in (Yu & Neely, 2020) to be $\mathcal{O}(T^{1/2})$ regret and $\mathcal{O}(1)$ constraint violation. Furthermore, (Neely & Yu, 2017) study a variant with time varying constraints $\{f_{i,t}(\cdot) \leq 0\}_{i=1}^{m}$ and and achieve $\mathcal{O}(T^{1/2})$ regret and long term constraint violation. (Sun et al., 2017), (Jenatton et al., 2016), and (Yi et al., 2020) study variations of this problem.

**Bandits with Unknown Linear Constraint**: The area of work most similar to ours is the study of safe bandits with unknown linear constraints. Initiated by (Moradipari et al., 2021), this line of works studies a linear bandit setting, where a linear constraint is imposed on every action $a_t$ of the form of $\langle f, a_t \rangle - b \leq 0$ with unknown $f$ and known $b$. Similar settings involving linear bandit problems with uncertain and per-round constraints have been studied by (Amani & Thrampoulidis, 2021), (Pacchiano et al., 2021), (Hutchinson et al., 2024). (Pacchiano et al., 2021) study the a setting where constraints are satisfied in expectation, and (Pacchiano et al., 2024) and (Hutchinson et al., 2024) improves this to high probability satisfaction.

**Safe Convex Optimization with Unknown Constraint(s)**: Safe convex optimization with unknown linear constraints and noisy feedback was studied in (Usmanova et al., 2019). (Fereydounian et al., 2020) seeks to optimize a fixed convex function given unknown linear constraints, and focuses on sample complexity. Closest to our work is that of (Chaudhary & Kalathil, 2022), where the authors study time varying cost functions and achieve $\mathcal{O}(T^{2/3})$ regret. A recent concurrent work due to (Hutchinson et al., 2025) improves this to $\mathcal{O}(\sqrt{T})$. (Hutchinson & Alizadeh, 2024a) studies a variant with non-linear constraints and first order feedback, and (Hutchinson & Alizadeh, 2024b) studies a variant with $d + 1$ points of feedback.

**Per Timestep Tradeoff Between Loss Minimization and Constraint Information Gain** : The SO-PGD algorithm due to (Chaudhary & Kalathil, 2022) adopts an explore-first then exploit strategy which results in a $O(T^{2/3})$ regret bound, whereas the ROFUL algorithm due to (Hutchinson et al., 2024) strikes a better balance between regret minimization and conservative exploration of the constraint set. The Decision Estimation Coefficient (DEC) due to (Foster et al., 2023), (Foster et al., 2022) explicitly strikes a balance loss minimization and the information gained due to obser-

vation. Our proposed algorithm seeks to similarly balance loss minimization and exploration of the constraint set.

## 3. Setup and Preliminary

We consider the problem of online learning with unknown constraints imposed on actions the learner is allowed to play from. The learning problem proceeds for $T$ rounds as follows.

For $t = 1, \ldots, T$:

- Nature selects a *context* $x_t \in \mathcal{X}$
- The learner selects a *safe action* $a_t \in \mathcal{A}$
- Nature selects an *outcome* $y_t \in \mathcal{Y}$
- The learner receives *safety feedback* $z_t \in \mathcal{Z}$

where $\mathcal{X}$ is the *context space*, $\mathcal{A}$ is the *action space*, $\mathcal{Y}$ is the *outcome space* and $\mathcal{Z}$ is the *safety feedback space*. Here, the context chosen by nature $x \in \mathcal{X}$, action chosen by learner $a \in \mathcal{A}$, and outcome chosen by nature $y \in \mathcal{Y}$ define the *loss* suffered by the learner $\ell(a, x, y)$, for a bounded loss function $\ell : \mathcal{A} \times \mathcal{X} \times \mathcal{Y} \to [0, 1]$. We assume a *full information* feedback setting with respect to the losses - the learner observes $y_t \in \mathcal{Y}$.

Given a context $x \in \mathcal{X}$, an action chosen by the learner $a \in \mathcal{A}$ is considered *safe* if $f^\star(a, x) \leq 0$ for an unknown function $f^\star \in \mathcal{F} \subseteq \mathcal{A} \times \mathcal{X} \to [-1, 1]$. While $f^\star$ is unknown to the learner, the *safety function class* $\mathcal{F}$ is known and encodes prior knowledge about the safety constraint. $\mathcal{F}$ may be parameterized by linear models, neural networks, or other rich function approximators depending on the problem setting. At every timestep $t$, the learner receives feedback on unknown safety function $f^\star$ through safety feedback $z_t \sim p_{\text{signal}}(f^\star(a_t, x_t))$, where $p_{\text{signal}} : [-1, 1] \to \Omega(\mathcal{Z})$ and $\Omega(\mathcal{Z})$ denotes the set of distributions over $\mathcal{Z}$. We highlight the fact that given $a_t$ and $x_t$, $z_t$ is conditionally independent of the history. This captures a wide variety of feedback settings, including those where $\mathcal{Z} = \mathbb{R}$ and $z_t = f^\star(a_t, x_t) + \xi_t$ for a noise variable $\xi_t$, and those where $\mathcal{Z} = \{0, 1\}$ and $z_t$ is the result of a coin flip with success probability determined by $f^\star(a_t, x_t)$. Whenever the feedback function is not made explicit, we assume that the feedback is given by $z_t = f^\star(a_t, x_t) + \xi_t$. Throughout the paper, we will compare the performance of our algorithm against a known policy class $\Pi \subseteq \mathcal{X} \to \mathcal{A}$. The goal of the learner is to minimize the regret defined as:

$$\text{Regret}_T :=$$
$$\sum_{t=1}^{T} \ell(a_t, x_t, y_t) - \min_{\substack{\pi \in \Pi : \forall t \\ f^\star(\pi(x_t), x_t) \leq 0}} \sum_{t=1}^{T} \ell(\pi(x_t), x_t, y_t)$$

which is the regret with respect to the optimal policy $\pi$ in hindsight that is safe on every round, i.e. $\forall t \in$

$[T], f^\star(\pi(x_t), x_t) \leq 0$. The learner in turn is also only allowed to take actions $a_t$ s.t. $f^\star(a_t, x_t) \leq 0$. Since we are interested in making no constraint violations while taking our actions, learning is impossible unless we are at least given an initial set of safe actions. For any context $x \in \mathcal{X}$, we assume that we are given a non-empty set $\mathcal{A}_0(x) \subseteq \mathcal{A}$ that is guaranteed to be safe. For settings with no context (i.e. $\mathcal{X} = \emptyset$), we denote the initial safe set as $\mathcal{A}_0$.

### 3.1. Additional Notation

We use the notation $\Pi_S(x)$ to denote the projection of a vector $x \in \mathbb{R}^d$ onto some convex set $S \subseteq \mathbb{R}^d$. For a positive definite matrix $M \in \mathbb{R}^{d \times d}$ and vector $x \in \mathbb{R}^d$ we denote the norm induced by $M$ as $\|x\|_M := \sqrt{x^\intercal M x}$. We denote the convex hull of a set $S$ as $\text{Conv}(S)$. Let $\{x_s\}_{s=1}^t := x_1, \ldots, x_t$ be shorthand for a sequence. For a scalar-valued function class $\mathcal{F} \subseteq \mathcal{A} \times \mathcal{X} \to [0, 1]$, we denote $\Delta_{\mathcal{F}}(a, x) := \sup_{f, f' \in \mathcal{F}} f(a, x) - f'(a, x)$. For a vector-valued function class $\mathcal{F} \subseteq \mathcal{A} \times \mathcal{X} \to [-1, 1]^m$, denote $\Delta_{\mathcal{F}}^\infty(a, x) := \sup_{f, f' \in \mathcal{F}} \|f(a, x) - f'(a, x)\|_\infty$, where $\|\cdot\|_\infty$ is the $\ell_\infty$ norm. For a set $\mathcal{G}$, we denote $\Omega(\mathcal{G})$ as the set of distributions over $\mathcal{G}$. For a vector $x \in \mathbb{R}^d$ and $i \in [d]$, let $x[i]$ be the $i$'th coordinate of $x$. We adopt a non-asymptotic big-oh notation: for functions $f, g : \mathcal{X} \to \mathbb{R}_+$, $f = \mathcal{O}(g)$ if there exists some constant $C > 0$ such that $f(x) \leq Cg(x)$ for all $x \in \mathcal{X}$. We write $f = o(g)$ if for every constant $c$, there exists a $x_0$ such that $f(x) \leq cg(x)$ for all $x \geq x_0$.

### 3.2. Online Regression Oracles and Signal Functions

Similar to prior works concerned with estimation of function classes (Foster et al., 2018a), (Foster & Rakhlin, 2020), (Foster et al., 2021), (Sekhari et al., 2023a), (Sekhari et al., 2023b), we assume our algorithms have access to an online regression oracle, Oracle$_{\text{OR}}$. However, unlike these prior works that assume that the provided online regression oracle enjoys a sublinear regret bound, we require that our oracle satisfy a slightly weaker condition that allows for algorithms geared towards realizability.

**Assumption 3.1** (Online Regression Oracle). The algorithm Oracle$_{\text{OR}}$ guarantees that for any (possibly adversarially chosen) sequence $\{a_t, x_t, z_t\}_{t=1}^T$, for any $\delta \in (0, 1)$, with probability at least $1 - \delta$, generates predictions $\{\hat{z}_t\}_{t=1}^T$ satisfying:

$$\sum_{t=1}^{T} (\hat{z}_t - f^*(a_t, x_t))^2 \leq \text{Reg}_{\text{OR}}(T, \delta, \mathcal{F})$$

Assumption 3.1 is closely linked with the model $p_{\text{signal}}$ that produces feedback about constraint value, and any regret-minimizing oracle for strongly convex losses can be converted into an oracle that satisfies the assumption with high probability. We formalize this in Lemma C.6 in the appendix. For instance, if $z_t \sim p_{\text{signal}}(f^\star(a_t, x_t))$ is given

by $z_t = f^\star(a_t, x_t) + \xi_t$ where $\xi_t$ is any sub-gaussian distributed random variable, then any online square loss regression algorithm on class $\mathcal{F}$ that uses $z_t$ as corresponding outcomes will satisfy Assumption 3.1. Similarly, if $z_t \in \{0, 1\}$ is drawn as $P(z_t = 1|f^\star(a_t, x_t)) \propto \exp(f^\star(a_t, x_t)) = p_{\text{signal}}(f^\star(a_t, x_t))$ (ie. the Boltzman distribution), then one can show that Assumption 3.1 is satisfied by running any online logistic regression algorithm over class $\mathcal{F}$ with $z_t$ as labels.

(Rakhlin & Sridharan, 2014) characterized the minimax rates for online square loss regression in terms of the offset sequential Rademacher complexity for arbitrary $\mathcal{F}$. This for example, leads to regret bounds of the form, $\text{Reg}_{\text{OR}}(T, \mathcal{F}) = \mathcal{O}(\log |\mathcal{F}|)$ for finite function classes $\mathcal{F}$, and $\text{Reg}_{\text{OR}}(T, \mathcal{F}) = \mathcal{O}(d \log(T))$ when $\mathcal{F}$ is a $d$-dimensional linear class. More examples can be found in (Rakhlin & Sridharan, 2014) (Section 4) and efficient implementations can be found in (Krishnamurthy et al., 2017) and (Foster et al., 2018a).

### 3.3. Online Learning Oracles with Given Constraints

Next, we consider the following online learning problem with *given constraints*, which will be instrumental in solving our problem of online learning with unknown constraints. On every round $t$, nature first announces a context $x_t \in \mathcal{X}$ and constraint set corresponding to this context $\mathcal{A}_t(x_t) \subseteq \mathcal{A}$. Because the context is fixed on every timestep, we drop the dependence on $x_t$ and abbreviate the constraint set as $\mathcal{A}_t$. The learner responds with a distribution $p_t \in \Omega(\mathcal{A}_t)$ and draws $a_t \sim p_t$. Nature then produces a loss function $y_t \in \mathcal{Y}$. The learner suffers loss $\ell(a_t, y_t)$.

Let $\Pi \subseteq \mathcal{X} \to \mathcal{A}$ be some given policy class the learner is benchmarked against. A policy $\pi \in \Pi$ satisfies the given constraint at time $t$ if $\pi(x_t) \in \mathcal{A}_t$. Let us denote the set of policies that satisfy *all* given constraints (in hindsight) as $\Pi_T := \{\pi \in \Pi : \forall t, \pi(x_t) \in \mathcal{A}_t\}$.

The goal of the learner is to minimize regret w.r.t. the best policy in $\Pi_T$. We assume access to an online learning oracle, $\text{Oracle}_{\text{OL}}$ that has bounded regret for the problem of online learning with given constraints:

**Assumption 3.2** (Online Learning Oracle)**.** For any sequence of adversarially chosen sets, contexts and outcomes $\{\mathcal{A}_t, x_t, y_t\}_{t=1}^T$ and any $\delta \in (0, 1)$ with probability at least $1 - \delta$, the algorithm $\text{Oracle}_{\text{OL}}$ produces a sequence of distributions $\{p_t\}_{t=1}^T$ satisfying $p_t \in \Omega(\mathcal{A}_t)$ for all $t \in [T]$ with expected regret bounded as:

$$\sum_{t=1}^T \mathbb{E}_{a_t \sim p_t} [\ell(a_t, x_t, y_t)] - \min_{\pi \in \Pi_T} \sum_{t=1}^T \ell(\pi(x_t), x_t, y_t)$$
$$\leq \text{Reg}_{\text{OL}}(T, \delta, \Pi)$$

In our application to online learning with unknown con-

straints, we will pass increasingly accurate approximations of the true set of safe actions for a given context $x_t$ (i.e. $\{a \in \mathcal{A} \mid f^*(a, x_t) \leq 0\}$) in place of $\mathcal{A}_t$ to $\text{Oracle}_{\text{OL}}$. The complexity of the *unconstrained* variant of Assumption 3.2 is well understood ((Rakhlin et al., 2010)). On a first glance, it may appear that the adversarially chosen constraint sets $\mathcal{A}_t$ may pose a significant challenge to learnability. We show that this is not the case using online symmetrization arguments developed in ((Rakhlin et al., 2010)) and a minimax analysis:

**Proposition 3.3.** *There exists an algorithm satisfying Assumption 3.2 with*

$$\text{Reg}_{\text{OL}}(T, \delta, \Pi) \leq 2 \, \text{Rad}_{\ell \circ \Pi}^{\text{seq}}(T)$$

*where* $\text{Rad}_{\ell \circ \Pi}^{\text{seq}}(T)$ *is the sequential Rademacher complexity of the loss class, defined as:*

$$\text{Rad}_{\ell \circ \Pi}^{\text{seq}}(T) :=$$
$$\sup_{\mathbf{y}, \mathbf{x}} \mathbb{E}_\epsilon \left[ \sup_{\pi \in \Pi} \sum_{t=1}^T \epsilon_t \ell \left( \pi(\mathbf{x}_t(\epsilon_{1:t-1})), \mathbf{x}_t(\epsilon_{1:t-1}), \mathbf{y}_t(\epsilon_{1:t-1}) \right) \right]$$

*where in the above supremum over* $\mathbf{y}$ *and* $\mathbf{x}$ *are taken over all mappings of the form* $\mathbf{y} : \bigcup_{t=0}^{T-1} \{\pm 1\}^t \mapsto \mathcal{Y}$ *and* $\mathbf{x} : \bigcup_{t=0}^{T-1} \{\pm 1\}^t \mapsto \mathcal{X}$ *respectively.*

Various properties and examples of bounds on $\text{Rad}_{\ell \circ \Pi}^{\text{seq}}(T)$ can be found in (Rakhlin et al., 2010). Notably, this result is non-constructive and only guarantees the existence of such regret minimizing oracles. In section 5 we provide a constructive gradient-descent based algorithm in the online linear optimization setting.

### 3.4. Eluder Dimension

Before delving into our main results, we recall the following definition of $\epsilon$-dependence and eluder dimension (Russo & Roy, 2013), (Foster et al., 2021), (Foster et al., 2023).

**Definition 3.4.**

- An action, context pair $(a, x) \in \mathcal{A} \times \mathcal{X}$ is $\epsilon$-dependent on $\{a_i, x_i\}_{i=1}^t \subseteq \mathcal{A} \times \mathcal{X}$ w.r.t. $\mathcal{F}$ if every $f, f' \in \mathcal{F}$ satisfying $\sqrt{\sum_{i=1}^t (f(a_i, x_i) - f'(a_i, x_i))^2} \leq \epsilon$ also satisfies $f(a, x) - f'(a, x) \leq \epsilon$. $(a, x)$ is $\epsilon$-independent w.r.t. $\mathcal{F}$ if $a$ is not $\epsilon$-dependent on $\{(a_i, x_i)\}_{i=1}^t$.

- The eluder dimension $\mathcal{E}(\mathcal{F}, \epsilon)$ is the length of the longest sequence of pairs in $\mathcal{A} \times \mathcal{X}$ such that for some $\epsilon' > \epsilon$, each pair is $\epsilon'$-independent of its predecessors.

The eluder dimension is bounded for a variety of function classes. For example, when $\mathcal{F}$ is finite, $\mathcal{E}(\mathcal{F}, \epsilon) \leq |\mathcal{F}| - 1$, and when $\mathcal{F}$ the class of linear functions, $\mathcal{E}(\mathcal{F}, \epsilon) \leq \mathcal{O}(d \log(1/\epsilon))$. The eluder dimension of function class $\mathcal{F}$ will be a component of our regret bounds.

# 4. Main Results

We first propose our complexity measure $\mathcal{V}_\kappa(\cdot)$ that captures the per timestep trade-off between loss minimization and information gain w.r.t safety function set $\mathcal{F}$. We show that this complexity characterizes the difficulty of online learning with unknown constraints by showing an upper bound featuring $\mathcal{V}_\kappa(\cdot)$ and a lower bound that demonstrates a bounded $\mathcal{V}_\kappa(\cdot)$ is asymptotically necessary. Finally, we supplement our results by slightly modifying our algorithm to attain *long term* constraint satisfaction allowing us to contribute to the literature on online learning with long term constraints. Notably, $\mathcal{V}_\kappa(\cdot)$ does not appear in this bound, suggesting this complexity measure is inherent to per-timestep constraint satisfaction.

## 4.1. Complexity Measure and Main Theorem

Given the oracle assumptions, we introduce a few building blocks, define our complexity measure $\mathcal{V}_\kappa$, and provide our main theorem. We first introduce sets of functions likely to contain our true safety function $f^*$, and from these sets of functions, build a pair of sets approximating the true safe action set.

Suppose up until some timestep $t$ we have played actions $\{a_s\}_{s=1}^{t-1}$ and seen contexts and noisy observations $\{x_s, z_s\}_{s=1}^{t-1}$ so far. Given the history $\{a_s, x_s, z_s\}_{s=1}^{t-1}$ suppose $\text{Oracle}_{\text{OR}}$ produces a sequence $\{\hat{z}_t\}_{s=1}^{t-1}$. Let us define a *version space* $\mathcal{F}_t \subset \mathcal{F}_0$ defined as:

$$\mathcal{F}_t := $$
$$\{f \in \mathcal{F}_0 : \sum_{s=1}^{t-1}(f(a_s, x_s) - \hat{z}_s)^2 \leq \text{Reg}_{\text{OR}}(T, \delta, \mathcal{F}_0)\} \quad (1)$$

By Assumption 3.1, the sum of squares deviations $\sum_t(\hat{z}_t - f^\star(a_t, x_t))^2 \leq \text{Reg}_{\text{OR}}(T, \delta, \mathcal{F}_0)$ with high probability. Therefore, the version spaces $\mathcal{F}_t$ contain $f^*$ with high probability. Now suppose we have some set $\mathcal{G} \subseteq \mathcal{F}$ containing the true safety function $f^*$. Given a context $x \in \mathcal{X}$, we define the *optimistic action set* $O(\mathcal{G}, x)$, and *pessimistic action set* $P(\mathcal{G}, x)$ as:

$$O(\mathcal{G}, x) = \{a \in \mathcal{A} \mid \exists f \in \mathcal{G}, f(a, x) \leq 0\}$$
$$P(\mathcal{G}, x) = \{a \in \mathcal{A} \mid \forall f \in \mathcal{G}, f(a, x) \leq 0\} \quad (2)$$

As shorthand, we denote $O_t := O(\mathcal{F}_t, x_t)$, $P_t := P(\mathcal{F}_t, x_t)$. The optimistic set represents the set of all actions that *could* be safe, and the pessimistic set represents the set of all actions that are *guaranteed* to be safe. We capture this notion in the following proposition:

**Proposition 4.1.** *Suppose $f^* \in \mathcal{G} \subseteq \mathcal{F}$. Then:*

$$P(\mathcal{G}, x) \subseteq \{a \in \mathcal{A} : f^\star(a, x) \leq 0\} \subseteq O(\mathcal{G}, x)$$

Intuitively, this proposition suggests an algorithm playing from $P(\mathcal{F}_t, x_t)$ is always safe, and competitiveness with

respect to $O(\mathcal{F}_t, x_t)$ implies competitiveness with all safe actions. We will delve further into this intuition later in this section.

For a set $\mathcal{G} \subseteq \mathcal{F}$, we recall the definition of the *width* of $\mathcal{G}$ with respect to action $a$ and context $x$ introduced in (Russo & Roy, 2013):

$$\Delta_\mathcal{G}(a, x) := \sup_{g, g' \in \mathcal{G}} g(a, x) - g'(a, x) \quad (3)$$

For any action $a \in O(\mathcal{G}, x)$ and $g \in \mathcal{G}$, $g(a, x) \leq \Delta_\mathcal{G}(a, x)$ - hence the width captures a notion of how far an optimistic action is from being pessimistic. Simultaneously if $\Delta_\mathcal{G}(a, x)$ is small, all functions in $\mathcal{G}$ have similar values on $a$, indicating that $a$ provides little information if our goal is to differentiate members of $\mathcal{G}$. These two facts will be instrumental in our analysis.

We now utilize the regret-minimizing properties of $\text{Oracle}_{\text{OL}}$ to be competitive w.r.t. $O_t = O(\mathcal{F}_t, x_t)$. Suppose we pass given context $x_t$ and optimistic set $O_t$ as inputs to the online learning oracle $\text{Oracle}_{\text{OL}}$. Denote $\tilde{p}_t \in \Omega(O_t)$ as the distribution recommended by $\text{Oracle}_{\text{OL}}$. Because $O_t$ contains all constraint-satisfying actions with high probability, combined with the regret guarantee of $\text{Oracle}_{\text{OL}}$ in Assumption 3.2, this guarantees that actions drawn from $\tilde{p}_t$ will have good performance in regret.

As Proposition 4.1 suggests, playing actions from $P_t$ ensures constraint satisfaction. Motivated by this fact, we aim to play actions drawn from $\Omega(P_t)$ optimizing a certain objective striking a balance between matching the performance of (potentially unsafe) $\tilde{p}_t \in \Omega(O_t)$ and information acquisition captured by width. To this end, let $\mathbf{M}$ be any mapping $\Omega(O_t) \to \Omega(P_t)$ parameterized by the function class $\mathcal{F}_t$ and context $x_t$. [1] We will show that when we draw actions $a_t \sim p_t = \mathbf{M}(\tilde{p}_t; \mathcal{F}_t, x_t)$, we can bound regret by quantities involving the eluder dimension, $\text{Reg}_{\text{OR}}$, $\text{Reg}_{\text{OL}}$, and a novel complexity measure $\mathcal{V}_\kappa^\mathbf{M}$ defined as:

$$\mathcal{V}_\kappa^\mathbf{M}(\tilde{p}_t, \mathcal{F}_t, x_t) :=$$
$$\sup_{y \in \mathcal{Y}} \left\{ \mathbb{E}_{a_t \sim \mathbf{M}(\tilde{p}_t; \mathcal{F}_t, x_t)} [\ell(a_t, x_t, y)] - \mathbb{E}_{\tilde{a}_t \sim \tilde{p}_t} [\ell(\tilde{a}_t, x_t, y)] \right\}$$
$$- \kappa \mathbb{E}_{a_t \sim \mathbf{M}(\tilde{p}_t; \mathcal{F}_t, x_t)} [\Delta_{\mathcal{F}_t}(a_t, x_t)]$$

Consequently for a given $\kappa$, the optimal mapping $\mathbf{M}^*$ is to optimize for this complexity measure through a saddle-point optimization:

$$\mathbf{M}_\kappa^*(\tilde{p}_t, \mathcal{F}_t, x_t) :=$$
$$\underset{p_t \in \Omega(P_t)}{\arg\min} \sup_{y \in \mathcal{Y}} \left\{ \mathbb{E}_{a_t \sim p_t} [\ell(a_t, x_t, y)] - \mathbb{E}_{\tilde{a}_t \sim \tilde{p}_t} [\ell(\tilde{a}_t, x_t, y)] \right\} \quad (4)$$
$$- \kappa \mathbb{E}_{a_t \sim p_t} [\Delta_{\mathcal{F}_t}(a_t, x_t)]$$

---

[1] Wherever it is clear what the arguments to the mapping are we may drop them e.g. in a non-contextual setting, the context argument is dropped.

We note that the above optimization can be (inefficiently) solved to desired accuracy through standard techniques by treating it as a two-player game where the min-player chooses $p$ and the max-player choses $f, f', y$ and we form an $\epsilon$-net over the set of actions. Later in section 5, we will give examples of efficient mappings that can be used instead of this optimal mapping. Let us define the complexity measure $\mathcal{V}_\kappa(\cdot) := \mathcal{V}_\kappa^{\mathbf{M}^*}(\cdot)$ as the value attained by the optimal mapping:

$$
\mathcal{V}_\kappa(\tilde{p}_t, \mathcal{F}_t, x_t) :=
$$
$$
\inf_{p_t \in \Omega(P_t)} \sup_{y \in \mathcal{Y}} \left\{ \underset{a_t \sim p_t}{\mathbb{E}} [\ell(a_t, x_t, y)] - \underset{\tilde{a}_t \sim \tilde{p}_t}{\mathbb{E}} [\ell(\tilde{a}_t, x_t, y)] \right\} \quad (5)
$$
$$
- \kappa \underset{a_t \sim p_t}{\mathbb{E}} [\Delta_{\mathcal{F}_t}(a_t, x_t)]
$$

$\mathcal{V}_\kappa$ is similar in spirit to the Decision Estimation Coefficient (DEC) of (Foster et al., 2023) in that it balances a notion of worst-case loss with one of information gain. Specifically, $\mathcal{V}_\kappa$ captures the per-step tension between loss minimization with respect to $\tilde{a}_t \sim \tilde{p}_t$ produced by $\mathsf{Oracle_{OL}}$ and information gain with respect to safety function set $\mathcal{F}_t$, measured by the width of action $a_t$ on $\mathcal{F}_t$. We now present our main theorem:

**Theorem 4.2.** *For any $\delta \in (0,1)$ there exists an algorithm that with probability at least $1 - 3\delta$, produces a sequence of actions $\{a_t\}_{t=1}^T$ that are safe and enjoys the following bound on regret:*

$$
Regret_T \leq \inf_{\kappa > 0} \left\{ \sum_{t=1}^T \mathcal{V}_\kappa(\tilde{p}_t, \mathcal{F}_t, x_t) \right.
$$
$$
\left. + \kappa \inf_\alpha \left\{ \alpha T + \tfrac{20}{\alpha} \mathrm{Reg_{OR}}(T, \delta, \mathcal{F}_0) \mathcal{E}(\mathcal{F}_0, \alpha) \right\} \right\}
$$
$$
+ \mathrm{Reg_{OL}}(T, \delta, \Pi) + \sqrt{2T \log(\tfrac{1}{\delta})}
$$

*Moreover, there exists a problem setting such that if there exists $\kappa > 0$ satisfying*

$$
\lim_{T \to \infty} \frac{1}{T} \sum_{t=1}^T \mathcal{V}_\kappa(\tilde{p}_t, \mathcal{F}_t, x) > 0,
$$

*then, no safe algorithm (with high probability) can ensure that $Regret_T = o(T)$.*

Theorem 4.2 follows by combining the upper bound (Theorem 4.3) with the lower bound (Theorem 4.4), presented in the following subsections.

## 4.2. Algorithm and Upper Bound

We now present an algorithm that attains the upper bound in Theorem 4.2. While using the *optimal mapping* $\mathbf{M}^*$ gives us the best upper bound, solving the optimization involved (eq. 4) may be difficult. To address this, we allow the use of any (potentially efficient) mapping $\mathbf{M}$.

---

**Algorithm 1** General Constrained Online Learning

1: Input: $\mathsf{Oracle_{OL}}$, $\mathsf{Oracle_{OR}}$, $\mathcal{A}_0(\cdot)$, $\delta \in (0,1)$, $\kappa$, $\mathbf{M}$
2: $\mathcal{F}_0 = \{f \in \mathcal{F} : \forall x \in \mathcal{X}, \forall a \in \mathcal{A}_0(x), f(a, x) \leq 0\}$
3: **for** $t = 1, \dots, T$ **do**
4:     **Receive context** $x_t$
$$
\mathcal{F}_t = \{f \in \mathcal{F}_0 : \sum_{s=1}^{t-1} (f(a_s, x_s) - \hat{z}_s)^2 \leq \mathrm{Reg_{OR}}(T, \delta, \mathcal{F}_0)\}
$$
5:     $O_t = O(\mathcal{F}_t, x_t)$, $P_t = P(\mathcal{F}_t, x_t)$
6:     $\tilde{p}_t = \mathsf{Oracle_{OL}}_t(x_t, O_t)$
7:     $p_t = \mathbf{M}(\tilde{p}_t; \mathcal{F}_t, x_t)$
8:     Draw $a_t \sim p_t$
9:     **Receive noisy feedback** $z_t$
10:    Update $\hat{z}_t = \mathsf{Oracle_{OR}}_t(a_t, x_t, z_t)$
11:    **Play $a_t$ and receive** $y_t$
12: **end for**

---

**Theorem 4.3.** *For any $\delta \in (0,1)$ with probability at least $1 - 3\delta$, Algorithm 1 when using mapping $\mathbf{M}$ produces a sequence of actions $\{a_t\}_{t=1}^T$ that are safe and enjoys the following bound on regret:*

$$
Regret_T \leq \inf_{\kappa > 0} \left\{ \sum_{t=1}^T \mathcal{V}_\kappa^{\mathbf{M}}(\tilde{p}_t, \mathcal{F}_t, x_t) \right.
$$
$$
\left. + \kappa \inf_\alpha \left\{ \alpha T + \tfrac{20}{\alpha} \mathrm{Reg_{OR}}(T, \delta, \mathcal{F}_0) \mathcal{E}(\mathcal{F}_0, \alpha) \right\} \right\}
$$
$$
+ \mathrm{Reg_{OL}}(T, \delta, \Pi) + \sqrt{2T \log(\tfrac{1}{\delta})}
$$

*Furthermore, using $\kappa^* = \kappa^*(\mathcal{F}, \mathbf{M})$ defined as*

$$
\sup_{\hat{\mathcal{F}} \subseteq \mathcal{F}, x \in \mathcal{X}, \tilde{p} \in \Delta(O(\hat{\mathcal{F}}, x)), y \in \mathcal{Y}} \frac{\mathbb{E}_{a \sim \mathbf{M}(\tilde{p}; \hat{\mathcal{F}}, x)}[\ell(a, x, y)] - \mathbb{E}_{\tilde{a} \sim \tilde{p}}[\ell(\tilde{a}, x, y)]}{\mathbb{E}_{a \sim \mathbf{M}(\tilde{p}; \hat{\mathcal{F}}, x)}[\Delta_{\hat{\mathcal{F}}}(a, x)]}
$$

*we have*

$$
Regret_T \leq \kappa^* \inf_\alpha \left\{ \alpha T + \frac{20 \mathrm{Reg_{OR}}(T, \delta, \mathcal{F}_0) \mathcal{E}(\mathcal{F}_0, \alpha)}{\alpha} \right\}
$$
$$
+ \mathrm{Reg_{OL}}(T, \delta, \Pi) + \sqrt{T \log(\delta^{-1})}
$$

The proof follows by leveraging the low-regret guarantee of $\mathsf{Oracle_{OL}}$, manipulating the definition of $\mathcal{V}_\kappa^{\mathbf{M}}$ on a per-timestep basis, and bounding the sum of information gain terms $\Delta_{\mathcal{F}_t}(a_t, x_t)$ by an extension of techniques first presented in (Russo & Roy, 2013). In the next section, we give concrete examples of efficient $\mathbf{M}$ which yield bounded $\kappa^*$ for a few important settings.

## 4.3. Lower Bound

Assuming the existence of $\mathsf{Oracle_{OL}}$ with bounded $\mathrm{Reg_{OL}}$ is reasonable as $f^*$ could potentially be the always-safe zero function $f_0 : \mathcal{A} \mapsto 0$, reducing the problem down to exactly the unconstrained online learning problem as $O_t = \mathcal{A}$. Being able to perform regression on $\mathcal{F}$ using a $\mathsf{Oracle_{OR}}$ with bounded $\mathrm{Reg_{OR}}$ and assumption that eluder dimension

of $\mathcal{F}$ is well behaved are also commonplace. However, the reader may question if the sum $\sum_{t=1}^{T} \mathcal{V}_\kappa(\tilde{p}_t, \mathcal{F}_t, x_t)$ must necessarily be small for safe learnability. We show that at least asymptotically, $\sum_{t=1}^{T} \mathcal{V}_\kappa(\tilde{p}_t, \mathcal{F}_t, x_t)$ indeed must be small to guarantee sublinear regret. We capture this notion in the following theorem:

**Theorem 4.4.** *Suppose we are given some $\mathcal{A}_0$, $\mathcal{X} = \{\}$, $\mathcal{Y} = \{\}$, $f^\star \in \mathcal{F}$ and losses $\ell : \mathcal{A} \mapsto \mathbb{R}$ are fixed such that for any $a \in \mathcal{A}$ satisfying $f^\star(a) > 0$, $\ell(a) = \min_{a^* \in \mathcal{A}: f^\star(a^*) \leq 0} \ell(a^*)$. Furthermore suppose $\forall \epsilon$, $\mathcal{E}(\mathcal{F}, \epsilon)$ is finite. Then, if there exists $\kappa > 0$ such that*

$$\lim_{T \to \infty} \frac{1}{T} \sum_{t=1}^{T} \mathcal{V}_\kappa(\tilde{p}_t, \mathcal{F}_t) > 0,$$

*then, safe learning is impossible, i.e. no algorithm that is safe on every round (with high probability) can ensure that $Regret_T = o(T)$.*

On a high level, we show that if $\kappa$ is such that $\lim_{T \to \infty} \frac{1}{T} \sum_{t=1}^{T} \mathcal{V}_\kappa(\tilde{p}_t, \mathcal{F}_t) \geq c^* > 0$, then there must be a $P^* \supset \mathcal{A}_0$ that is guaranteed to be safe and $\forall a \in P^*$ we can estimate $f^\star(a)$ to arbitrary accuracy. Simultaneously we show that this $P^*$ has the property that all actions within it are sub-optimal in loss by at least $c^*$ when compared to best safe action. Further, this set $P^*$ is non-expandable, meaning that we cannot find more actions that are guaranteed to be safe based on playing $a \in P^*$. Once we show the existence of $P^*$ satisfying the aforementioned properties, we can simply announce to any learning algorithm $\mathcal{A}_0 = P^*$. Because the set is non-expandable, and since the learning algorithm must be safe, we conclude any algorithm is doomed to only play actions within $P^*$ - which is known to be $c^*$ sub-optimal.

### 4.4. Per-Timestep Constraints vs Long Term Constraints

Suppose we are only interested in ensuring that the sum of constraint violations $\sum_{t=1}^{T} f^\star(a_t, x_t)$ is $o(T)$, as is the goal in the line of works studying online learning with long term constraints (Mahdavi et al., 2012), (Yu & Neely, 2020), (Sun et al., 2017). We show that this can be done without use of the mapping $\mathbf{M}$ present in the main algorithm - the idea will be to simply play the output of the online learning algorithm given sets $\{O_t\}_{t=1}^{T}$. Algorithm 2 defined in the appendix guarantees:

**Lemma 4.5.** *For any $\delta \in (0, 1)$ with probability at least $1 - 2\delta$, Algorithm 2 produces a sequence of actions $\{a_t\}_{t=1}^{T}$ that satisfies:*

$Regret_T \leq \text{Reg}_{\text{OL}}(T, \delta, \Pi)$ *and*

$$\sum_{t=1}^{T} f^\star(a_t, x_t) \leq \inf_\alpha \left\{ \alpha T + \frac{20 \text{Reg}_{\text{OR}}(T, \delta, \mathcal{F}_0) \mathcal{E}(\mathcal{F}_0, \alpha)}{\alpha} \right\}$$

Notably, $\mathcal{V}_\kappa(\cdot)$ does not appear in either bound. This motivates the question: is assuming access to an online learning oracle and online regression oracle and that the eluder dimension of $\mathcal{F}$ is small enough for us to create algorithms that make no constraint violations with high probability?

Unfortunately the answer is no. We need more assumptions on the initial safe set - which is what the mapping is utilizes. Specifically consider the case where $\ell(a, x, y) = y^\top a$, the constraint set is $\mathcal{F} = \{(a, x) \mapsto f^\top a : \|f\|_2 \leq 1\}$ and suppose $\mathcal{A}_0 = \{0\}$. In this case $\mathcal{E}(\mathcal{F}, \epsilon) = d \log(1/\epsilon)$, and both the online learning oracle and online regression oracle are readily available and satisfy Assumptions 3.1 and 3.2 (e.g. use gradient descent and online linear regression algorithm). However, since $\mathcal{A}_0 = \{0\}$ the initial pessimistic set is $P_1 = \{0\}$. However since we are forced to play in this set, we don't gain any information about $f^\star$ and hence in the subsequent rounds $\mathcal{F}_t = \mathcal{F}$ and $P_t = P_1$. Thus we cannot hope to play anything other than the single safe choice $a_0 = 0$ which prevents us from achieving low regret.

## 5. Examples

In this section, we will give examples of function classes $\mathcal{F}$ where we can construct mappings $\mathbf{M}$ that have bounded $\kappa^*(\mathcal{F}, \mathbf{M})$. Consequently, this results in concrete regret bounds from Theorem 4.2 if these mappings are used in Algorithm 1.

### 5.1. Finite Action Spaces

We first consider the setting of finite action spaces, where $\mathcal{A} = [K]$, $\mathcal{X}$ is arbitrary, $\mathcal{F}^{\text{FAS}} \subseteq \mathcal{A} \times \mathcal{X} \to [-1, 1]$, and losses are functions $\ell : \mathcal{A} \times \mathcal{X} \times \mathcal{Y} \to [0, 1]$. Suppose we make the following assumption that promises some separation between function values:

**Assumption 5.1.** $\forall \mathcal{F} \subseteq \mathcal{F}^{\text{FAS}}$ s.t. $f^* \in \mathcal{F}$ and $P(\mathcal{F}, x) \neq O(\mathcal{F}, x)$, $\max_{a \in P(\mathcal{F}, x)} \Delta_\mathcal{F}(a, x) \geq \Delta_0 > 0$

This assumption is in fact *necessary* for safe learning for large $T$. To see this, suppose $\mathcal{F}^* \subseteq \mathcal{F}^{\text{FAS}}$ satisfies $P(\mathcal{F}^*, x) \neq O(\mathcal{F}^*, x)$ but $\forall a \in P(\mathcal{F}^*, x), \Delta_{\mathcal{F}^*}(a, x) = 0$. Then, we have no hope of shrinking $\mathcal{F}^*$, and consequently expanding $P(\mathcal{F}^*, x)$. If the adversarial losses have values of 1 for all $a \in P(\mathcal{F}^*, x)$, and values of 0 for all $a \notin P(\mathcal{F}^*, x)$, we'd suffer constant loss for all subsequent rounds.

Now, for a set of functions $\mathcal{F} \subseteq \mathcal{F}^{\text{FAS}}$ and context $x$, let $a^\Delta(\mathcal{F}, x) := \text{argmax}_{a \in P(\mathcal{F}, x)} \Delta_\mathcal{F}(a, x)$ be the width-maximizing action in the pessimistic set. We define a mapping for this setting as:

$$\mathbf{M}^{\text{FAS}}(\tilde{p}; \mathcal{F}, x) := \begin{cases} \mathbf{e}_{a^\Delta(\mathcal{F}, x)} & \text{if } P(\mathcal{F}, x) \neq O(\mathcal{F}, x) \\ \tilde{p} & \text{otherwise} \end{cases}$$

Where $\mathbf{e}_a$ is the distribution that places all its mass on $a$. The

above mapping leads to an explore then exploit algorithm with respect to $\mathcal{F}^{\text{FAS}}$ that plays the maximum width safe action until no uncertainty remains w.r.t $\mathcal{F}^{\text{FAS}}$. We show that $\mathbf{M}^{\text{FAS}}$ as defined above has bounded $\kappa^*(\mathcal{F}^{\text{FAS}}, \mathbf{M}^{\text{FAS}})$.

**Lemma 5.2.** *Suppose Assumption 5.1 holds. Then,* $\kappa^*(\mathcal{F}^{\text{FAS}}, \mathbf{M}^{\text{FAS}}) \leq \frac{2}{\Delta_0}$.

*Remark* 5.3. We note that we only require the *existence* of $\Delta_0$ and not knowledge of it. $\Delta_0$ is only required to set $\kappa^* = \kappa^*(\mathcal{F}^{\text{FAS}}, \mathbf{M}^{\text{FAS}})$. In appendix section C.1, we describe a procedure of adaptively selecting time-varing $\kappa_t$ at every timestep such that $\mathcal{V}_{\kappa_t}(t) < 0$ and $\max_t \kappa_t \leq 2\kappa^*$.

### 5.2. Linear Constraints

Next, we consider the setting of linear constraints where $\mathcal{A} \subseteq \mathbb{R}^d$, $\mathcal{X} = \emptyset$, $\Pi = \{\cdot \mapsto a \mid a \in \mathcal{A}\}$ and losses are functions $\ell : \mathcal{A} \times \mathcal{Y} \to [0, 1]$. We show a randomized variant of the scaling-based *doubly-optimistic* method presented in (Hutchinson et al., 2024) (Hutchinson et al., 2025) has bounded $\kappa^*$. Suppose we make the following assumption on $\mathcal{A}$ and $\ell$:

**Assumption 5.4.** The action set $\mathcal{A} \subseteq \mathbb{R}^d$ is convex, compact, and bounded, $\max_{a \in \mathcal{A}} \|a\|_2 \leq D_a$. The losses are lipschitz with constant $D_\ell$, $\forall y \in \mathcal{Y}, \forall a, a' \in \mathcal{A}, |\ell(a, y) - \ell(a', y)| \leq D_\ell \|a - a'\|$. The constants $D_a, D_\ell$ are known to the learner.

Let us consider the following function class: $\mathcal{F}^{\text{Lin}} = \{(a, x) \mapsto \langle f, a \rangle - b \mid f \in \mathbb{R}^d\}$ where $b > 0$ is fixed, and the unknown constraint is $\langle f^*, a \rangle - b \leq 0$. Suppose the initial safe given to the learner is $\mathcal{A}_0 = \{a \in \mathcal{A} : \|a\| \leq b\}$. Then, any $\mathcal{F}_0 = \{f \in \mathbb{R}^d : \|f\| \leq 1\}$, since $f$ with $\|f\| > 1$ has $\langle f, b\frac{f}{\|f\|}\rangle - b > 0$, yet $b\frac{f}{\|f\|} \in \mathcal{A}_0$ violating the promised initial safe set.

*Remark* 5.5. Suppose that $\mathcal{A}_0$ is the $\ell_1$ ball of diameter $b$ instead of the $\ell_2$ ball of diameter $b$. Then $\mathcal{F}_0$ becomes the unit $\ell_\infty$ ball - and the eluder dimension $\mathcal{E}(\mathcal{F}_0, \epsilon)$ increases by a factor of $\log(d)$.

Now, for a set of functions $\mathcal{F}$ and $\tilde{a} \in O(\mathcal{F})$, let $\gamma(\tilde{a}; \mathcal{F}) := \max \{\gamma \in [0, 1] : \gamma \tilde{a} \in P(\mathcal{F})\}$. Define $\mathbf{M}^{\text{Lin}}(\tilde{p}; \mathcal{F})$ as the distribution induced by drawing $\tilde{a} \sim \tilde{p}$ and outputting $\gamma(\tilde{a}; \mathcal{F})\tilde{a}$. In other words, we scale down each action to ensure that it is safe. We show that this scaling-based mapping $\mathbf{M}^{\text{Lin}}$ has bounded $\kappa^*(\mathcal{F}^{\text{Lin}}, \mathbf{M}^{\text{Lin}})$.

**Lemma 5.6.** *Suppose Assumption 5.4 holds. Then,* $\kappa^*(\mathcal{F}^{\text{Lin}}, \mathbf{M}^{\text{Lin}}) \leq \frac{D_\ell D_a}{b}$.

Using the mapping $\mathbf{M}^{\text{Lin}}$, we can get concrete algorithms for the setting of linear constraints, no contexts, and linear losses. Specifically, we set $\text{Oracle}_{\text{OR}}$ to be the Vovk-Azoury-Warmuth forecaster (Vovk, 1997), (Azoury & Warmuth, 2001) which satisfies Assumption 3.1 with $\text{Reg}_{\text{OR}}(T, \delta, \mathcal{F}) \leq \mathcal{O}(d \log(\frac{T}{d\delta}))$. In the case of linear

losses, where $\ell(a_t, \cdot, y_t) = \langle \ell_t, a_t \rangle, \ell_t \in \mathbb{R}^d$ we provide an online gradient descent based algorithm satisfying Assumption 3.2 (Algorithm 3). It is a randomized algorithm that plays elements of the convex hull of $O(\mathcal{F}_t)$ and is stated in the appendix.

**Lemma 5.7.** *Suppose Assumption 5.4 holds. Then Algorithm 3 satisfies Assumption 3.2 with:*

$$\text{Reg}_{\text{OL}}(T, \delta, \Pi) \leq 4D_\ell D_a \sqrt{T \log(2/\delta)}$$

Finally, (Russo & Roy, 2013) show that the Eluder dimension of the linear function class is $\mathcal{E}(\mathcal{F}_{\text{Linear}}, \epsilon) = \mathcal{O}(d \log(1/\epsilon))$. Combining these facts with our main regret bound, we have:

**Corollary 5.8.** *In linear case, for any $\delta > 0$, with probability at least $1 - \delta$ Algorithm 1 satisfies: $\text{Regret}_T = \mathcal{O}\left(\frac{d}{b} \log(\frac{T}{d})\sqrt{T \log(\delta^{-1})}\right)$*

### 5.3. Composing an Activation Function

Suppose we know that some function class $\mathcal{G} \subseteq \mathcal{A} \times \mathcal{X} \to [-1, 1]$ has bounded $\kappa^*(\mathcal{G}, \mathbf{M}_{\mathcal{G}})$ for some mapping $\mathbf{M}_{\mathcal{G}}$. Consider a fixed activation function $\sigma : [-1, 1] \to [-1, 1]$ satisfying the following:

**Assumption 5.9.** $\sigma : [-1, 1] \to [-1, 1]$ is a fixed, differentiable non-decreasing function such that $\sigma(0) = 0$. Furthermore, there exists $\underline{c}$ such that for all $x \in [-1, 1]$, $0 < \underline{c} \leq \sigma'(x)$, where $\sigma'$ is the derivative of $\sigma$.

Consider the following function class formed by composing $\mathcal{G}$ with $\sigma$: $\sigma(\mathcal{G}) = \{(a, x) \mapsto \sigma(g(a, x)) \mid g \in \mathcal{G}\}$ The unknown constraint is then $\sigma(g^*(a, x)) \leq 0$ for some $g^* \in \mathcal{G}$. We show that if $\kappa^*(\mathcal{G}, \mathbf{M}_{\mathcal{G}})$ is bounded for some mapping $\mathbf{M}_{\mathcal{G}}$, $\kappa^*(\sigma(\mathcal{G}), \mathbf{M}_{\mathcal{G}})$ is also bounded. As a consequence of Lemma 5.6 for $\mathcal{F}^{\text{Lin}}$ and the below lemma for compositions with activation functions, generalized linear function classes ((Kakade et al., 2011)) have bounded $\kappa^*$.

**Lemma 5.10.** *Let $\mathcal{G}$ be a function class with bounded $\kappa^*(\mathcal{G}, \mathbf{M}_{\mathcal{G}})$ for some mapping $\mathbf{M}_{\mathcal{G}}$. Suppose assumption 5.4 holds. Then $\kappa^*(\sigma(\mathcal{G}), \mathbf{M}_{\mathcal{G}}) \leq \frac{\kappa^*(\mathcal{G}, \mathbf{M}_{\mathcal{G}})}{\underline{c}}$.*

### 5.4. Vector-Valued Constraints

Our algorithms naturally generalize to settings where the learner must satisfy *vector-valued* constraints. Specifically, given a context $x \in \mathcal{X}$, an action $a \in \mathcal{A}$ is now considered *safe* if $\|\mathring{f}(a, x)\|_\infty \leq 0$ for unknown vector-valued function $\mathring{f} \in \mathcal{F} \subseteq \mathcal{A} \times \mathcal{X} \to [-1, 1]^m$. We formalize this notion in the appendix subsection C.4 - it is a straightforward extension of our main results. Notably, for the setting of $m$ linear constraints, with probability at least $1 - \delta$, regret is $O\left(\frac{md}{b} \log(\frac{T}{d})\sqrt{T \log(\delta^{-1})}\right)$.

Interestingly, we can interpret each of the $m$ coordinates as an independent safety feature, all of which must adhere to safety. Consequently, $|\mathring{f}(a)[i]|$ denotes how unsafe action $a$ is on safety feature $i$, and the constraint is that it cannot exceed $b$. In this scenario, suppose the safety feedback space is $\mathcal{Z} = [m]$ and let the safety feedback function $p_{\text{signal}}(\mathring{f}(a))$ be given by: $P(z = i \mid a) = \frac{\exp(|\mathring{f}(a)[i]|)}{\sum_{j \in [m]} \exp(|\mathring{f}(a)[j]|)}$ as per the Boltzmann distribution on $|\mathring{f}(a)[1]| \ldots |\mathring{f}(a)[j]|$. This is a natural form of feedback known as the Bradley-Luce-Shepherd rule ((Christiano et al., 2017), (Bradley & Terry, 1952)) that outputs which constraint is most likely to be violated - when given $m$ options of varying magnitudes, human-generated feedback has been shown to follow such a distribution ((Ghosal et al., 2023)). In this setting, we can use the logistic regression oracle w.r.t $\mathcal{F}$ as $\text{Oracle}_{\text{OR}}$. When learning linear predictors, (Foster et al., 2018b) provide an efficient algorithm with $\text{Reg}_{\text{OR}} \leq d \log T$.

### 5.4.1. POLYTOPIC CONSTRAINTS WITH SCALAR FEEDBACK

Suppose the safety function class is polytopic: $\mathcal{F}^{\text{Poly}} = \{(a, x) \mapsto Fa - b \mid F \in \mathbb{R}^{d \times m}\}$

Furthermore, suppose the unknown constraint is $\|\mathring{f}(a)\|_\infty \leq 0 \in \mathbb{R}$, for some $\mathring{f} \in \mathcal{F}^{\text{Poly}}$. In this setting, $\mathbf{M}^{\text{Lin}}$ as defined in Lemma 5.6 has bounded $\kappa^*$ for $\mathcal{F}^{\text{Poly}}$:

**Lemma 5.11.** *Suppose Assumption 5.4 holds. Then,* $\kappa^*(\mathcal{F}^{\text{Poly}}, \mathbf{M}^{\text{Lin}}) \leq \frac{D_\ell D_a}{b}$.

## Acknowledgments

KS acknowledges support from LinkedIn-Cornell grant.

## Impact Statement

This paper presents work whose goal is to advance the field of Machine Learning. There are many potential societal consequences of our work, none which we feel must be specifically highlighted here.

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

# A. Proofs from Section 3: Setup

**Definition A.1.** A function $\Phi : [-1, 1] \to \mathbb{R}$ is $\lambda$-strongly convex if for all $z, z' \in [-1, 1]$, it satisfies

$$\frac{\lambda}{2}(z' - z)^2 \leq \Phi(z') - \Phi(z) + \phi(z)(z - z')s$$

where $\phi(\cdot)$ is the derivative of $\Phi$.

The following formulation of link functions is standard in the literature (Sekhari et al., 2023a)

**Definition A.2.** For a link function $\phi$ that is the derivative of a $\lambda$-strongly convex function $\Phi$, we define the associated loss:

$$\ell_\phi(z, z') := \Phi(z) - \frac{z(z' + 1)}{2}$$

**Assumption A.3** (Online Regression Oracle, Regret Version). The algorithm $\mathsf{Oracle_{OR}}$ guarantees that for any (possibly adversarially chosen) sequence $\{a_t, x_t\}_{t=1}^T$ generates predictions $\{\hat{z}_t\}_{t=1}^T$ satisfying:

$$\sum_{t=1}^T \ell_\phi(\hat{z}_t, z_t) - \inf_{f \in \mathcal{F}} \sum_{t=1}^T \ell_\phi(f(a_t, x_t), z_t) \leq \mathrm{Reg}_{\mathrm{OR}}^\phi(T, \mathcal{F})$$

where $z_t \sim \phi(f^*(a_t, x_t))$.

The following lemma is adapted from (Sekhari et al., 2023a), Lemma 9 and related to (Agarwal, 2013), Lemma 2

**Lemma A.4.** *Suppose that $z_t$ is generated with a link function $\phi$ that is $\lambda$-strongly convex. Suppose that the regression oracle satisfies assumption A.3. Then for any $\delta \in (0, 1)$ and $T \geq 3$, with probability at least $1 - \delta$, the regression oracle satisfies assumption 3.1 with:*

$$\mathrm{Reg}_{\mathrm{OR}}(T, \delta, \mathcal{F}) \leq \frac{4}{\lambda}\mathrm{Reg}_{\mathrm{OR}}^\phi(T, \mathcal{F}) + \frac{16 + 24\lambda}{\lambda^2} \log\left(4\delta^{-1}\log(T)\right).$$

*Proof.* The proof is an application of (Sekhari et al., 2023a) Lemma 9. $\qquad\square$

*Proposition* (Proposition 3.3 restated). There exists an algorithm satisfying Assumption 3.2 with

$$\mathrm{Reg}_{\mathrm{OL}}(T, \delta, \Pi) \leq 2\,\mathrm{Rad}_{\ell \circ \Pi}^{\mathrm{seq}}(T)$$

where $\mathrm{Rad}_{\ell \circ \Pi}^{\mathrm{seq}}(T)$ is the sequential Rademacher complexity of the loss class, defined as:

$$\mathrm{Rad}_{\ell \circ \Pi}^{\mathrm{seq}}(T) := \sup_{\mathbf{y}, \mathbf{x}} \mathbb{E}_\epsilon \left[ \sup_{\pi \in \Pi} \sum_{t=1}^T \epsilon_t \ell\left(\pi(\mathbf{x}_t(\epsilon_{1:t-1})), \mathbf{x}_t(\epsilon_{1:t-1}), \mathbf{y}_t(\epsilon_{1:t-1})\right) \right]$$

where in the above supremum over $\mathbf{y}$ and $\mathbf{x}$ are taken over all mappings of the form $\mathbf{y} : \bigcup_{t=0}^{T-1}\{\pm 1\}^t \mapsto \mathcal{Y}$ and $\mathbf{x} : \bigcup_{t=0}^{T-1}\{\pm 1\}^t \mapsto \mathcal{X}$ respectively.

*Proof.* We show that expected regret is bounded by $2\mathrm{Rad}_{\ell \circ \Pi}^{\mathrm{seq}}(T)$ through a minimax analysis with sequential symmetrization techniques that are now standard from (Rakhlin et al., 2010). We use the notation $\langle \mathrm{Operator}_t \rangle_{t=1}^T [A]$ to denote $\mathrm{Operator}_1\{\mathrm{Operator}_2\{\ldots \mathrm{Operator}_T\{A\}\ldots\}\}$. Furthermore, we denote the set of safe policies in hindsight as $\Pi_T := \{\pi \in \Pi : \forall t, \pi(x_t) \in \mathcal{A}_t\}$. We view our online learning setting as a repeated game between adversary and learner where on each round $t$ adversary picks a context and a set $\mathcal{A}_t$ learner picks a (randomized) action from this set and finally adversary picks $y_t$ for that round. The value of this game can we written as:

$$\mathrm{Val}_T = \left\langle \sup_{x_t, \mathcal{A}_t} \inf_{p_t \in \Omega(\mathcal{A}_t)} \sup_{y_t \in \mathcal{Y}} \mathbb{E}_{a_t \sim p_t} \right\rangle_{t=1}^T \left[ \sum_{t=1}^T \ell(a_t, x_t, y_t) - \min_{\pi \in \Pi_T} \sum_{t=1}^T \ell(\pi(x_t), x_t, y_t) \right]$$

$$= \left\langle \sup_{x_t, \mathcal{A}_t} \sup_{q_t \in \Omega(\mathcal{Y})} \inf_{a_t \in \mathcal{A}_t} \mathbb{E}_{y_t \sim q_t} \right\rangle_{t=1}^T \left[ \sum_{t=1}^T \ell(a_t, x_t, y_t) - \min_{\pi \in \Pi_T} \sum_{t=1}^T \ell(\pi(x_t), x_t, y_t) \right]$$

$$= \left\langle \sup_{x_t, \mathcal{A}_t} \sup_{q_t \in \Omega(\mathcal{Y})} \mathbb{E}_{y_t \sim q_t} \right\rangle_{t=1}^T \left[ \sum_{t=1}^T \inf_{a_t \in \mathcal{A}_t} \mathbb{E}_{y_t \sim q_t} \left[ \ell(a_t, x_t, y_t) \right] - \min_{\pi \in \Pi_T} \sum_{t=1}^T \ell(\pi(x_t), x_t, y_t) \right]$$

$$\leq \left\langle \sup_{x_t, \mathcal{A}_t} \sup_{q_t \in \Omega(\mathcal{Y})} \mathbb{E}_{y_t \sim q_t} \right\rangle_{t=1}^T \left[ \max_{\pi \in \Pi_T} \left\{ \sum_{t=1}^T \mathbb{E}_{y_t \sim q_t} \left[ \ell(\pi(x_t), x_t, y_t) \right] - \ell(\pi(x_t), x_t, y_t) \right\} \right]$$

$$\leq \left\langle \sup_{x_t, \mathcal{A}_t} \sup_{q_t \in \Omega(\mathcal{Y})} \mathbb{E}_{y_t, y_t' \sim q_t} \right\rangle_{t=1}^T \left[ \max_{\pi \in \Pi_T} \left\{ \sum_{t=1}^T \left( \ell(\pi(x_t), x_t, y_t') - \ell(\pi(x_t), x_t, y_t) \right) \right\} \right]$$

$$= \left\langle \sup_{x_t, \mathcal{A}_t} \sup_{q_t \in \Omega(\mathcal{Y})} \mathbb{E}_{y_t, y_t' \sim q_t} \mathbb{E}_{\epsilon_t} \right\rangle_{t=1}^T \left[ \max_{\pi \in \Pi_T} \left\{ \sum_{t=1}^T \epsilon_t \left( \ell(\pi(x_t), x_t, y_t') - \ell(\pi(x_t), x_t, y_t) \right) \right\} \right]$$

$$\leq \left\langle \sup_{x_t, \mathcal{A}_t} \sup_{y_t, y_t' \in \mathcal{Y}} \mathbb{E}_{\epsilon_t} \right\rangle_{t=1}^T \left[ \max_{\pi \in \Pi_T} \left\{ \sum_{t=1}^T \epsilon_t \left( \ell(\pi(x_t), x_t, y_t') - \ell(\pi(x_t), x_t, y_t) \right) \right\} \right]$$

$$\leq \left\langle \sup_{x_t, \mathcal{A}_t} \sup_{y_t, y_t' \in \mathcal{Y}} \mathbb{E}_{\epsilon_t} \right\rangle_{t=1}^T \left[ \max_{\pi \in \Pi} \left\{ \sum_{t=1}^T \epsilon_t \left( \ell(\pi(x_t), x_t, y_t') - \ell(\pi(x_t), x_t, y_t) \right) \right\} \right]$$

where first line is obtained using repeated application of minimax theorem (which holds with minor assumptions on action sets and context set etc. that can be found in (Rakhlin et al., 2010)). Second line is a rearrangement. The next line is by noting that each loss-minimizing action $a_t \in \mathcal{A}_t$ has smaller losses than $\pi(x_t) \in \mathcal{A}_t$. The rest of the steps above are standard sequential symmetrization arguments. The key step is the last inequality above where we use the fact that $\Pi_T \subseteq \Pi$. But once this is done, the inner terms are devoid of $\mathcal{A}_t$'s and so we drop them in the supremums and this results in two times the sequential Rademacher complexity of the loss class yielding:

$$\mathrm{Val}_T \leq 2 \, \mathrm{Rad}_{\ell \circ \Pi}^{\mathrm{seq}}(T)$$

Since the minmax value $\mathrm{Val}_T$ is bounded, there exists a regret minimizing algorithm with required bound and this concludes the proof. $\qquad \square$

## B. Proofs from Section 4: Main Results

*Proposition* (Proposition 4.1 restated). Suppose $f^* \in \mathcal{G} \subseteq \mathcal{F}$. Then:

$$P(\mathcal{G}, x) \subseteq \{ a \in \mathcal{A} : f^\star(a, x) \leq 0 \} \subseteq O(\mathcal{G}, x)$$

*Proof.* First suppose $a$ is such that $f^*(a, x_t) \leq 0$. Since $f^* \in \mathcal{G}$, $a \in O(\mathcal{G}, x)$. Hence, $\{ a \in \mathcal{A} : f^\star(a, x_t) \leq 0 \} \subseteq O(\mathcal{G}, x_t)$. Now suppose $a \in P(\mathcal{F}, x_t)$. Then, $\forall f \in \mathcal{G}$, $f(a, x_t) \leq 0$. Since $f^* \in \mathcal{G}$, $f^*(a, x_t) \leq 0$, and hence $P(\mathcal{G}, x_t) \subseteq \{ a \in \mathcal{A} : f^\star(a, x_t) \leq 0 \}$ $\qquad \square$

*Theorem* (Theorem 4.2 restated). For any $\delta \in (0, 1)$ there exists an algorithm that with probability at least $1 - 3\delta$, produces a sequence of actions $\{a_t\}_{t=1}^T$ that are safe and enjoys the following bound on regret:

$$\mathrm{Regret}_T \leq \inf_{\kappa > 0} \left\{ \sum_{t=1}^T \mathcal{V}_\kappa(\tilde{p}_t, \mathcal{F}_t, x_t) + \kappa \inf_\alpha \left\{ \alpha T + \frac{20}{\alpha} \mathrm{Reg}_{\mathrm{OR}}(T, \delta, \mathcal{F}_0) \mathcal{E}(\mathcal{F}_0, \alpha) \right\} \right\} + \mathrm{Reg}_{\mathrm{OL}}(T, \delta, \Pi) + \sqrt{2T \log(\tfrac{1}{\delta})}$$

Moreover, there exists a problem setting such that if there exists $\kappa > 0$ satisfying

$$\lim_{T \to \infty} \frac{1}{T} \sum_{t=1}^T \mathcal{V}_\kappa(\tilde{p}_t, \mathcal{F}_t, x) > 0,$$

then, safe learning is impossible, i.e. no algorithm that is safe on every round (with high probability) can ensure that $\mathrm{Regret}_T = o(T)$.

*Proof.* The first statement follows by applying Theorem 4.3 using the optimal mapping $\mathbf{M} = \mathbf{M}^*$ (defined in eq. 4). The second statement follows from Theorem 4.4. $\qquad \square$

## B.1. Proofs of Upper Bounds

**Lemma B.1.** *With probability at least $1 - \delta$, for all $t \in [T]$, $f^* \in \mathcal{F}_t$.*

*Proof.* This follows immediately from Assumption 3.1 which guarantees with probability at least $1 - \delta$.

$$\sum_{s=1}^{T} (f^*(a_s, x_s) - \hat{z}_s)^2 \leq \text{Reg}_{\text{OR}}(T, \delta, \mathcal{F}_0)$$

and hence for any $t \in [T]$,

$$\sum_{s=1}^{t-1} (f^*(a_s, x_s) - \hat{z}_s)^2 \leq \text{Reg}_{\text{OR}}(T, \delta, \mathcal{F}_0)$$

which shows $f^* \in \mathcal{F}_t$. $\qquad\square$

**Proposition B.2.** *Let $\{\mathcal{F}_t\}_{t=1}^{T}$ be the sequence of version spaces generated by Algorithm 1. For any $\delta \in (0,1)$ with probability at least $(1 - \delta)$ we have for any $t \in [T]$*

$$P(\mathcal{F}_t, x_t) \subseteq \{a \in \mathcal{A} : f^\star(a, x_t) \leq 0\} \subseteq O(\mathcal{F}_t, x_t)$$

*Proof.* By Lemma B.1, we have with probability at least $1 - \delta$ that $f^* \in \mathcal{F}_t$ simultaneously for all $t \in [T]$. Applying Proposition 4.1 immediately yields the result. $\qquad\square$

The following lemma bounds the number of times the width of the set $\mathcal{F}_t$ can exceed some threshold, and is a variant of Proposition 3 of (Russo & Roy, 2013). It is slightly different as our $\mathcal{F}_t$ are constructed around the predictions produced by Oracle$_{\text{OR}}$. We state it for completeness.

**Lemma B.3.** *Let the sequence $\{\mathcal{F}_t, a_t, \hat{z}_t\}_{t=1}^{T}$ be generated by Algorithm 1. Then, for any sequence of adversarial contexts $\{x_t\}_{t=1}^{T}$, and $\epsilon > 0$, it holds that*

$$\sum_{t=1}^{T} \mathbb{1}\{\Delta_{\mathcal{F}_t}(a_t, x_t) > \epsilon\} \leq \left( \frac{4\text{Reg}_{\text{OR}}(T, \delta, \mathcal{F}_0)}{\epsilon^2} + 1 \right) \mathcal{E}(\mathcal{F}_0, \epsilon)$$

*Proof.* First we claim that for $t \in [T]$ if $\Delta_{\mathcal{F}_t}(a_t, x_t) \geq \epsilon$, then $(a_t, x_t)$ must be $\epsilon$-dependent on at most $\frac{4\text{Reg}_{\text{OR}}(T, \delta, \mathcal{F}_0)}{\epsilon^2}$ disjoint subsequences of $(a_1, x_1) \cdots (a_{t-1}, x_{t-1})$. Since $\Delta_{\mathcal{F}_t}(a_t, x_t) > \epsilon$, there must exist two functions $f, f' \in \mathcal{F}_t$ satisfying $f(a_t, x_t) - f'(a_t, x_t) > \epsilon$. By the definition of $\epsilon$-dependence, if $(a_t, x_t)$ is $\epsilon$-dependent on a sequence $(a_{i_1}, x_{i_1}) \cdots (a_{i_\tau}, x_{i_\tau})$ of its predecessors, we must have $\sum_{j=1}^{\tau} (f(a_{i_j}, x_{i_j}) - f'(a_{i_j}, x_{i_j}))^2 > \epsilon^2$. Therefore, if $(a_t, x_t)$ is $\epsilon$-dependent on $N$ such subsequences it follows that $\sum_{j=1}^{t-1} (f(a_j, x_j) - f'(a_j, x_j))^2 > N\epsilon^2$. Therefore:

$$\begin{aligned}
N\epsilon^2 &< \sum_{j=1}^{t-1} (f(a_j, x_j) - f'(a_j, x_j))^2 \\
&= \sum_{j=1}^{t-1} (f(a_j, x_j) - \hat{z}_j + \hat{z}_j - f'(a_j, x_j))^2 \\
&\leq 2 \sum_{j=1}^{t-1} (f(a_j, x_j) - \hat{z}_j)^2 + 2 \sum_{j=1}^{t-1} (f'(a_j, x_j) - \hat{z}_j)^2 \\
&\leq 4\text{Reg}_{\text{OR}}(T, \delta, \mathcal{F}_0)
\end{aligned}$$

where the second inequality follows from $(a + b)^2 \leq 2a^2 + 2b^2$ and the third follows from $f, f' \in \mathcal{F}_t$.

Second, we claim that for any $k \in [T]$ and any sequence $(a_1, x_1) \cdots (a_k, x_k)$, there must be a $j \leq k$ such that $(a_j, x_j)$ is $\epsilon$-dependent on at least $N = \lceil k/\mathcal{E}(\mathcal{F}_0, \epsilon) - 1 \rceil$ disjoint subsequences of its predecessors. We will show an iterative process of finding such an index $j$. Let $S_1 \cdots S_N$ be $N$ subsequences initialized as $S_i = \{(a_i, x_i)\}$ for $i \in [N]$. For $j \in [N+1, k]$ first check if $(a_j, x_j)$ is $\epsilon$-dependent of all $S_i$, $i \in [N]$. If it is, we have found the index $j$ satisfying our condition. Otherwise, pick a $S_i$ such that $x_j$ is $\epsilon$-independent of $S_j$, and add $x_j$ to that $S_i$. By the definition of eluder dimension, the maximum size of each $S_i$, $i \in [N]$ is $\mathcal{E}(\mathcal{F}_0, \epsilon)$, and because $N * \mathcal{E}(\mathcal{F}_0, \epsilon) \leq k - 1$, the process will terminate.

Now, let $(a_{i_1}, x_{i_1}) \cdots (a_{i_k}, x_{i_k})$ be the subsequence such that for $j \in [k]$, $\Delta_{\mathcal{F}_t}(a_{i_j}, x_{i_j}) > \epsilon$. By the first claim, each element of this subsequence is $\epsilon$-dependent on at most $\frac{4\text{Reg}_{\text{OR}}(T, \delta, \mathcal{F}_0)}{\epsilon^2}$ disjoint subsequences. By the second claim, there is some element that is $\epsilon$-dependent on at least $\lfloor (k-1)/\mathcal{E}(\mathcal{F}_0, \epsilon) \rfloor$ disjoint subsequences. It follows that $\lceil (k/\mathcal{E}(\mathcal{F}_0, \epsilon) - 1 \rceil \leq \frac{4\text{Reg}_{\text{OR}}(T, \delta, \mathcal{F}_0)}{\epsilon^2}$, and hence $k \leq \left( \frac{4\text{Reg}_{\text{OR}}(T, \delta, \mathcal{F}_0)}{\epsilon^2} + 1 \right) \mathcal{E}(\mathcal{F}_0, \epsilon)$ $\qquad\square$

The following Lemma utilizes Lemma B.3 to upper bound the sum of $\Delta_{\mathcal{F}_t}$. It is similar in spirit to Lemma 2 of (Russo & Roy, 2013), but our analysis is different and captures a trade-off between $T$ and $\text{Reg}_{\text{OR}}(T, \delta, \mathcal{F}_0)\mathcal{E}(\mathcal{F}_0, \cdot)$.

**Lemma B.4.** *Let the sequence $\{\mathcal{F}_t, p_t\}_{t=1}^T$ be generated by Algorithm 1. Then, for any sequence of adversarial contexts $\{x_t\}_{t=1}^T$,*

$$\sum_{t=1}^T \mathop{\mathbb{E}}_{a_t \sim p_t} [\Delta_{\mathcal{F}_t}(a_t, x_t)] \leq \inf_\alpha \left\{ \alpha T + \frac{20\text{Reg}_{\text{OR}}(T, \delta, \mathcal{F}_0)\mathcal{E}(\mathcal{F}_0, \alpha)}{\alpha} \right\}$$

*Proof.* For a run of Algorithm 1, let $\{a_t\}_{t=1}^T$ be any sequence of actions drawn $a_t \sim p_t$ for all $t \in [T]$. Furthermore, to simplify the notation, let us denote $\Delta_t := \Delta_{\mathcal{F}_t}(a_t, x_t)$. Let us consider some arbitrary $\alpha > 0$. Then, for this sequence of actions and contexts,

$$\sum_{t=1}^T \Delta_{\mathcal{F}_t}(a_t, x_t) := \sum_{t=1}^T \Delta_t$$

$$\overset{(i)}{=} \sum_{t: \Delta_t \leq \alpha} \Delta_t + \sum_{i=0}^{\log(2/\alpha)-1} \left( \sum_{t: 2^i \alpha < \Delta_t \leq 2^{i+1}\alpha} \Delta_t \right)$$

$$\leq \alpha T + \sum_{i=0}^{\log(2/\alpha)-1} \left( \sum_{t: 2^i \alpha < \Delta_t \leq 2^{i+1}\alpha} 2^{i+1}\alpha \right)$$

$$\overset{(ii)}{\leq} \alpha T + \sum_{i=0}^{\log(2/\alpha)-1} \left( 2^{i+1}\alpha \left( \frac{4\text{Reg}_{\text{OR}}(T, \delta, \mathcal{F}_0)}{2^{2i}\alpha^2} + 1 \right) \mathcal{E}(\mathcal{F}_0, 2^i\alpha) \right)$$

$$\overset{(iii)}{\leq} \alpha T + \sum_{i=0}^{\log(2/\alpha)-1} \left( 2^{i+1}\alpha \left( \frac{5\text{Reg}_{\text{OR}}(T, \delta, \mathcal{F}_0)}{2^{2i}\alpha^2} \right) \mathcal{E}(\mathcal{F}_0, 2^i\alpha) \right)$$

$$\leq \alpha T + \sum_{i=0}^{\log(2/\alpha)-1} \left( \frac{10\text{Reg}_{\text{OR}}(T, \delta, \mathcal{F}_0)}{2^i\alpha} \right) \mathcal{E}(\mathcal{F}_0, 2^i\alpha)$$

$$\overset{(iv)}{\leq} \alpha T + \mathcal{E}(\mathcal{F}, \alpha) \sum_{i=0}^{\infty} \frac{10\text{Reg}_{\text{OR}}(T, \delta, \mathcal{F}_0)}{2^i\alpha}$$

$$\overset{(v)}{\leq} \alpha T + \frac{20\text{Reg}_{\text{OR}}(T, \delta, \mathcal{F}_0)}{\alpha}\mathcal{E}(\mathcal{F}_0, \alpha)$$

In (i) we set the upper bound to the sum as $\log(2/\alpha) - 1$ since all functions $f \in \mathcal{F}$ map to $[-1, 1]$, hence $\Delta_t \leq 2$ so it is enough to consider $i : 2^{i+1}\alpha \leq 2$ and (ii) follows from Lemma B.3, (iii) follows from the fact that $1 \leq \frac{\text{Reg}_{\text{OR}}(T, \delta, \mathcal{F})}{(2^i\alpha)^2}$ for $i \in [\log(2/\alpha) - 1]$ if $T > 1$, (iv) follows from the fact that $\mathcal{E}(\mathcal{F}, \cdot)$ is nonincreasing in its second argument, and (v) is an

upper bound from the sum of an infinite series. Therefore, for any sequence $\{\mathcal{F}_t, a_t, x_t\}_{t=1}^T$ generated by Algorithm 1 we have

$$\sum_{t=1}^T \Delta_{\mathcal{F}_t}(a_t, x_t) \le \alpha T + \frac{20\mathrm{Reg}_{\mathrm{OR}}(T, \delta, \mathcal{F}_0)}{\alpha} \mathcal{E}(\mathcal{F}_0, \alpha)$$

Now, since this holds for any sequence $\{\mathcal{F}_t, a_t\}_{t=1}^T$ generated by the algorithm and adversarial contexts $\{x_t\}_{t=1}^T$, it holds in expectation over the algorithm's draws. $\qquad\square$

*Theorem* (Theorem 4.2 restated). For any $\delta \in (0,1)$ with probability at least $1 - 3\delta$, Algorithm 1 when using mapping $\mathbf{M}$ produces a sequence of actions $\{a_t\}_{t=1}^T$ that are safe and enjoys the following bound on regret:

$$\mathrm{Regret}_T \le \inf_{\kappa>0} \left\{ \sum_{t=1}^T \mathcal{V}_\kappa^{\mathbf{M}}(\tilde{p}_t, \mathcal{F}_t, x_t) + \kappa \inf_\alpha \left\{ \alpha T + \frac{20}{\alpha}\mathrm{Reg}_{\mathrm{OR}}(T, \delta, \mathcal{F}_0)\mathcal{E}(\mathcal{F}_0, \alpha) \right\} \right\} + \mathrm{Reg}_{\mathrm{OL}}(T, \delta, \Pi) + \sqrt{2T \log(\tfrac{1}{\delta})}$$

where,

$$\mathcal{V}_\kappa^{\mathbf{M}}(\tilde{p}_t; \mathcal{F}_t, x_t) = \sup_{y \in \mathcal{Y}} \left\{ \mathbb{E}_{a_t \sim \mathbf{M}(\tilde{p}_t; \mathcal{F}_t, x_t)}[\ell(a_t, x_t, y)] - \mathbb{E}_{\tilde{a}_t \sim \tilde{p}_t}[\ell(\tilde{a}_t, x_t, y)] \right\} - \kappa \mathbb{E}_{a_t \sim \mathbf{M}(\tilde{p}_t; \mathcal{F}_t, x_t)}[\Delta_{\mathcal{F}_t}(a_t, x_t)]$$

Furthermore, using $\kappa^* = \kappa^*(\mathcal{F}, \mathbf{M}) := \sup_{\hat{\mathcal{F}} \subseteq \mathcal{F}, x \in \mathcal{X}, \tilde{p} \in \Delta(O(\hat{\mathcal{F}}, x)), y \in \mathcal{Y}} \frac{\mathbb{E}_{a \sim \mathbf{M}(\tilde{p}; \hat{\mathcal{F}}, x)}[\ell(a,x,y)] - \mathbb{E}_{\tilde{a} \sim \tilde{p}}[\ell(\tilde{a}, x, y)]}{\mathbb{E}_{a \sim \mathbf{M}(\tilde{p}; \hat{\mathcal{F}}, x)}[\Delta_{\hat{\mathcal{F}}}(a, x)]}$:

$$\mathrm{Regret}_T \le \kappa^* \inf_\alpha \left\{ \alpha T + \frac{20\mathrm{Reg}_{\mathrm{OR}}(T, \delta, \mathcal{F}_0)\mathcal{E}(\mathcal{F}_0, \alpha)}{\alpha} \right\} + \mathrm{Reg}_{\mathrm{OL}}(T, \delta, \Pi) + \sqrt{T \log(\delta^{-1})}$$

*Proof.* By Proposition B.2, with probability at least $1 - \delta$, if we play actions from $P_t$, we can guarantee the all the constraints are satisfied. On the other hand, to bound the regret of our algorithm w.r.t. the optimal action in hindsight that also satisfies constraint on every round, note that

$$\mathrm{Regret}_T := \sum_{t=1}^T \ell(a_t, x_t, y_t) - \min_{\substack{\pi \in \Pi: \forall t \\ f^\star(\pi(x_t), x_t) \le 0}} \sum_{t=1}^T \ell(\pi(x_t), x_t, y_t)$$

$$\overset{(i)}{\le} \sum_{t=1}^T \ell(a_t, x_t, y_t) - \min_{\substack{\pi \in \Pi: \forall t \\ \pi(x_t) \in O_t}} \sum_{t=1}^T \ell(a, x_t, y_t)$$

$$\le \sum_{t=1}^T \left( \ell(a_t, x_t, y_t) - \mathbb{E}_{\tilde{a}_t \sim p_t}[\ell(\tilde{a}_t, x_t, y_t)] \right) + \sum_{t=1}^T \mathbb{E}_{\tilde{a}_t \sim p_t}[\ell(\tilde{a}_t, x_t, y_t)] - \min_{\substack{\pi \in \Pi: \forall t \\ \pi(x_t) \in O_t}} \ell(a, x_t, y_t)$$

$$\le \sum_{t=1}^T \left( \ell(a_t, x_t, y_t) - \mathbb{E}_{\tilde{a}_t \sim p_t}[\ell(\tilde{a}_t, x_t, y_t)] \right) + \mathrm{Reg}_{\mathrm{OL}}(T, \delta, \Pi)$$

$$\overset{(ii)}{\le} \sum_{t=1}^T \left( \mathbb{E}_{a_t \sim p_t}[\ell(a_t, x_t, y_t)] - \mathbb{E}_{\tilde{a}_t \sim \tilde{p}_t}[\ell(\tilde{a}_t, x_t, y_t)] \right) + \mathrm{Reg}_{\mathrm{OL}}(T, \delta, \Pi) + \sqrt{T \log(\delta^{-1})}$$

$$\le \inf_{\kappa>0} \left\{ \sum_{t=1}^T \mathcal{V}_\kappa^{\mathbf{M}}(\tilde{p}_t; P_t, \mathcal{F}_t, x_t) + \kappa \sum_{t=1}^T \mathbb{E}_{a_t \sim p_t}[\Delta_{\mathcal{F}_t}(a_t, x_t)] \right\} + \mathrm{Reg}_{\mathrm{OL}}(T, \delta, \Pi) + \sqrt{T \log(\delta^{-1})}$$

where (i) follows from the fact that by Proposition B.2, a policy $\pi$ satisfying $\forall t, f^\star(\pi(x_t), x_t) \le 0$ satisfies $\forall t, \pi(x_t) \in O_t$, (ii) is an application of Hoeffding Azuma to bound $\sum_{t=1}^T \ell(a_t, x_t, y_t) - \sum_{t=1}^T \mathbb{E}_{a_t \sim p_t}[\ell(a_t, x_t, y_t)]$ and:

$$\mathcal{V}_\kappa^{\mathbf{M}}(\tilde{p}_t; \mathcal{F}_t, x_t) = \sup_{y \in \mathcal{Y}} \left\{ \mathbb{E}_{a_t \sim \mathbf{M}(\tilde{p}_t; \mathcal{F}_t, x_t)}[\ell(a_t, x_t, y)] - \mathbb{E}_{\tilde{a}_t \sim \tilde{p}_t}[\ell(\tilde{a}_t, x_t, y)] \right\} - \kappa \mathbb{E}_{a_t \sim \mathbf{M}(\tilde{p}_t; \mathcal{F}_t, x_t)}[\Delta_{\mathcal{F}_t}(a_t, x_t)]$$

by Lemma B.4 we can bound the $\sum_{t=1}^{T} \kappa \, \mathbb{E}_{a_t \sim p_t} [\Delta_{\mathcal{F}_t}(a_t, x_t)]$ term, hence,

$$\text{Regret}_T \leq \inf_{\kappa > 0} \left\{ \sum_{t=1}^{T} \mathcal{V}_\kappa^{\mathbf{M}}(\tilde{p}_t; \mathcal{F}_t, x_t) + \kappa \inf_\alpha \left\{ \alpha T + \frac{20\text{Reg}_{\text{OR}}(T, \delta, \mathcal{F}_0) \mathcal{E}(\mathcal{F}_0, \alpha)}{\alpha} \right\} \right\}$$
$$+ \text{Reg}_{\text{OL}}(T, \delta, \Pi) + \sqrt{T \log(\delta^{-1})}$$

This concludes the first bound - which holds with probability at least $1 - 3\delta$ as we take a union bound over the online regression oracle guarantee, the online learning oracle guarantee, and the application of Hoeffding Azuma. To conclude the second part of the statement, we need to show that for

$$\kappa^* = \kappa^*(\mathcal{F}, \mathbf{M}) := \sup_{\hat{\mathcal{F}} \subseteq \mathcal{F}, x \in \mathcal{X}, \tilde{p} \in \Delta(O(\hat{\mathcal{F}}, x)), y \in \mathcal{Y}} \frac{\mathbb{E}_{a \sim \mathbf{M}(\tilde{p}; \hat{\mathcal{F}}, x)} [\ell(a, x, y)] - \mathbb{E}_{\tilde{a} \sim \tilde{p}} [\ell(\tilde{a}, x, y)]}{\mathbb{E}_{a \sim \mathbf{M}(\tilde{p}; \hat{\mathcal{F}}, x)} [\Delta_{\hat{\mathcal{F}}}(a, x)]}$$

we have that $\mathcal{V}_{\kappa^*}^{\mathbf{M}}(\tilde{p}_t; \mathcal{F}_t, x_t) \leq 0$. To this end, note that

$$\mathcal{V}_{\kappa^*}^{\mathbf{M}}(\tilde{p}_t; \mathcal{F}_t, x_t)$$
$$= \sup_{y \in \mathcal{Y}} \left\{ \mathbb{E}_{a_t \sim \mathbf{M}(\tilde{p}_t; \mathcal{F}_t, x_t)} [\ell(a_t, x_t, y)] - \mathbb{E}_{\tilde{a}_t \sim \tilde{p}_t} [\ell(\tilde{a}_t, x_t, y)] \right\} - \kappa^* \mathbb{E}_{a_t \sim \mathbf{M}(\tilde{p}_t; \mathcal{F}_t, x_t)} [\Delta_{\mathcal{F}_t}(a_t, x_t)]$$
$$= \sup_{y \in \mathcal{Y}} \left\{ \mathbb{E}_{a_t \sim \mathbf{M}(\tilde{p}_t; \mathcal{F}_t, x_t)} [\ell(a_t, x_t, y)] - \mathbb{E}_{\tilde{a}_t \sim \tilde{p}_t} [\ell(\tilde{a}_t, x_t, y)] \right\}$$
$$- \left( \sup_{\hat{\mathcal{F}} \subseteq \mathcal{F}, x \in \mathcal{X}, \tilde{p} \in \Delta(O(\hat{\mathcal{F}}, x)), y \in \mathcal{Y}} \frac{\mathbb{E}_{a \sim \mathbf{M}(\tilde{p}; \hat{\mathcal{F}}, x)} [\ell(a, x, y)] - \mathbb{E}_{\tilde{a} \sim \tilde{p}} [\ell(\tilde{a}, x, y)]}{\mathbb{E}_{a \sim \mathbf{M}(\tilde{p}; \hat{\mathcal{F}}, x)} [\Delta_{\hat{\mathcal{F}}}(a, x)]} \right) \mathbb{E}_{a_t \sim \mathbf{M}(\tilde{p}_t; \mathcal{F}_t, x_t)} [\Delta_{\mathcal{F}_t}(a_t, x_t)]$$
$$\leq \sup_{y \in \mathcal{Y}} \left\{ \mathbb{E}_{a_t \sim \mathbf{M}(\tilde{p}_t; \mathcal{F}_t, x_t)} [\ell(a_t, x_t, y)] - \mathbb{E}_{\tilde{a}_t \sim \tilde{p}_t} [\ell(\tilde{a}_t, x_t, y)] \right\}$$

$$- \left( \frac{\sup_{y \in \mathcal{Y}} \{ \mathbb{E}_{a_t \sim \mathbf{M}(\tilde{p}_t; \mathcal{F}_t, x_t)} [\ell(a_t, x_t, y)] - \mathbb{E}_{\tilde{a}_t \sim \tilde{p}_t} [\ell(\tilde{a}_t, x_t, y)] \}}{\mathbb{E}_{a_t \sim \mathbf{M}(\tilde{p}_t; \mathcal{F}_t, x_t)} [\Delta_{\mathcal{F}_t}(a_t, x_t)]} \right) \mathbb{E}_{a_t \sim \mathbf{M}(\tilde{p}_t; P_t, \mathcal{F}_t, x_t)} [\Delta_{\mathcal{F}_t}(a_t, x_t)]$$
$$= 0$$

$\square$

## B.2. Proofs of Lower Bounds

We formalize the notion of $P^* \supset \mathcal{A}_0$ with the properties described in subsection 4.3 with the following lemma, and show its existence.

**Lemma B.5.** *Assume that we have a fixed loss function $\ell : \mathcal{A} \mapsto \mathbb{R}$ such that for any $a \in \mathcal{A}$ satisfying $f^\star(a) > 0$, $\ell(a) = \min_{a^* \in \mathcal{A}: f^\star(a^*) \leq 0} \ell(a^*)$. Furthermore, assume that the eluder dimension of $\mathcal{F}$ at any scale $\epsilon > 0$, (with input space $\mathcal{A}$) is bounded. If for some $c^* > 0$, $\kappa \geq 0$, any regret minimizing oracles $\text{Oracle}_{\text{OL}}$ and $\text{Oracle}_{\text{OR}}$ (assuming regret in both cases is $o(T)$) $\lim_{T \to \infty} \frac{1}{T} \sum_{t=1}^{T} \mathcal{V}_\kappa(\tilde{p}_t, \mathcal{F}_t) \geq c^*$ then, there exists a set $P^* \supseteq \mathcal{A}_0$ with the following properties,*

1. *Set $P^*$ satisfies constraints, i.e. $\forall a \in P^*$, $f^\star(a) \leq 0$*

2. *Define $\mathcal{F}^* = \{ f : \forall a \in P^*, \, f(a) = f^\star(a) \}$. For every action $a \in \mathcal{A} \setminus P^*$, $\exists f \in \mathcal{F}^*$ such that $f(a) > 0$. That is, $P^*$ cannot be expanded to a larger set guaranteed to satisfy constraint.*

3. *$P^*$ is such that $\inf_{a \in P^*} \ell(a) - \inf_{a \in \mathcal{A}: f^\star(a) \leq 0} \ell(a) \geq c^\star$*

*Proof.* First, since loss is fixed and using the property of the loss assumed, any online learning oracle that minimizes regret would have to return distributions over actions $\tilde{p}_t$'s such that $\lim_T \frac{1}{T} \sum_{t=1}^{T} \mathbb{E}_{\tilde{a}_t \sim \tilde{p}_t} [\ell(\tilde{a}_t)] = \inf_{a \in \mathcal{A}} \ell(a)$. We have from the premise that $\lim_{T \to \infty} \frac{1}{T} \sum_{t=1}^{T} \mathcal{V}_\kappa(\tilde{p}_t, \mathcal{F}_t) \geq c^*$ and by the definition of $\mathcal{V}_\kappa(\cdot) = \inf_{\mathbf{M}} \mathcal{V}_\kappa^{\mathbf{M}}(\cdot)$, every mapping $\mathbf{M}$ satisfies

$\lim_{T\to\infty} \frac{1}{T} \sum_{t=1}^{T} \mathcal{V}_\kappa^{\mathbf{M}}(\tilde{p}_t; \mathcal{F}_t) \geq c^*$. Hence this means that for any mapping giving us distributions $p_t$, we have that

$$\lim_{T\to\infty} \frac{1}{T} \sum_{t=1}^{T} \mathbb{E}_{a_t\sim p_t} [\ell(a_t)] - \inf_{a\in\mathcal{A}:f^\star(a)\leq 0} \ell(a) \geq c^*$$

since $\mathbb{E}_{a\sim p_t} [\Delta_{\mathcal{F}_t}(a_t)] \geq 0$. Further, note that since the loss is fixed, if at some point we are able to find distribution $p_t$ such that $\mathbb{E}_{a_t\sim p_t} [\ell(a_t)] - \inf_{a\in\mathcal{A}:f^\star(a)} \ell(a) < c^*$ then by returning this distribution we would violate the premise that $\lim_{T\to\infty} \frac{1}{T} \sum_{t=1}^{T} \mathcal{V}_\kappa(\tilde{p}_t, \mathcal{F}_t) \geq c^*$. Hence we have that for any mapping, and any $t$, $\mathbb{E}_{a_t\sim\mathbf{M}(\tilde{p}_t;P_t,\mathcal{F}_t)} [\ell(a_t)] - \inf_{a\in\mathcal{A}:f^\star(a)\leq 0} \ell(a) \geq c^*$. Since this holds for all mappings, let us consider the following mapping

$$\mathbf{M}(\tilde{p}_t; P_t, \mathcal{F}_t) = \begin{cases} \delta\left(\mathrm{argmin}_{a\in P_t} \ell(a)\right) & \text{if } \min_{a\in P_t} \ell(a) < \inf_{a\in\mathcal{A}:f^\star(a)\leq 0} \ell(a) + c^* \\ \delta\left(\mathrm{argmax}_{a\in P_t} \Delta_{\mathcal{F}_t}(a)\right) & \text{otherwise} \end{cases}$$

where $\delta(\cdot)$ is the point mass distribution. In the above we assume the argmin and argmax exists otherwise we can do a limiting argument. The above mapping is a valid one since loss is fixed and given. Now note that since we already showed that any valid mapping satisfies $\mathbb{E}_{a_t\sim\mathbf{M}(\tilde{p}_t;P_t,\mathcal{F}_t)} [\ell(a_t)] - \inf_{a\in\mathcal{A}:f^\star(a)\leq 0} \ell(a) \geq c^*$ we can conclude $\min_{a\in P_t} \ell(a) \geq \inf_{a\in\mathcal{A}:f^\star(a)\leq 0} \ell(a) + c^*$. Now define the set

$$P^* = \bigcup_{t\geq 1} P_t$$

Since $P_t$'s are all guaranteed to be safe, we have that $P^*$ is also safe, satisfying property 1. Second, since for every $t$, $\min_{a\in P_t} \ell(a) \geq \inf_{a\in\mathcal{A}:f^\star(a)\leq 0} \ell(a) + c^*$ we have that $\inf_{a\in P^*} \ell(a) - \inf_{a\in\mathcal{A}:f^\star(a)\leq 0} \ell(a) \geq c^*$. Thus, $P^*$ satisfies property 3 as well. Finally, to prove property 2, we use the assumption that eluder dimension for any scale $\epsilon$ is finite and that the regression oracle ensures that regret is sub-linear. Specifically, assume that online regression oracle guarantees an anytime regret guarantee of $\phi_\delta(t)$ with probability $1 - \delta$ for any $t$ rounds. In this case, using Lemma B.3 we have that with probability at least $1 - \delta$, for all $T \geq 1$ and all $\epsilon > 0$,

$$\sum_{t=1}^{T} \mathbf{1}\{\Delta_{\mathcal{F}_t}(a_t) > \epsilon\} \leq \left(\frac{4\phi_\delta(T)}{\epsilon^2} + 1\right) \mathcal{E}(\mathcal{F}, \epsilon)$$

for $a_t$'s produced by the above mapping. However, since we are picking $a_t$'s that maximize $\Delta_{\mathcal{F}_t}(a_t)$ on every round and because the $P_t$'s are nested, the indicators are in descending order. Hence, with probability at least $1 - \delta$, for any $\epsilon \in (0, 1]$, let $T_\epsilon$ be the smallest integer such that

$$\frac{T_\epsilon}{\phi_\delta(T_\epsilon)} > \frac{5\mathcal{E}(\mathcal{F}, \epsilon)}{\epsilon^2} ,$$

This is where the condition that regret bound $\phi_\delta(T_\epsilon)$ is $o(T_\epsilon)$ is needed so that the above yields a valid lower bound on $T_\epsilon$. We have that for any $t$, for every action $a \in P_t$, $\mathcal{F}_{t+T_\epsilon}$ is such that $\sup_{f\in\mathcal{F}_{t+T_\epsilon}} |f(a) - f^\star(a)| \leq \epsilon$. The reason we take $\mathcal{F}_{t+T_\epsilon}$ is because $t$ is the first round in which actions in $P_t$ not in earlier sets come into consideration and so we need $T_\epsilon$ more rounds to ensure that for all actions in this set, estimation error is smaller than $\epsilon$. Thus if we consider the set $\bigcap_{t\geq 1} \mathcal{F}_t$, this set corresponds to the set $\mathcal{F}^* = \{f : \forall a \in P^*, \ f(a) = f^\star(a)\}$. Further, by definition of $P_t$'s we have that if there were some action $a$ such that $\forall f \in \mathcal{F}^*$ $f(a) \leq 0$, then this action would be contained in $P^*$. Thus we conclude that every action not in $P^*$ is such that it evaluates to a positive number for some function $f \in \mathcal{F}^*$. Thus we have shown property 2 as well. $\qquad\square$

Once we have the existence of $P^* \supset \mathcal{A}_0$ with the properties described in subsection 4.3 we show that learning is impossible.

**Proposition B.6.** *If there exists a set $P^*$ that has the following properties,*

1. *Set $P^*$ satisfies constraints, i.e. $\forall a \in P^*$, $f^\star(a) \leq 0$*

2. *Define $\mathcal{F}^* = \{f : \forall a \in P^*, f(a) = f^\star(a)\}$. For every action $a \in \mathcal{A} \setminus P^*$, $\exists f \in \mathcal{F}^*$ such that $f(a) > 0$. That is, $P^*$ cannot be expanded to a larger set guaranteed to satisfy constraint.*

3. *$P^*$ is such that $\inf_{a\in P^*} \ell(a) - \inf_{a\in\mathcal{A}:f^\star(a)\leq 0} \ell(a) \geq c^\star$*

*Then, safe learning is impossible, and any learning algorithm that is guaranteed to satisfy constraints on every round (with high probability) has a regret lower bounded by $\mathrm{Regret}_T \geq Tc^*$.*

*Proof.* By property 1, we are guaranteed that $P^*$ is safe so we can start any algorithm with initial safe set $\mathcal{A}_0 = P^*$. Since any safe algorithm must play actions that it can guarantee are safe with high probability, initially any algorithm initialized with $P^*$ has to play from within this set till it can verify some action outside of this set is safe. However by property 3, any action within $P^*$ is at least $c^*$ suboptimal. Any feedback $z_t$ we obtain in the process of playing actions $a_t \in P^*$ would certainly help us evaluate $f^\star(a_t)$ more accurately. However, property 2 implies that even if we were given the values of $f^\star$ for every action in the set $P^*$, we still would not be able to find another action outside of this set that we can conclude is safe unless we make further assumptions on $f^\star$. This is because, each probe/feedback by playing action $a_t$ yields value of $f^\star(a_t) + \xi_t$. Since the noise $\xi_t$ is a standard normal variable, at best we might be able to learn only $f^\star(a)$ for every $a \in P^*$. However, even if we had this information, the best we could conclude is that $f^\star$ is one of the functions in $\mathcal{F}^*$. However, property 2 ensures that for every $a \in \mathcal{A} \setminus P^*$, there is a function in $f \in \mathcal{F}^*$ that matches the value of $f^\star$ on on every action in $P^*$ but has $f(a) > 0$. Since we have no information about which $f \in \mathcal{F}^*$ is the true $f^\star$, no learning algorithm will be able to safely try any action outside of $P^*$ and so any safe learning algorithm will suffer a sub-optimality of at least $c^*$ on every round and thus $\mathrm{Regret}_T \geq Tc^*$ □

*Theorem* (Theorem 4.4 Restated). Suppose we are given some $\mathcal{A}_0$, $\mathcal{X} = \{\}$, $\mathcal{Y} = \{\}$, $f^\star \in \mathcal{F}$ and losses $\ell : \mathcal{A} \mapsto \mathbb{R}$ are fixed such that for any $a \in \mathcal{A}$ satisfying $f^\star(a) > 0$, $\ell(a) = \min_{a^* \in \mathcal{A}: f^\star(a^*) \leq 0} \ell(a^*)$. Furthermore suppose $\forall \epsilon$, $\mathcal{E}(\mathcal{F}, \epsilon)$ is finite. Then, if there exists $\kappa > 0$ such that

$$\lim_{T \to \infty} \frac{1}{T} \sum_{t=1}^{T} \mathcal{V}_\kappa(\tilde{p}_t, \mathcal{F}_t) > 0,$$

then, safe learning is impossible, i.e. no algorithm that is safe on every round (with high probability) can ensure that $\mathrm{Regret}_T = o(T)$.

*Proof.* Combining Lemma B.5 and Proposition B.6 trivially yields the statement of the theorem. □

## B.3. Proofs of Per-Timestep Constraints vs Long Term Constraints

*Lemma* (Lemma 4.5 restated). For any $\delta \in (0, 1)$, there exists an algorithm that with probability at least $1 - 2\delta$ produces a sequence of actions $\{a_t\}_{t=1}^T$ that satisfies:

$$\mathrm{Regret}_T \leq \mathrm{Reg}_{\mathrm{OL}}(T, \delta, \Pi) \text{ and } \sum_{t=1}^{T} f^*(a_t, x_t) \leq \inf_\alpha \left\{ \alpha T + \frac{20\mathrm{Reg}_{\mathrm{OR}}(T, \delta, \mathcal{F}_0)\mathcal{E}(\mathcal{F}_0, \alpha)}{\alpha} \right\}$$

We provide a modified version of Algorithm 1, stated in Algorithm 2, where we do not maintain an pessimistic set, and directly play the output of the Oracle$_{\mathrm{OL}}$. We claim that Algorithm 2 satisfies the guarantee from Lemma 4.5.

---

**Algorithm 2** Online Learning with Long Term Constraints

---

1: Input: Oracle$_{\mathrm{OL}}$, Oracle$_{\mathrm{OR}}$, $\mathcal{A}_0(\cdot)$, $\delta \in (0, 1)$
2: Initialize $\mathcal{F}_0 = \{f \in \mathcal{F} : \forall x \in \mathcal{X}, \forall a \in \mathcal{A}_0(x), f(a, x) \leq 0\}$
3: **for** $t = 1, \cdots, T$ **do**
4:     **Receive context** $x_t$
5:     $\mathcal{F}_t = \{f \in \mathcal{F}_0 : \sum_{s=1}^{t-1}(f(a_s, x_s) - \hat{z}_s)^2 \leq \mathrm{Reg}_{\mathrm{OR}}(T, \delta, \mathcal{F}_0)\}$
6:     $O_t = O(\mathcal{F}_t, x_t)$ **// Optimistic set; cf. eq (2)**
7:     $p_t = \mathrm{Oracle}_{\mathrm{OL}t}(x_t, O_t)$
8:     Draw $a_t \sim p_t$
9:     **Receivenoisy feedback** $z_t$
10:     Update $\hat{z}_t = \mathrm{Oracle}_{\mathrm{OR}t}(a_t, x_t, z_t)$
11:     **Play** $a_t$ **and receive** $y_t$
12: **end for**

---

*Proof.* By Lemma B.1, we know that with probability at least $1 - \delta$, for every $T$ simultaneously, $f^* \in \mathcal{F}_t$. Now for a given timestep $t \in [T]$ consider the action played by the algorithm $a_t$. Because $a_t$ was generated by $\mathsf{Oracle}_{\mathsf{OL}}$, $a_t \in O_t$. For this action, let $\underline{f_t} := \mathrm{argmin}_{f \in \hat{\mathcal{F}}_t} f(a_t, x_t)$. Then

$$\underline{f_t}(a_t, x_t) \leq 0$$
$$\underline{f_t}(a_t, x_t) - \underline{f_t}(a_t, x_t) + f^*(a_t, x_t) \leq \Delta_{\mathcal{F}_t}(a_t, x_t)$$
$$f^*(a_t, x_t) \leq \Delta_{\mathcal{F}_t}(a_t, x_t)$$

Summing up all terms, we get

$$\sum_{t=1}^{T} f^*(a_t, x_t) \leq \sum_{t=1}^{T} \Delta_{\mathcal{F}_t}(a_t, x_t)$$

Now, using Lemma B.4 with each $p_t$ defined as point distributions putting all its mass on $a_t$, we can further bound the above as:

$$\sum_{t=1}^{T} f^*(a_t, x_t) \leq \inf_{\alpha} \left\{ \alpha T + \frac{20 \mathrm{Reg}_{\mathsf{OR}}(T, \delta, \mathcal{F}_0) \mathcal{E}(\mathcal{F}_0, \alpha)}{\alpha} \right\}$$

Finally, since we are just playing the output of our oracle $\mathsf{Oracle}_{\mathsf{OL}}$, the regret bound is simply

$$\mathrm{Regret}_T := \sum_{t=1}^{T} \ell(a_t, x_t, y_t) - \min_{\substack{\pi \in \Pi: \forall t \\ f^\star(\pi(x_t), x_t) \leq 0}} \sum_{t=1}^{T} \ell(\pi(x_t), x_t, y_t)$$
$$\leq \sum_{t=1}^{T} \ell(a_t, x_t, y_t) - \min_{\substack{\pi \in \Pi: \forall t \\ \pi(x_t) \in O_t}} \sum_{t=1}^{T} \ell(a, x_t, y_t)$$
$$\leq \mathrm{Reg}_{\mathsf{OL}}(T, \delta, \Pi)$$

where the second inequality follows from the fact that by Proposition B.2, a policy $\pi$ satisfying $\forall t, f^\star(\pi(x_t), x_t) \leq 0$ satisfies $\forall t, \pi(x_t) \in O_t$. The theorem statement holds with probability at least $1 - 2\delta$ as we apply a union bound over the online regression oracle guarantee and the online learning oracle guarantee $\qquad \square$

## C. Proofs from Section 5: Examples

### C.1. Proofs for Finite Action Spaces

*Procedure for selecting adaptive $\kappa_t$*: Let us define $\kappa^* : \kappa^*(\mathcal{F}^{\mathrm{FAS}}, \mathbf{M}^{\mathrm{FAS}})$ (defined in Theorem 4.3 statement). $\kappa^*$ need to be known - but we will show we can attain similar as or potentially much better results than $\kappa^*$ nonetheless. For timestep $t$, given context $x_t$ and $\mathcal{F}_t, P_t, \tilde{p}_t$ generated by the algorithm, let $V_t(\kappa)$ be the value of the saddle-point optimization in eq 5, which is monotonically decreasing in $\kappa$. Observe that since $\inf_{\alpha} \left\{ \alpha T + \frac{20 \mathrm{Reg}_{\mathsf{OR}}(T, \delta, \mathcal{F}_0) \mathcal{E}(\mathcal{F}_0, \alpha)}{\alpha} \right\} \geq \sqrt{T}$, $\kappa$ ranges from $(0, \sqrt{T}]$ in order for the regret bounds to be $o(T)$. Therefore, starting with $\kappa_t = \sqrt{T}$, repeatedly halve $\kappa_t$ while $V_t(\kappa_t) < 0$. Set $\kappa_t$ to be the last $\kappa_t$ satisfying $V_t(\kappa_t) < 0$, and play $p_t$ as the minimizer of $V_t(\kappa_t)$. By definition, $\kappa^*$ satisfies $V_t(\kappa^*) < 0$, and therefore at worst $\kappa_t < 2\kappa^*$ while simultaneously $\kappa_t$ may be be much smaller as we are adapting to $x_t, \mathcal{F}_t, P_t, \tilde{p}_t$. Consequently, our final regret bound would depend on $\max_t \kappa_t \leq 2\kappa^*$. Thus we can easily obtain the result competing with $\kappa^{**}$ corresponding to the best mapping. As a further note, we can relax the condition $V_t(\kappa_t) < 0$ by $V_t(\kappa_t) < T^{-p}$ for some appropriate $p$ to get more general rates.

*Lemma* (Lemma 5.2 restated). Suppose Assumption 5.1 holds. Then, $\kappa^*(\mathcal{F}^{\mathrm{FAS}}, \mathbf{M}^{\mathrm{FAS}}) \leq \frac{2}{\Delta_0}$.

*Proof.* First suppose $P_t \neq O_t$. Then we have $\mathbf{M}^{\mathrm{FAS}}(\tilde{p}; \mathcal{F}, x) = \mathbf{e}_{a^\triangle(\mathcal{F}, x)}$. Let $\mathcal{F}^*$ be the maximizer of $\kappa^*(\mathcal{F}^{\mathrm{FAS}}, \mathbf{M}^{\mathrm{FAS}})$. Then:

$$\kappa^*(\mathcal{F}^{\mathrm{FAS}}, \mathbf{M}^{\mathrm{FAS}}) = \sup_{x \in \mathcal{X}, \tilde{p} \in \Delta(O(\mathcal{F}^*, x)), y \in \mathcal{Y}} \frac{\mathbb{E}_{a \sim \mathbf{M}^{\mathrm{FAS}}(\tilde{p}; \mathcal{F}^*, x)} [\ell(a, x, y)] - \mathbb{E}_{\tilde{a} \sim \tilde{p}} [\ell(\tilde{a}, x, y)]}{\mathbb{E}_{a \sim \mathbf{M}^{\mathrm{FAS}}(\tilde{p}; \mathcal{F}^*, x)} [\Delta_{\mathcal{F}^*}(a, x)]}$$

$$= \sup_{x \in \mathcal{X}, \tilde{p} \in \Delta(O(\mathcal{F}^*,x)), \ell \in [0,1]^K} \frac{\langle \ell, \mathbf{e}_{a^\Delta(\mathcal{F}^*,x)} - \tilde{p} \rangle}{\Delta_{\mathcal{F}^*}(a^\Delta(\mathcal{F}^*,x),x)}$$

$$\leq \sup_{x \in \mathcal{X}, \tilde{p} \in \Delta(O(\mathcal{F}^*,x)), \ell \in [0,1]^K} \frac{\|\ell\|_\infty \|\mathbf{e}_{a^\Delta(\mathcal{F}^*,x)} - \tilde{p}\|_1}{\Delta_0}$$

$$\leq \frac{2}{\Delta_0}$$

where the first inequality follows from the fact that given a context $x$, $a^\Delta(\mathcal{F}^*, x)$ maximizes $\Delta_{\mathcal{F}^*}(\cdot, x)$, and hence by Assumption 5.1 $\Delta_{\mathcal{F}^*}(a^\Delta(\mathcal{F}^*, x), x) \geq \Delta_0$.

Now if $P_t = O_t$, $\mathbf{M}^{\mathrm{FAS}}(\tilde{p}_t, P_t, \mathcal{F}_t) = \tilde{p}_t$. Clearly, by definition, $\kappa^*(\mathcal{F}^{\mathrm{FAS}}, \mathbf{M}^{\mathrm{FAS}}) = 0$.

$\square$

### C.2. Proofs for Linear Constraints

*Lemma* (Lemma 5.6 restated). Suppose Assumption 5.4 holds. Then, $\kappa^*(\mathcal{F}^{\mathrm{Lin}}, \mathbf{M}^{\mathrm{Lin}}) \leq \frac{D_\ell D_a}{b}$.

We first introduce a lemma that lower bounds $\gamma(\tilde{a}; \mathcal{F})$.

**Lemma C.1.** *Let $\mathcal{F}$ be an arbitrary subset of $\mathcal{F}^{\mathrm{Lin}}$ and consider some $\tilde{a} \in O(\mathcal{F})$. $\gamma(\tilde{a}; \mathcal{F}) := \max\{\gamma \in [0,1] : \gamma\tilde{a} \in P(\mathcal{F})\}$ is lower bounded as:*

$$\gamma(\tilde{a}; \mathcal{F}) \geq \frac{b}{b + \Delta_{\mathcal{F}}(\tilde{a})}$$

*Proof.* Recall that $\mathcal{F} \subseteq \mathcal{F}^{\mathrm{Lin}} := \{(a, x) \mapsto \langle f, a \rangle - b | f \in \mathbb{R}^d\}$. Define $\underline{f} := \mathrm{argmin}_{f \in \mathcal{F}} f(\tilde{a})$, and let $\overline{f}$ be some arbitrary function in $\mathcal{F}$. From the definition of $\tilde{a} \in O(\mathcal{F})$, we have $\underline{f}(\tilde{a}) \leq 0$. Then:

$$\underline{f}(\tilde{a}) \leq 0$$
$$\underline{f}(\tilde{a}) + b \leq b$$
$$\underline{f}(\tilde{a}) + \overline{f}(\tilde{a}) - \underline{f}(\tilde{a}) + b \leq b + \Delta_{\mathcal{F}}(\tilde{a})$$
$$\overline{f}(\tilde{a}) + b \leq b + \Delta_{\mathcal{F}}(\tilde{a})$$

where the third inequality follows from the definition of $\Delta_{\mathcal{F}}(\cdot)$. Let $\alpha = \frac{b}{b + \Delta_{\mathcal{F}}(\tilde{a})}$. Notice that $\overline{f}(\tilde{a}) + b$ is a linear function $\langle \overline{w}, \tilde{a} \rangle$ for some $\overline{w} \in \mathbb{R}^d$. It follows that $\alpha(\overline{f}(\tilde{a}) + b) = \alpha \langle \overline{w}, \tilde{a} \rangle = \langle \overline{w}, \alpha\tilde{a} \rangle = \overline{f}(\alpha\tilde{a}) + b$. Using this fact,

$$\overline{f}(\tilde{a}) + b \leq b + \Delta_{\mathcal{F}}(\tilde{a})$$
$$\alpha\left(\overline{f}(\tilde{a}) + b\right) \leq \alpha(b + \Delta_{\mathcal{F}}(\tilde{a}))$$
$$\overline{f}(\alpha\tilde{a}) + b \leq b$$
$$\overline{f}(\alpha\tilde{a}) \leq 0$$

Since $\overline{f}$ was an arbitrary function in $\mathcal{F}$, this shows that $\alpha\tilde{a} \in P(\mathcal{F})$. Since we defined $\gamma(\tilde{a}; \mathcal{F}) = \max\{\gamma \in [0,1] : \gamma\tilde{a} \in P(\mathcal{F})\}$

$$\gamma(\tilde{a}; \mathcal{F}) \geq \alpha = \frac{b}{b + \Delta_{\mathcal{F}}(\tilde{a})}$$

$\square$

We now prove Lemma 5.6.

*Proof.* Let $\mathcal{F}^*$ be the maximizer of $\kappa^*(\mathcal{F}^{\mathrm{Lin}}, \mathbf{M}^{\mathrm{Lin}})$. Using the definition of $\kappa^*$,

$$
\begin{aligned}
\kappa^*(\mathcal{F}^{\mathrm{Lin}}, \mathbf{M}^{\mathrm{Lin}}) &= \sup_{x \in \mathcal{X}, \tilde{p} \in \Delta(O(\mathcal{F}^*, x)), y \in \mathcal{Y}} \frac{\mathbb{E}_{a \sim \mathbf{M}^{\mathrm{Lin}}(\tilde{p}; \mathcal{F}^*, x)}[\ell(a, x, y)] - \mathbb{E}_{\tilde{a} \sim \tilde{p}}[\ell(\tilde{a}, x, y)]}{\mathbb{E}_{a \sim \mathbf{M}^{\mathrm{Lin}}(\tilde{p}; \mathcal{F}^*, x)}[\Delta_{\mathcal{F}^*}(a, x)]} \\
&= \sup_{\tilde{a} \in O(\mathcal{F}^*), y \in \mathcal{Y}} \frac{\ell(\mathbf{M}^{\mathrm{Lin}}(\tilde{a}; \mathcal{F}^*), y) - \ell(\tilde{a}, y)}{\Delta_{\mathcal{F}^*}(\mathbf{M}^{\mathrm{Lin}}(\tilde{a}; \mathcal{F}^*))} \\
&= \sup_{\tilde{a} \in O(\mathcal{F}^*), y \in \mathcal{Y}} \frac{\ell(\gamma(\tilde{a}; \mathcal{F}^*)\tilde{a}, y) - \ell(\tilde{a}, y)}{\Delta_{\mathcal{F}^*}(\gamma(\tilde{a}; \mathcal{F}^*)\tilde{a})} \\
&= \sup_{\tilde{a} \in O(\mathcal{F}^*), y \in \mathcal{Y}} \frac{D_\ell \|\gamma(\tilde{a}; \mathcal{F}^*)\tilde{a} - \tilde{a}\|}{\Delta_{\mathcal{F}^*}(\gamma(\tilde{a}; \mathcal{F}^*)\tilde{a})} \\
&= \sup_{\tilde{a} \in O(\mathcal{F}^*), y \in \mathcal{Y}} \frac{D_\ell D_a (1 - \gamma(\tilde{a}; \mathcal{F}^*))}{\gamma(\tilde{a}; \mathcal{F}^*)\Delta_{\mathcal{F}^*}(\tilde{a})}
\end{aligned}
$$

Now Lemma C.1 implies

$$
\frac{1 - \gamma(\tilde{a}; \mathcal{F}^*)}{\gamma(\tilde{a}; \mathcal{F}^*)} \leq \left(1 - \frac{b}{b + \Delta_{\mathcal{F}^*}(\tilde{a})}\right)\frac{b + \Delta_{\mathcal{F}^*}(\tilde{a})}{b} = \left(\frac{\Delta_{\mathcal{F}^*}(\tilde{a})}{b + \Delta_{\mathcal{F}^*}(\tilde{a})}\right)\frac{b + \Delta_{\mathcal{F}^*}(\tilde{a})}{b} = \frac{\Delta_{\mathcal{F}^*}(\tilde{a})}{b}
$$

and hence

$$
\kappa^*(\mathcal{F}^{\mathrm{Lin}}, \mathbf{M}^{\mathrm{Lin}}) \leq \frac{D_\ell D_a}{b}
$$

$\square$

### C.2.1. PROOF OF LEMMA 5.7

We present a constructive online learning oracle $\mathsf{Oracle}_{\mathrm{OL}}$ for the case of linear cost functions. Recall that $\mathsf{Oracle}_{\mathrm{OL}}$ must satisfy Assumption 3.2 restated below:

*Assumption* (Online Learning Oracle). For any sequence of adversarially chosen sets $\{\mathcal{A}_t\}_{t=1}^T$ and any $\delta \in (0, 1)$ with probability at least $1 - \delta$, the algorithm $\mathsf{Oracle}_{\mathrm{OL}}$ produces a sequence of distributions $\{p_t\}_{t=1}^T$ satisfying $p_t \in \Omega(\mathcal{A}_t)$ for all $t \in [T]$ with expected regret bounded as:

$$
\sum_{t=1}^T \mathbb{E}_{a_t \sim p_t}[\ell(a_t, x_t, y_t)] - \min_{a \in \cap_{t=1}^T \mathcal{A}_t} \sum_{t=1}^T \ell(a, x_t, y_t) \leq \mathrm{Reg}_{\mathrm{OL}}(T, \delta, \Pi)
$$

Algorithm 3 is stated below, and it is a projected online gradient descent based algorithm. Let the losses encountered at timesteps $t \in [T]$ be denoted by $\ell(a_t, \cdot, y_t) = \langle \ell_t, a_t \rangle$.

---

**Algorithm 3** $\mathsf{Oracle}_{\mathrm{OL}}$ for Linear Losses

---

1: Input: $\mathcal{A}, D_a, D_\ell, \delta \in (0, 1), \eta$
2: **for** timesteps $t = 1, \cdots, T$ **do**
3:     Receive $\mathcal{A}_t = O_t$
4:     $\bar{a}_t \leftarrow \Pi_{\mathrm{Conv}(\mathcal{A}_t)}(\bar{a}_{t-1} - \eta\ell_{t-1})$
5:     Decompose $\bar{a}_t = \sum_{i=1}^{d+1} p_{t,i} a_{t,i}, \forall i, a_{t,i} \in \mathcal{A}_t$
6:     Sample $\tilde{a}_t \sim p_t$
7:     Receive $\nabla_t = \ell_t$
8: **end for**

---

We briefly describe the the steps in Algorithm 3.
**Convex Hulls of $\mathcal{A}_t$ (line 4)**

Because the action sets $\mathcal{A}_t = O_t$ are sublevel sets of a minimum of affine functions, they are not necessarily convex, making them incompatible with projection based online learning algorithms. In order to address this, we take the convex hull of $\mathcal{A}_t$, $\mathrm{Conv}(\mathcal{A}_t)$, as our projection set.

**Projected Online Gradient Descent (line 4)**
Our algorithm then performs projected online gradient descent in sets $\mathrm{Conv}(\mathcal{A}_t)$, generating a sequence of vectors $\{\bar{a}_1 \cdots \bar{a}_T\}$ produced by $\bar{a}_t = \Pi_{\mathrm{Conv}(\mathcal{A}_t)}(\bar{a}_{t-1} - \eta \ell_{t-1})$. We note that while we use projected online gradient descent, because the vectors $\{\bar{a}_1 \cdots \bar{a}_T\}$ are maintained and updated independently, we could alternatively use a projected variant of any other online convex optimization algorithm that guarantees low regret instead.

**Sampling a Point in $\mathcal{A}_t$ (line 5)**
Due to Carathéodory's theorem, we know that can write any $\bar{a}_t \in \mathrm{Conv}(\mathcal{A}_t)$ as a linear combination of at most $d+1$ vectors in $\mathcal{A}_t$, $\bar{a}_t = \sum_{i=1}^{d+1} p_{t,i} a_{t,i}, \forall i, a_{t,i} \in \mathcal{A}_t$. In line 4, we perform this decomposition, and in line 5, we sample the vector $\tilde{a}_t$ according to this distribution $p_t$. Notably, the point $\tilde{a}_t$ satisfies $\mathbb{E}[\tilde{a}_t] = \bar{a}_t$.

*Lemma* (Lemma 5.7 restated). Suppose Assumption 5.4 holds. Then Algorithm 3 satisfies Assumption 3.2 with:

$$\mathrm{Reg}_{\mathrm{OL}}(T, \delta, \Pi) \leq 4 D_\ell D_a \sqrt{T \log(2/\delta)}$$

*Proof.* At every timestep $t \in [T]$, Algorithm 3 receives a set $\mathcal{A}_t = O_t$, and produces a $\bar{a}_t$ by:

$$\bar{a}_t \leftarrow \Pi_{\mathrm{Conv}(\mathcal{A}_t)}(\bar{a}_{t-1} - \eta \ell_{t-1})$$

then, it decomposes each $\bar{a}_t$ as:

$$\bar{a}_t = \sum_{i=1}^{d+1} p_{t,i} a_{t,i} \forall i, a_{t,i} \in \mathcal{A}_t$$

and then $\tilde{a}_t$ is produced by sampling: $\tilde{a}_t \sim p_t$. We analyze the regret of Algorithm 3 by decomposing it into two terms:

$$
\begin{aligned}
\mathrm{Reg}_{\mathrm{OL}}(T, \delta, \Pi) &= \sum_{t=1}^{T} \langle \ell_t, \tilde{a}_t \rangle - \min_{a \in \bigcap_{t=1}^{T} \mathcal{A}_t} \langle \ell_t, a \rangle \\
&= \underbrace{\sum_{t=1}^{T} \langle \ell_t, \tilde{a}_t \rangle - \langle \ell_t, \bar{a}_t \rangle}_{\text{Term I}} + \underbrace{\sum_{t=1}^{T} \langle \ell_t, \bar{a}_t \rangle - \min_{a \in \bigcap_{t=1}^{T} \tilde{a}_t} \langle \ell_t, a \rangle}_{\text{Term II}}
\end{aligned}
$$

**Bounding Term I**
We show that Term I is a difference between a bounded random variable and its expectation, and use Hoeffding's inequality to bound it. Let $S_T := \sum_{t=1}^{T} \langle \ell_t, \tilde{a}_t \rangle$. Then:

$$\mathbb{E}[S_T] = \mathbb{E}[\sum_{t=1}^{T} \langle \ell_t, \tilde{a}_t \rangle] = \sum_{t=1}^{T} \langle \ell_t, \mathbb{E}[\tilde{a}_t] \rangle = \sum_{t=1}^{T} \langle \ell_t, \bar{a}_t \rangle$$

where the second equality follows by linearity of expectation. Note that each summand in $S_T$ satisfies $|\langle \ell_t, \tilde{a}_t \rangle| \leq \|\ell_t\|\|\tilde{a}_t\| \leq D_\ell D_a$. Hence, by Hoeffding's inequality, with probability at least $1 - \delta$,

$$\text{Term I} = \sum_{t=1}^{T} \langle \ell_t, \tilde{a}_t \rangle - \langle \ell_t, \bar{a}_t \rangle \leq |S_T - \mathbb{E}[S_T]| \leq \sqrt{2 T D_\ell^2 D_a^2 \log(2/\delta)} \tag{6}$$

**Bounding Term II**
Term II captures the performance of the online gradient descent portion, line 4, of Algorithm 3. A difference is that the projection set is time-varying - yet this does not pose a problem for us since we only need to guarantee performance w.r.t. a $a^*$ in the intersection of all the sets. Let $a^* := \mathrm{argmin}_{a \in \cap_{t=1}^{T} \mathcal{A}_t} \langle \ell_t, a \rangle$. with this,

$$\text{Term II} = \sum_{t=1}^{T} \langle \ell_t, \bar{a}_t \rangle - \min_{a \in \bigcap_{t=1}^{T} \tilde{a}_t} \langle \ell_t, a \rangle = \sum_{t=1}^{T} \langle \ell_t, \bar{a}_t - a^* \rangle$$

Therefore, it is sufficient to bound the terms $\langle \ell_t, \bar{a}_t - a^* \rangle$. For any timestep $t$, we have:

$$\|\bar{a}_{t+1} - a^*\|_2^2 = \|\Pi_{\mathcal{A}_t}(\bar{a}_t - \eta\ell_t) - a^*\|_2^2$$
$$\leq \|\bar{a}_t - \eta\ell_t - a^*\|_2^2$$
$$= \|\bar{a}_t - a^*\|_2^2 + \eta^2\|\ell_t\|_2^2 - 2\eta\langle \ell_t, \bar{a}_t - a^* \rangle$$

where the inequality follows from the fact that $a^* \in \cap_{t=1}^T \mathcal{A}_t \subseteq \mathcal{A}_t$, so projection to $\mathcal{A}_t$ only decreases the distance. Rearranging,

$$\langle \ell_t, \bar{a}_t - a^* \rangle \leq \frac{1}{2\eta}\left(\|\bar{a}_t - a^*\|_2^2 - \|\bar{a}_{t+1} - a^*\|_2^2\right) + \frac{\eta}{2}\|\ell_t\|_2^2$$

Summing up the terms $t \in [T]$, we get:

$$\sum_{t=1}^T \langle \ell_t, \bar{a}_t - a^* \rangle \leq \frac{1}{2\eta}\sum_{t=1}^T \left(\|\bar{a}_t - a^*\|_2^2 - \|\bar{a}_{t+1} - a^*\|_2^2\right) + \frac{\eta}{2}\sum_{t=1}^T \|\ell_t\|_2^2$$
$$= \frac{1}{2\eta}(\|\bar{a}_1 - a^*\|_2^2 - \|\bar{a}_{T+1} - a^*\|_2^2) + \frac{\eta}{2}TD_\ell^2$$
$$\leq \frac{4D_a^2}{2\eta} + \frac{\eta}{2}TD_\ell^2$$

Setting $\eta = \frac{2D_a}{D_\ell\sqrt{T}}$, we get

$$\text{Term II} = \sum_{t=1}^T \langle \ell_t, \bar{a}_t - a^* \rangle \leq 2D_aD_\ell\sqrt{T} \tag{7}$$

Combining the bounds from equations 6 and 7, we get

$$\sum_{t=1}^T \langle \ell_t, \tilde{a}_t \rangle - \min_{a \in \cap_{t=1}^T a_t} \langle \ell_t, a \rangle = \text{Term I} + \text{Term II} \leq 4D_aD_\ell\sqrt{T\log(2/\delta)}$$

$\square$

### C.3. Proofs for Composing an Activation Function

*Lemma* (Lemma 5.10 restated). Let $\mathcal{G}$ be a function class with bounded $\kappa^*(\mathcal{G}, \mathbf{M}_\mathcal{G})$ for some mapping $\mathbf{M}_\mathcal{G}$. Suppose assumption 5.4 holds. Then $\kappa^*(\sigma(\mathcal{G}), \mathbf{M}_\mathcal{G}) \leq \frac{\kappa^*(\mathcal{G}, \mathbf{M}_\mathcal{G})}{\underline{c}}$.

*Proof.* For any set $\mathcal{F} \subseteq \sigma(\mathcal{G})$ let $\mathcal{G}_\mathcal{F}$ be the set for which $\mathcal{F} = \sigma(\mathcal{G}_\mathcal{F})$. Since $\sigma(0) = 0$ and $\sigma$ is non decreasing, for any $x \in \mathcal{X}$ we have $P(\mathcal{F}, x) = P(\mathcal{G}_\mathcal{F}, x)$ and $O(\mathcal{F}, x) = O(\mathcal{G}_\mathcal{F}, x)$. Given a context $x \in \mathcal{X}$, notice that $\mathbf{M}_\mathcal{G}$ is a mapping from distributions over optimistic sets to pessimistic sets. It follows that $\mathbf{M}_\mathcal{G}$ is a valid mapping satisfying $\mathbf{M}_\mathcal{G}(\cdot; \mathcal{G}_\mathcal{F}, x) = \mathbf{M}_\mathcal{G}(\cdot; \mathcal{F}, x)$. Let $\mathcal{F}^*$ be the maximizer of $\kappa^*(\sigma(\mathcal{G}), \mathbf{M}_\mathcal{G})$. Using the definition of $\kappa^*$,

$$\kappa^*(\sigma(\mathcal{G}), \mathbf{M}_\mathcal{G}) = \sup_{x \in \mathcal{X}, \tilde{p} \in \Delta(O(\mathcal{F}^*, x)), y \in \mathcal{Y}} \frac{\mathbb{E}_{a \sim \mathbf{M}_\mathcal{G}(\tilde{p}; \mathcal{F}^*, x)}[\ell(a, x, y)] - \mathbb{E}_{\tilde{a} \sim \tilde{p}}[\ell(\tilde{a}, x, y)]}{\mathbb{E}_{a \sim \mathbf{M}_\mathcal{G}(\tilde{p}; \mathcal{F}^*, x)}[\Delta_{\mathcal{F}^*}(a, x)]}$$

$$= \sup_{x \in \mathcal{X}, \tilde{a} \in O(\mathcal{F}^*, x), y \in \mathcal{Y}} \frac{\ell(\mathbf{M}_\mathcal{G}(\tilde{a}; \mathcal{F}^*, x), x, y) - \ell(\tilde{a}, x, y)}{\Delta_{\mathcal{F}^*}(\mathbf{M}_\mathcal{G}(\tilde{a}; \mathcal{F}^*, x))}$$

$$= \sup_{x \in \mathcal{X}, \tilde{a} \in O(\mathcal{F}^*, x), y \in \mathcal{Y}} \frac{\ell(\mathbf{M}_\mathcal{G}(\tilde{a}; \mathcal{G}_{\mathcal{F}^*}, x), x, y) - \ell(\tilde{a}, x, y)}{\Delta_{\mathcal{F}^*}(\mathbf{M}_\mathcal{G}(\tilde{a}; \mathcal{F}^*, x))}$$

$$= \sup_{x \in \mathcal{X}, \tilde{a} \in O(\mathcal{F}^*, x), y \in \mathcal{Y}} \frac{\Delta_{\mathcal{G}_{\mathcal{F}^*}}(\mathbf{M}_\mathcal{G}(\tilde{a}; \mathcal{G}_{\mathcal{F}^*}, x))}{\Delta_{\mathcal{F}^*}(\mathbf{M}_\mathcal{G}(\tilde{a}; \mathcal{F}^*, x))} \frac{\ell(\mathbf{M}_\mathcal{G}(\tilde{a}; \mathcal{G}_{\mathcal{F}^*}, x), x, y) - \ell(\tilde{a}, x, y)}{\Delta_{\mathcal{G}_{\mathcal{F}^*}}(\mathbf{M}_\mathcal{G}(\tilde{a}; \mathcal{G}_{\mathcal{F}^*}, x))}$$

$$\leq \kappa^*(\mathcal{G}, \mathbf{M}_\mathcal{G}) \frac{\Delta_{\mathcal{G}_{\mathcal{F}^*}}(\mathbf{M}_\mathcal{G}(\tilde{a}; \mathcal{G}_{\mathcal{F}^*}, x))}{\Delta_{\mathcal{F}^*}(\mathbf{M}_\mathcal{G}(\tilde{a}; \mathcal{F}^*, x))}$$

Now, for any $a \in \mathcal{A}, x \in \mathcal{X}$ and $\mathcal{F}$ let $\overline{g} := \operatorname{argmax}_{g \in \mathcal{G}_\mathcal{F}} g(a, x)$ and $\underline{g} := \operatorname{argmin}_{g \in \mathcal{G}_\mathcal{F}} g(a, x)$

$$\frac{\Delta_{\mathcal{G}_\mathcal{F}}(a)}{\Delta_\mathcal{F}(a)} = \frac{\overline{g}(a, x) - \underline{g}(a, x)}{\sigma(\overline{g}(a, x)) - \sigma(\underline{g}(a, x))}$$

$$\leq \frac{\overline{g}(a, x) - \underline{g}(a, x)}{\underline{c}(\overline{g}(a, x) - \underline{g}(a, x))}$$

$$\leq \frac{1}{\underline{c}}$$

where the second inequality follows from $\sigma(0) = 0$ and for any $x \in [-1, 1]$, $\sigma'(x)$ is bounded below by $\underline{c}$. Therefore $\kappa^*(\sigma(\mathcal{G}), \mathbf{M}_\mathcal{G}) \leq \frac{\kappa^*(\mathcal{G}, \mathbf{M}_\mathcal{G})}{\underline{c}}$ □

## C.4. Proofs for Vector-Valued Constraints

Recall that given a context $x \in \mathcal{X}$, an action $a \in \mathcal{A}$ is considered *safe* if $\|\mathring{f}(a, x)\|_\infty \leq 0$ for unknown vector-valued function $\mathring{f} \in \mathcal{F} \subseteq \mathcal{A} \times \mathcal{X} \to [-1, 1]^m$. Notably, Proposition 3.2 continues hold - guaranteeing the existence of an oracle $\mathsf{Oracle}_{\mathsf{OL}}$. We recall that for a vector-valued function class $\mathcal{F} \subseteq \mathcal{A} \times \mathcal{X} \to [-1, 1]^m$, we denote $\Delta_\mathcal{F}^\infty(a, x) := \sup_{f, f' \in \mathcal{F}} \|f(a, x) - f'(a, x)\|_\infty$, where $\|\cdot\|_\infty$ is the $\ell_\infty$ norm. We define the following vector valued analogues of optimistic and pessimistic sets:

$$O(\mathcal{G}, x) = \{a \in \mathcal{A} \mid \exists f \in \mathcal{G}, \|f(a, x)\|_\infty \leq 0\}$$
$$P(\mathcal{G}, x) = \{a \in \mathcal{A} \mid \forall f \in \mathcal{G}, \|f(a, x)\|_\infty \leq 0\}$$
$$\tag{8}$$

---

**Algorithm 4** General Online Learning with Vector-Valued Constraints

---

1: Input: $\mathsf{Oracle}_{\mathsf{OL}}$, $\mathsf{Oracle}_{\mathsf{OR}}$, Initial safe set $\mathcal{A}_0, \delta \in (0, 1)$
2: $\mathcal{F}_0^i = \{(a, x) \mapsto f(a, x)[i] : f \in \mathcal{F} : \forall a \in \mathcal{A}_0, \forall x \in \mathcal{X}, |f(a, x)[i]| \leq 0\}$
3: $\mathcal{F}_0 = \mathcal{F}_0^1 \times \dots \mathcal{F}_0^m$
4: **for** $t = 1, \dots, T$ **do**
5:     **Receive context** $x_t$
6:     $\mathcal{F}_t = \{f \in \mathcal{F}_0 : \sum_{s=1}^{t-1}(\|f(a_s, x_s) - \hat{z}_s\|_\infty)^2 \leq \mathsf{Reg}_{\mathsf{OR}}(T, \delta, \mathcal{F}_0)\}$
7:     $O_t = O(\mathcal{F}_t, x_t), P_t = P(\mathcal{F}_t, x_t)$ **// Optimistic/Pessimistic; cf. eq (8)**
8:     $\tilde{p}_t = \mathsf{Oracle}_{\mathsf{OL}t}(x_t, O_t)$
9:     $p_t = \mathbf{M}(\tilde{p}_t; \mathcal{F}_t, x_t)$
10:     Draw $a_t \sim p_t$
11:     **Receive noisy feedback** $z_t$
12:     Update $\hat{z}_t = \mathsf{Oracle}_{\mathsf{OR}t}(a_t, x_t, z_t)$
13:     **Play** $a_t$ **and receive** $y_t$
14: **end for**

---

The following assumption is a variant of Assumption 3.1 that applies to multiple constraints.

**Assumption C.2** (Online Regression Oracle with Vector Valued Constraints)**.** The algorithm $\mathsf{Oracle}_{\mathsf{OR}}$ guarantees that for any (possibly adversarially chosen) sequence $\{a_t, x_t\}_{t=1}^T$, for any $\delta \in (0, 1)$, with probability at least $1 - \delta$, generates predictions $\{\hat{z}_t\}_{t=1}^T$ with each $\hat{z}_t \in [-1, 1]^m$ satisfying:

$$\sum_{t=1}^T \|\hat{z}_t - \mathring{f}(a_t, x_t)\|_\infty^2 \leq \mathsf{Reg}_{\mathsf{OR}}(T, \delta, \mathcal{F})$$

The following formulation of link functions is standard in the literature (Sekhari et al., 2023b).

**Definition C.3.** A function $\Phi : [-1,1]^m \to \mathbb{R}$ is $\lambda$-strongly convex if for all $v, v' \in [-1,1]^m$, it satisfies

$$\frac{\lambda}{2}\|v' - v\|_2^2 \leq \Phi(v') - \Phi(v) + \langle \phi(v), v - v' \rangle$$

where $\phi(\cdot) = \nabla\Phi(\cdot)$.

**Definition C.4.** For a link function $\phi(\cdot) = \nabla\Phi(\cdot)$ for a $\lambda$-strongly convex function $\Phi$, we define the associated loss:

$$\ell_\phi(v, z) := \Phi(v) - v[z]$$

**Assumption C.5** (Online Regression Oracle with Vector Valued Constraints, Regret Version). The algorithm $\mathsf{Oracle_{OR}}$ guarantees that for any (possibly adversarially chosen) sequence $\{a_t, x_t\}_{t=1}^T$ generates predictions $\{\hat{z}_t\}_{t=1}^T$ satisfying:

$$\sum_{t=1}^T \ell_\phi(\hat{z}_t, z_t) - \inf_{\mathring{f} \in \mathcal{F}} \sum_{t=1}^T \ell_\phi(\mathring{f}(a_t, x_t), z_t) \leq \mathrm{Reg}_{\mathsf{OR}}^\phi(T, \mathcal{F})$$

where $z_t \sim \phi(\mathring{f}(a_t, x_t))$.

The following lemma is adapted from (Sekhari et al., 2023b), Lemma 5 and related to (Agarwal, 2013), Lemma 2

**Lemma C.6.** *Suppose that $z_t$ is generated with a link function $\phi$ that is the gradient of a $\lambda$-strongly convex function. Suppose that the regression oracle satisfies assumption C.5. Then for any $\delta \in (0,1)$ and $T \geq 3$, with probability at least $1 - \delta$, the regression oracle satisfies assumption C.2 with:*

$$\mathrm{Reg}_{\mathsf{OR}}(T, \delta, \mathcal{F}) \leq \frac{4}{\lambda}\mathrm{Reg}_{\mathsf{OR}}^\phi(T, \mathcal{F}) + \frac{112}{\lambda^2}\log\left(4\delta^{-1}\log(T)\right).$$

*Proof.* The proof is an application of (Sekhari et al., 2023b) Lemma 5. $\qquad\square$

**Lemma C.7** (Variant of Lemma B.1 for Vector Valued Constraints). *With probability at least $1 - \delta$, for all $t \in [T]$, $\mathring{f} \in \mathcal{F}_t$.*

*Proof.* First, notice that as $\mathring{f}$ is safe for all actions in $\mathcal{A}_0$, $\mathring{f} \in \mathcal{F}_0$. Assumption C.2 guarantees with probability at least $1 - \delta$.

$$\sum_{s=1}^T \|\mathring{f}(a_s, x_s) - \hat{z}_s\|_\infty^2 \leq \mathrm{Reg}_{\mathsf{OR}}(T, \delta, \mathcal{F}_0)$$

and hence for any $t \in [T]$,

$$\sum_{s=1}^{t-1} \|\mathring{f}(a_s, x_s) - \hat{z}_s\|_\infty^2 \leq \mathrm{Reg}_{\mathsf{OR}}(T, \delta, \mathcal{F}_0)$$

which shows $\mathring{f} \in \mathcal{F}_t$. $\qquad\square$

**Proposition C.8** (Variant of Proposition B.2 for Vector Valued Constraints). *Let $\{\mathcal{F}_t\}_{t=1}^T$ be the sequence of version spaces generated by Algorithm 4. For any $\delta \in (0,1)$ with probability at least $(1 - \delta)$ we have for any $t \in [T]$*

$$P(\mathcal{F}_t, x_t) \subseteq \{a \in \mathcal{A} : \|\mathring{f}(a, x_t)\|_\infty \leq 0\} \subseteq O(\mathcal{F}_t, x_t)$$

*Proof.* By Lemma C.7, we have with probability at least $1 - \delta$ that $\mathring{f} \in \mathcal{F}_t$ simultaneously for all $t \in [T]$. Take some arbitrary $t \in [T]$. First suppose $a$ is such that $\|\mathring{f}(a, x_t)\|_\infty \leq 0$. Since $\mathring{f} \in \mathcal{F}_t$, $a \in O(\mathcal{F}_t, x)$. Hence, $\{a \in \mathcal{A} : \|\mathring{f}(a, x_t)\| \leq 0\} \subseteq O(\mathcal{F}_t, x_t)$. Now suppose $a \in P(\mathcal{F}, x_t)$. Then, $\forall f \in \mathcal{F}_t$, $\|f(a, x_t)\|_\infty \leq 0$. Since $\mathring{f} \in \mathcal{F}_t$, $\|\mathring{f}(a, x_t)\|_\infty \leq 0$, and hence $P(\mathcal{F}_t, x_t) \subseteq \{a \in \mathcal{A} : \|\mathring{f}(a, x_t)\|_\infty \leq 0\}$ $\qquad\square$

**Lemma C.9** (Variant of Lemma B.3 for Vector Valued Constraints). *Let the sequence $\{\mathcal{F}_t, a_t, \hat{z}_t\}_{t=1}^T$ be generated by Algorithm 4. Then, for any sequence of adversarial contexts $\{x_t\}_{t=1}^T$, and $\epsilon > 0$, it holds that*

$$\sum_{t=1}^T \mathbb{1}\left\{\Delta_{\mathcal{F}_t}^\infty(a_t, x_t) > \epsilon\right\} \leq \sum_{t=1}^T \left(\frac{4\mathrm{Reg}_{\mathsf{OR}}(T, \delta, \mathcal{F}_0)}{\epsilon^2} + 1\right)\left(\sum_{i \in [m]} \mathcal{E}(\mathcal{F}_0^i, \epsilon)\right)$$

*Proof.* For all $i \in [m]$, let $\mathcal{F}_t^i := \{f^i \in \mathcal{F}_0^i \mid \sum_{s=1}^{t-1}(f^i(a_s, x_s) - \hat{z}_s^i)^2 \leq \mathrm{Reg}_{\mathrm{OR}}(T, \delta, \mathcal{F}_0)\}$ We appeal directly to Lemma B.3.

$$\sum_{t=1}^{T} \mathbb{1}\{\Delta_{\mathcal{F}_t}^{\infty}(a_t, x_t) > \epsilon\} = \sum_{t=1}^{T} \mathbb{1}\left\{\max_{i \in m} \Delta_{\mathcal{F}_t^i}(a_t, x_t) > \epsilon\right\}$$

$$\leq \sum_{i \in [m]} \sum_{t=1}^{T} \mathbb{1}\left\{\Delta_{\mathcal{F}_t^i}(a_t, x_t) > \epsilon\right\}$$

$$\leq \sum_{i \in [m]} \sum_{t=1}^{T} \left(\frac{4\mathrm{Reg}_{\mathrm{OR}}(T, \delta, \mathcal{F}_0)}{\epsilon^2} + 1\right) \mathcal{E}(\mathcal{F}_0^i, \epsilon)$$

$$\leq \sum_{t=1}^{T} \left(\frac{4\mathrm{Reg}_{\mathrm{OR}}(T, \delta, \mathcal{F}_0)}{\epsilon^2} + 1\right) \left(\sum_{i \in [m]} \mathcal{E}(\mathcal{F}_0^i, \epsilon)\right)$$

where the third line follows from Lemma B.3. $\qquad \square$

The following lemma is similar to Lemma B.4 - we state it for completeness.

**Lemma C.10** (Variant of Lemma B.4 for Vector Valued Constraints). *Let the sequence $\{\mathcal{F}_t, p_t\}_{t=1}^{T}$ be generated by Algorithm 4. Then, for any sequence of adversarial contexts $\{x_t\}_{t=1}^{T}$,*

$$\sum_{t=1}^{T} \mathbb{E}_{a_t \sim p_t} \left[\Delta_{\mathcal{F}_t}^{\infty}(a_t, x_t)\right] \leq \inf_{\alpha} \left\{\alpha T + \frac{20\mathrm{Reg}_{\mathrm{OR}}(T, \delta, \mathcal{F}_0) \sum_{i=1}^{m} \mathcal{E}(\mathcal{F}_0^i, \alpha)}{\alpha}\right\}$$

*Proof.* For a run of Algorithm 1, let $\{a_t\}_{t=1}^{T}$ be any sequence of actions drawn $a_t \sim p_t$ for all $t \in [T]$. Furthermore, to simplify the notation, let us denote $\Delta_t := \Delta_{\mathcal{F}_t}(a_t, x_t)$. Let us consider some arbitrary $\alpha > 0$. Then, for this sequence of actions and contexts,

$$\sum_{t=1}^{T} \Delta_{\mathcal{F}_t}(a_t, x_t) := \sum_{t=1}^{T} \Delta_t$$

$$\overset{(i)}{=} \sum_{t:\Delta_t \leq \alpha} \Delta_t + \sum_{i=0}^{\log(2/\alpha)-1} \left(\sum_{t:2^i \alpha < \Delta_t \leq 2^{i+1}\alpha} \Delta_t\right)$$

$$\leq \alpha T + \sum_{i=0}^{\log(2/\alpha)-1} \left(\sum_{t:2^i \alpha < \Delta_t \leq 2^{i+1}\alpha} 2^{i+1}\alpha\right)$$

$$\overset{(ii)}{\leq} \alpha T + \sum_{i=0}^{\log(2/\alpha)-1} \left(2^{i+1}\alpha \left(\frac{4\mathrm{Reg}_{\mathrm{OR}}(T, \delta, \mathcal{F}_0)}{2^{2i}\alpha^2} + 1\right) \left(\sum_{j \in [m]} \mathcal{E}(\mathcal{F}_0^j, 2^i \alpha)\right)\right)$$

$$\overset{(iii)}{\leq} \alpha T + \sum_{i=0}^{\log(2/\alpha)-1} \left(2^{i+1}\alpha \left(\frac{5\mathrm{Reg}_{\mathrm{OR}}(T, \delta, \mathcal{F}_0)}{2^{2i}\alpha^2}\right) \left(\sum_{j \in [m]} \mathcal{E}(\mathcal{F}_0^j, 2^i \alpha)\right)\right)$$

$$\leq \alpha T + \sum_{i=0}^{\log(2/\alpha)-1} \left(\frac{10\mathrm{Reg}_{\mathrm{OR}}(T, \delta, \mathcal{F}_0)}{2^i \alpha}\right) \left(\sum_{j \in [m]} \mathcal{E}(\mathcal{F}_0^j, 2^i \alpha)\right)$$

$$\overset{(iv)}{\leq} \alpha T + \left(\sum_{j \in [m]} \mathcal{E}(\mathcal{F}_0^j, \alpha)\right) \sum_{i=0}^{\infty} \frac{10\mathrm{Reg}_{\mathrm{OR}}(T, \delta, \mathcal{F}_0)}{2^i \alpha}$$

$$\overset{(v)}{\leq} \alpha T + \frac{20\mathrm{Reg}_{\mathrm{OR}}(T, \delta, \mathcal{F}_0) \sum_{j \in [m]} \mathcal{E}(\mathcal{F}_0^j, \alpha)}{\alpha}$$

In (i) we set the upper bound to the sum as $\log(2/\alpha) - 1$ since all functions $f \in \mathcal{F}$ map to $[-1, 1]$, hence $\Delta_t \leq 2$ so it is enough to consider $i : 2^{i+1}\alpha \leq 2$ and (ii) follows from Lemma B.3, (iii) follows from the fact that $1 \leq \frac{\text{Reg}_{\text{OR}}(T, \delta, \mathcal{F})}{(2^i\alpha)^2}$ for $i \in [\log(2/\alpha) - 1]$ if $T > 1$, (iv) follows from the fact that $\mathcal{E}(\mathcal{F}, \cdot)$ is nonincreasing in its second argument, and (v) is an upper bound from the sum of an infinite series. Therefore, for any sequence $\{\mathcal{F}_t, a_t, x_t\}_{t=1}^T$ generated by Algorithm 1 we have

$$\sum_{t=1}^T \Delta_{\mathcal{F}_t}(a_t, x_t)\alpha T + \frac{20\text{Reg}_{\text{OR}}(T, \delta, \mathcal{F}_0) \sum_{j\in[m]} \mathcal{E}(\mathcal{F}_0^j, \alpha)}{\alpha}$$

Now, since this holds for any sequence $\{\mathcal{F}_t, a_t\}_{t=1}^T$ generated by the algorithm and adversarial contexts $\{x_t\}_{t=1}^T$, it holds in expectation over the algorithm's draws. $\square$

The following is an analogue of Theorem 4.2 for multiple constraints.

**Corollary C.11.** *For any $\delta \in (0, 1)$ with probability at least $1 - 3\delta$, Algorithm 4 produces a sequence of actions $\{a_t\}_{t=1}^T$ that are safe, and enjoys the following bound on regret:*

$$Regret_T \leq \inf_{\kappa > 0} \left\{ \sum_{t=1}^T \mathcal{V}_\kappa^{\mathbf{M}}(\tilde{p}_t; \mathcal{F}_t, x_t) + \kappa \inf_\alpha \left\{ \alpha T + \frac{20\text{Reg}_{\text{OR}}(T, \delta, \mathcal{F}_0) \sum_{i\in[m]} \mathcal{E}(\mathcal{F}_0^i, \alpha)}{\alpha} \right\} \right\}$$
$$+ \text{Reg}_{\text{OL}}(T, \delta, \Pi) + \sqrt{2T\log(\delta^{-1})}$$

*where,*

$$\mathcal{V}_\kappa^{\mathbf{M}}(\tilde{p}_t; \mathcal{F}_t, x_t) = \sup_{y\in\mathcal{Y}} \left\{ \mathop{\mathbb{E}}_{a_t\sim\mathbf{M}(\tilde{p}_t;\mathcal{F}_t, x_t)} [\ell(a_t, x_t, y)] - \mathop{\mathbb{E}}_{\tilde{a}_t\sim\tilde{p}_t} [\ell(\tilde{a}_t, x_t, y)] \right\} - \kappa \mathop{\mathbb{E}}_{a_t\sim\mathbf{M}(\tilde{p}_t;\mathcal{F}_t, x_t)} \left[ \Delta_{\mathcal{F}_t}^\infty(a_t, x_t) \right]$$

*Further, if we use $\kappa^* = \kappa^*(\mathcal{F}, \mathbf{M}) := \sup_{\hat{\mathcal{F}}\subseteq\mathcal{F}, x\in\mathcal{X}, \tilde{p}\in\Delta(O(\hat{\mathcal{F}},x)), y\in\mathcal{Y}} \frac{\mathbb{E}_{a\sim\mathbf{M}(\tilde{p};\hat{\mathcal{F}},x)}[\ell(a,x,y)] - \mathbb{E}_{\tilde{a}\sim\tilde{p}}[\ell(\tilde{a},x,y)]}{\mathbb{E}_{a\sim\mathbf{M}(\tilde{p};\hat{\mathcal{F}},x)}[\Delta_{\hat{\mathcal{F}}}^\infty(a,x)]}$, then in the above, $\mathcal{V}_\kappa^{\mathbf{M}}(\tilde{p}_t; \mathcal{F}_t, x_t) \leq 0$ and so we can conclude that:*

$$Regret_T \leq \kappa^* \inf_\alpha \left\{ \alpha T + \frac{20\text{Reg}_{\text{OR}}(T, \delta, \mathcal{F}_0) \sum_{i\in[m]} \mathcal{E}(\mathcal{F}_0^i, \alpha)}{\alpha} \right\} + \text{Reg}_{\text{OL}}(T, \delta, \Pi) + \sqrt{T\log(\delta^{-1})}$$

*Proof.* By Proposition C.8, with probability at least $1 - \delta$, if we play actions from $P_t$, we can guarantee the all the constraints are satisfied. On the other hand, to bound the regret of our algorithm w.r.t. the optimal action in hindsight that also satisfies constraint on every round, note that

$$\text{Regret}_T := \sum_{t=1}^T \ell(a_t, x_t, y_t) - \min_{\substack{\pi\in\Pi:\forall t \\ \|\hat{f}(\pi(x_t), x_t)\|_\infty \leq 0}} \sum_{t=1}^T \ell(\pi(x_t), x_t, y_t)$$

$$\overset{(i)}{\leq} \sum_{t=1}^T \ell(a_t, x_t, y_t) - \min_{\substack{\pi\in\Pi:\forall t \\ \pi(x_t)\in O_t}} \sum_{t=1}^T \ell(a, x_t, y_t)$$

$$\leq \sum_{t=1}^T \left( \ell(a_t, x_t, y_t) - \mathop{\mathbb{E}}_{\tilde{a}_t\sim p_t} [\ell(\tilde{a}_t, x_t, y_t)] \right) + \sum_{t=1}^T \mathop{\mathbb{E}}_{\tilde{a}_t\sim p_t} [\ell(\tilde{a}_t, x_t, y_t)] - \min_{\substack{\pi\in\Pi:\forall t \\ \pi(x_t)\in O_t}} \ell(a, x_t, y_t)$$

$$\leq \sum_{t=1}^T \left( \ell(a_t, x_t, y_t) - \mathop{\mathbb{E}}_{\tilde{a}_t\sim p_t} [\ell(\tilde{a}_t, x_t, y_t)] \right) + \text{Reg}_{\text{OL}}(T, \delta, \Pi)$$

$$\overset{(ii)}{\leq} \sum_{t=1}^T \left( \mathop{\mathbb{E}}_{a_t\sim p_t} [\ell(a_t, x_t, y_t)] - \mathop{\mathbb{E}}_{\tilde{a}_t\sim\tilde{p}_t} [\ell(\tilde{a}_t, x_t, y_t)] \right) + \text{Reg}_{\text{OL}}(T, \delta, \Pi) + \sqrt{T\log(\delta^{-1})}$$

$$\leq \inf_{\kappa > 0} \left\{ \sum_{t=1}^{T} \mathcal{V}_\kappa^{\mathbf{M}}(\tilde{p}_t; \mathcal{F}_t, x_t) + \kappa \sum_{t=1}^{T} \mathop{\mathbb{E}}_{a_t \sim p_t} \left[ \Delta_{\mathcal{F}_t}^{\infty}(a_t, x_t) \right] \right\} + \mathrm{Reg}_{\mathrm{OL}}(T, \delta, \Pi) + \sqrt{T \log(\delta^{-1})}$$

where (i) follows from the fact that by Proposition C.8, a policy $\pi$ satisfying $\forall t, \|\mathring{f}(\pi(x_t), x_t)\|_\infty \leq 0$ satisfies $\forall t, \pi(x_t) \in O_t$, (ii) is an application of Hoeffding Azuma to bound $\sum_{t=1}^{T} \ell(a_t, x_t, y_t) - \sum_{t=1}^{T} \mathbb{E}_{a_t \sim p_t} [\ell(a_t, x_t, y_t)]$ and:

$$\mathcal{V}_\kappa^{\mathbf{M}}(\tilde{p}_t; \mathcal{F}_t, x_t) = \sup_{y \in \mathcal{Y}} \left\{ \mathop{\mathbb{E}}_{a_t \sim \mathbf{M}(\tilde{p}_t; \mathcal{F}_t, x_t)} [\ell(a_t, x_t, y)] - \mathop{\mathbb{E}}_{\tilde{a}_t \sim \tilde{p}_t} [\ell(\tilde{a}_t, x_t, y)] \right\} - \kappa \mathop{\mathbb{E}}_{a_t \sim \mathbf{M}(\tilde{p}_t; \mathcal{F}_t, x_t)} \left[ \Delta_{\mathcal{F}_t}^{\infty}(a_t, x_t) \right]$$

by Lemma C.10 we can bound the $\sum_{t=1}^{T} \kappa \mathbb{E}_{a_t \sim p_t} [\Delta_{\mathcal{F}_t}(a_t, x_t)]$ term, hence,

$$\mathrm{Regret}_T \leq \inf_{\kappa > 0} \left\{ \sum_{t=1}^{T} \mathcal{V}_\kappa^{\mathbf{M}}(\tilde{p}_t; \mathcal{F}_t, x_t) + \kappa \inf_{\alpha} \left\{ \alpha T + \frac{20 \mathrm{Reg}_{\mathrm{OR}}(T, \delta, \mathcal{F}_0) \sum_{i=1}^{m} \mathcal{E}(\mathcal{F}_0^i, \alpha)}{\alpha} \right\} \right\}$$
$$+ \mathrm{Reg}_{\mathrm{OL}}(T, \delta, \Pi) + \sqrt{T \log(\delta^{-1})}$$

This concludes the first bound - which holds with probability at least $1 - 3\delta$ as we take a union bound over the online regression oracle guarantee, the online learning oracle guarantee, and the application of Hoeffding Azuma. To conclude the second part of the statement, we need to show that for

$$\kappa^* = \kappa^*(\mathcal{F}, \mathbf{M}) := \sup_{\hat{\mathcal{F}} \subseteq \mathcal{F}, x \in \mathcal{X}, \tilde{p} \in \Delta(O(\hat{\mathcal{F}}, x)), y \in \mathcal{Y}} \frac{\mathbb{E}_{a \sim \mathbf{M}(\tilde{p}; \hat{\mathcal{F}}, x)} [\ell(a, x, y)] - \mathbb{E}_{\tilde{a} \sim \tilde{p}} [\ell(\tilde{a}, x, y)]}{\mathbb{E}_{a \sim \mathbf{M}(\tilde{p}; \hat{\mathcal{F}}, x)} \left[ \Delta_{\hat{\mathcal{F}}}^{\infty}(a, x) \right]}$$

we have that $\mathcal{V}_{\kappa^*}^{\mathbf{M}}(\tilde{p}_t; \mathcal{F}_t, x_t) \leq 0$. To this end, note that

$$\mathcal{V}_{\kappa^*}^{\mathbf{M}}(\tilde{p}_t; \mathcal{F}_t, x_t)$$
$$= \sup_{y \in \mathcal{Y}} \left\{ \mathop{\mathbb{E}}_{a_t \sim \mathbf{M}(\tilde{p}_t; \mathcal{F}_t, x_t)} [\ell(a_t, x_t, y)] - \mathop{\mathbb{E}}_{\tilde{a}_t \sim \tilde{p}_t} [\ell(\tilde{a}_t, x_t, y)] \right\} - \kappa^* \mathop{\mathbb{E}}_{a_t \sim \mathbf{M}(\tilde{p}_t; \mathcal{F}_t, x_t)} \left[ \Delta_{\mathcal{F}_t}^{\infty}(a_t, x_t) \right]$$
$$= \sup_{y \in \mathcal{Y}} \left\{ \mathop{\mathbb{E}}_{a_t \sim \mathbf{M}(\tilde{p}_t; \mathcal{F}_t, x_t)} [\ell(a_t, x_t, y)] - \mathop{\mathbb{E}}_{\tilde{a}_t \sim \tilde{p}_t} [\ell(\tilde{a}_t, x_t, y)] \right\}$$
$$- \left( \sup_{\hat{\mathcal{F}} \subseteq \mathcal{F}, x \in \mathcal{X}, \tilde{p} \in \Delta(O(\hat{\mathcal{F}}, x)), y \in \mathcal{Y}} \frac{\mathbb{E}_{a \sim \mathbf{M}(\tilde{p}; \hat{\mathcal{F}}, x)} [\ell(a, x, y)] - \mathbb{E}_{\tilde{a} \sim \tilde{p}} [\ell(\tilde{a}, x, y)]}{\mathbb{E}_{a \sim \mathbf{M}(\tilde{p}; \hat{\mathcal{F}}, x)} \left[ \Delta_{\hat{\mathcal{F}}}^{\infty}(a, x) \right]} \right) \mathop{\mathbb{E}}_{a_t \sim \mathbf{M}(\tilde{p}_t; \mathcal{F}_t, x_t)} \left[ \Delta_{\mathcal{F}_t}^{\infty}(a_t, x_t) \right]$$
$$\leq \sup_{y \in \mathcal{Y}} \left\{ \mathop{\mathbb{E}}_{a_t \sim \mathbf{M}(\tilde{p}_t; \mathcal{F}_t, x_t)} [\ell(a_t, x_t, y)] - \mathop{\mathbb{E}}_{\tilde{a}_t \sim \tilde{p}_t} [\ell(\tilde{a}_t, x_t, y)] \right\}$$
$$- \left( \frac{\sup_{y \in \mathcal{Y}} \{\mathbb{E}_{a_t \sim \mathbf{M}(\tilde{p}_t; \mathcal{F}_t, x_t)} [\ell(a_t, x_t, y)] - \mathbb{E}_{\tilde{a}_t \sim \tilde{p}_t} [\ell(\tilde{a}_t, x_t, y)]\}}{\mathbb{E}_{a_t \sim \mathbf{M}(\tilde{p}_t; \mathcal{F}_t, x_t)} \left[ \Delta_{\mathcal{F}_t}^{\infty}(a_t, x_t) \right]} \right) \mathop{\mathbb{E}}_{a_t \sim \mathbf{M}(\tilde{p}_t; P_t, \mathcal{F}_t, x_t)} \left[ \Delta_{\mathcal{F}_t}^{\infty}(a_t, x_t) \right]$$
$$= 0$$

$\square$

### C.4.1. PROOFS FOR POLYTOPIC CONSTRAINTS WITH SCALAR FEEDBACK

Recall that $\mathbf{M}^{\mathrm{Lin}}(\tilde{p}; \mathcal{F})$ is defined as the distribution induced by drawing $\tilde{a} \sim \tilde{p}$ and outputting $\gamma(\tilde{a}; \mathcal{F})\tilde{a}$, where $\gamma(\tilde{a}; \mathcal{F}) := \max \{\gamma \in [0, 1] : \gamma \tilde{a} \in P(\mathcal{F})\}$. First we show a lemma that lower bounds $\gamma(\tilde{a}; \mathcal{F})$.

**Lemma C.12.** *Let $\mathcal{F}$ be an arbitrary subset of $\mathcal{F}^{\mathrm{Poly}}$ and consider some $\tilde{a} \in O(\mathcal{F})$. $\gamma(\tilde{a}; \mathcal{F}) := \max \{\gamma \in [0, 1] : \gamma \tilde{a} \in P(\mathcal{F})\}$ is lower bounded as:*

$$\gamma(\tilde{a}; \mathcal{F}) \geq \frac{b}{b + \Delta_{\mathcal{F}}(\tilde{a})}$$

*Proof.* From the definition of $\tilde{a} \in O(\mathcal{F})$, we know $\|\underline{f}(\tilde{a})\|_\infty \leq b$ for some $\underline{f} \in \mathcal{F}$. Let $\overline{f}$ be some arbitrary function in $\mathcal{F}$. Let $\alpha = \frac{b}{b + \Delta_{\mathcal{F}}(\tilde{a})}$. Then:

$$\begin{aligned}
\|\overline{f}(\alpha\tilde{a})\|_\infty &= \|\underline{f}(\alpha\tilde{a}) + \overline{f}(\alpha\tilde{a}) - \underline{f}(\alpha\tilde{a})\|_\infty \\
&\leq \|\underline{f}(\alpha\tilde{a})\|_\infty + \|\overline{f}(\alpha\tilde{a}) - \underline{f}(\alpha\tilde{a})\|_\infty \\
&\leq \alpha(b + \Delta_{\mathcal{F}}(\tilde{a})) \\
&\leq b
\end{aligned}$$

This shows that $\alpha\tilde{a} \in P(\mathcal{F})$ as $\overline{f}$ was arbitrary. Since we defined $\gamma(\tilde{a}; \mathcal{F}) = \max\{\gamma \in [0,1] : \gamma\tilde{a} \in P(\mathcal{F})\}$, it follows that

$$\gamma(\tilde{a}; \mathcal{F}) \geq \alpha = \frac{b}{b + \Delta_{\mathcal{F}}(\tilde{a})}$$

$\square$

*Lemma.* (Lemma 5.11 Restated) Suppose Assumption 5.4 holds. Then, $\kappa^*(\mathcal{F}^{\text{Poly}}, \mathbf{M}^{\text{Lin}}) \leq \frac{D_\ell D_a}{b}$.

*Proof.* Given Lemma C.12, the proof is analogous to that of Lemma 5.6. $\square$

