# OpenReview forum: "Online Learning with Unknown Constraints"
_ICML.cc/2025/Conference — ICML 2025 poster_

### Official Review · Reviewer_sXWZ · 2025-03-03

**Overall Recommendation:** 2

**Summary:**

The paper provides new insights into the problem of online learning with unknown constraints. Lower and upper bounds that connect the difficulty of the problem with Eluder dimension are derived.

**Claims And Evidence:**

Yes

**Essential References Not Discussed:**

Not to my knowledge.

**Experimental Designs Or Analyses:**

No

**Methods And Evaluation Criteria:**

See Questions for Authors

**Other Comments Or Suggestions:**

See Questions for Authors.

**Other Strengths And Weaknesses:**

See Questions for Authors.

**Questions For Authors:**

Mainly I have doubts on whether the proposed methods are only theoretical constructs with very weak practical implications.
The concerns are noted below:

1)	The practical motivation of the problem setup and the how the proposed modelling of the safety sets and the associated feedback mechanisms are connected to real-life applications is highly unclear.
2)	Why is the constraint function assumed to be fixed? Environment changes over time can very well induce changes in functional forms of the constraint set.
3)	The feedback mechanism $P_{sig}$ is assumed to be known to select the online regression oracle. How practical is it to obtain such prior knowledge or to even verify such assumptions?
4)	The assumed regret minimization online oracle is only shown to be computationally feasible in a simplified setting of linear optimization. In general the computational feasibility of this assumption is unclear. This also adds into my concerns regarding the usefulness of the proposed methodologies in real life applications.
5)	In Assumption 3.2, the functional form of the loss $\ell$ remains fixed. Do we have to know in advance, the form of loss function in general (example, it is squared error, or loss from the GLM family like linear or logistic regression type losses). Or can the loss be selected adversarialy by nature?
6)	How computationally feasible is to do the sampling in Line 8 of the Algorithm 1?

7) Though the paper provides theoretical insights, the provided upper and lower bounds do not match.

**Relation To Broader Scientific Literature:**

It is well addressed in Related works, though I have concerns. See Questions for Authors.

**Theoretical Claims:**

No

---

> ### Author Rebuttal · Authors · 2025-04-01
>
> We thank the reviewer for their questions. If the reviewer wishes, we are more than happy to engage in follow-up discussions through OpenReview!
>
> We would like to highlight the fact that our paper is theoretical in nature, and our main focus is to establish information theoretically when safe learning is possible, and to develop algorithms for safe online learning with general safety function classes $\mathcal{F}$ given access to an online learning oracle and an online regression oracle.
>
> > The practical motivation of the problem setup and the how the proposed modelling of the safety sets and the associated feedback mechanisms are connected to real-life applications is highly unclear.
>
> We believe there is practical motivation in enforcing a strict per-round safety constraint. The feedback mechanism is general enough to encompass a standard $f^*(a_t,x_t) + $ (subgaussian gaussian noise) model commonly assumed in the literature as well as binary feedback following a Boltzmann distribution, which is a well-studied form of feedback in sociology known as the Bradley-Luce-Shepard rule.
>
> > Why is the constraint function assumed to be fixed? Environment changes over time can very well induce changes in functional forms of the constraint set.
>
> While other works (e.g. Neely et al 2017 "Online convex optimization with time varying constraints") consider time-varying constraints in the setting of "long term constraints" (a significantly weaker safety ask), in the setting of strict per-timestep constraints, one would need a fixed safety constraint for there to be any hope of learning such a function in a every-time safe way. Let us further recall that we allow for a context and that the safety function is also a function of this context. Thus one could encode changes in safety set over time by encoding time as a part of context.
>
> > The feedback mechanism $P_{sig}$ is assumed to be known to select the online regression oracle. How practical is it to obtain such prior knowledge or to even verify such assumptions?
>
> If we have prior knowledge about the problem at hand we can select for the appropriate feedback mechanism. For example, in settings with instrument readings with noise, we could adopt a standard  $f^*(a_t,x_t) + $ (subgaussian gaussian noise) model. With human-generated feedback models, we could use the Bradley-Luce-Shepherd rule and assume that the feedback is binary following a Boltzmann distribution.
>
>  > The assumed regret minimization online oracle is only shown to be computationally feasible in a simplified setting of linear optimization. In general the computational feasibility of this assumption is unclear. This also adds into my concerns regarding the usefulness of the proposed methodologies in real life applications.
>
> This is true, we only show a concrete implementation for safe linear optimization. In this paper, our focus is more theoretical in nature, instead opting to show the existence of such online learning oracles, and proving that the performance of our algorithm depends on variations of complexity measures found in the online learning literature.
>
> > In Assumption 3.2, the functional form of the loss $\ell$ remains fixed. Do we have to know in advance, the form of loss function in general (example, it is squared error, or loss from the GLM family like linear or logistic regression type losses). Or can the loss be selected adversarialy by nature?
>
> Yes, we assume we know the form the loss function takes - we believe this is a standard assumption in the online learning literature.
>
> > How computationally feasible is to do the sampling in Line 8 of the Algorithm 1?
>
> This wholly depends on choice of mapping $\mathbf M$ - for the linear optimization example, this is simply sampling from a set of $d+1$ vectors. Discovering other settings where all steps/oracles are computationally efficient is an active work in progress.
>
> > Though the paper provides theoretical insights, the provided upper and lower bounds do not match.
>
> This is true - getting exactly matching upper and lower bounds is a work in progress. The main take away of the lower bound was to show that at least asymptotically, minimizing the proposed complexity measure is necessary for safe learning.

---

### Official Review · Reviewer_zBP3 · 2025-03-15

**Overall Recommendation:** 3

**Summary:**

The paper addresses the problem of online learning with unknown safety constraints, where constraint satisfactions is required per round. The authors provide a general meta-algorithm based on the Online Learning oracle for online learning strategy with known constraints, and Online Regression oracle for constraints estimation. The algorithm is based on constructing Optimistic and Pessimistic safe action sets, and a mapping strategy from the optimistic to the pessimistic set. The paper provides an upper regret bound which is also depending on the Eluder dimension describing complexity of the constraint function class and on a new complexity measure $\mathcal V_{\kappa}(\cdot)$.  The paper also provides a lower bound demonstrating that if $\lim_{T \rightarrow \infty} \sum_{t = 1}^T \mathcal V_{\kappa}(\tilde p_t, \mathcal F_t) > 0$, then no safe algorithm can ensure a sublinear regret. The paper also provides concrete algorithm and bounds in some problem cases such as with linear constraints.

## Update after rebuttal:

I thank the authors for the response. I have also read the other reviews, and I decided to keep my initial score.

**Claims And Evidence:**

Yes, it seems so, mostly. However, there are some inconsistent notations and unclarities regarding the new complexity measure. In particular, the novel complexity measure proposed is recalled as $\mathcal V_t(\kappa)$ in the introduction, and $\mathcal V_{\kappa}(\cdot)$ further in the paper. It would be good to keep them consistent. Also, it is unclear to me why the complexity measure depends on the time step? In particular, it depends on $(\tilde p_t, \mathcal F_t, x_t)$ (Maybe a typo, but in Theorem 4.4 the dependence on the $x_t$ is omitted $(\tilde p_t, \mathcal F_t) $ ) Here $x_t$ is a context received at step $t$. How can the complexity measure dependent on the contexts? Should it maybe be rather defined in a way which is independent, like a worst case? And as an asymptotic sum? How can this complexity measure be estimated in advance in practice?

Also, at the beginning of Section 4 it is written that a bounded $\mathcal V_{\kappa}(\cdot)$ is asymptotically necessary. Although, does from theorem 4.4 it follows that one needs to have not just bounded complexity measures $\mathcal V_{\kappa}(\cdot)$, but their sum to be growing less than linearly on T.

Also, the intuition of notion $\kappa$ is not very clear. It would be great if it explained more.

**Essential References Not Discussed:**

Not that I am aware of.

**Experimental Designs Or Analyses:**

There are no experimental designs

**Methods And Evaluation Criteria:**

The proposed method makes sense, the problem seems to be quite general.

**Other Comments Or Suggestions:**

There are a few typos I noticed, apart from $\mathcal V$:

- line 260, right side: "with respect to with respect to"
- line 275, right side: comma

**Other Strengths And Weaknesses:**

The paper is quite clearly written in general and tries to add the intuition for any notion and claim they state. However, the number of the notations is quite high, and sometimes hard to follow. Also, the complexity notion $\mathcal V_t$ is questionable since it depends on time-step $t$ and on the particular realizations of the context $x_t$ and the estimated version space $\mathcal F_t$. Is there a way to have a more general notion of complexity?  Maybe the asymptotic sum directly, under some conditions on the feedback $\{x_t\}$?

**Questions For Authors:**

See Claims and Evidence

**Relation To Broader Scientific Literature:**

The paper provides new insights about the complexity of a per-step online learning, also in comparison to the complexity of cumulative constraints satisfaction.

**Theoretical Claims:**

I did not check the proofs in the appendix, and not all the derivations in the paper body. In general they seem correct

---

> ### Author Rebuttal · Authors · 2025-04-01
>
> We thank the reviewer for their review and helpful suggestions, which we will be sure to incorporate into our final version. If the reviewer wishes, we are more than happy to engage in follow-up discussions through OpenReview!
>
> > In particular, the novel complexity measure proposed is recalled as $\mathcal V_t(\kappa)$ in the introduction, and $\mathcal V_k(\cdot)$ further in the paper.
>
> Thank you for the feedback - we had made this notational choice in the "key contributions" section as we wanted to emphasize the dependence on $\kappa$, and avoid cluttering the definition with technical definitions of $\mathcal F_t, \tilde p_t$. We will update the measure as $\mathcal V_\kappa(t)$ in the "key contributions" to keep it more consistent.
>
> > Also, it is unclear to me why the complexity measure depends on the time step? In particular, it depends on $(\tilde p_t, \mathcal F_t, x_t)$
>
> This is true, the complexity measure really is a function of just those three parameters. The complexity measure captures the difficulty of matching a target $\tilde p_t$ using distributions from $P_t$ (defined through $\mathcal F_t$ and $x_t$) under worst case loss while off-setting information gain w.r.t $\mathcal F_t$, weight by $\kappa$. However, when applied to our Main Algorithm (Algorithm 1), the instantiations of those three parameters depend on time-step.
>
> > Maybe a typo, but in Theorem 4.4 the dependence on $x_t$ is omitted
>
> This is not a typo - as stated in the theorem statement, our Theorem 4.4 considers a simplified non-contextual setting (i.e.  $\mathcal X = \\{ \\}$). In non-contextual settings, we drop the context argument (mentioned in line 274 right column footnote).
>
> > How can the complexity measure dependent on the contexts? Should it maybe be rather defined in a way which is independent, like a worst case?
>
> This indeed is a valid alternative way of defining the complexity measure - we could instead have taken worst case contexts. However, as the pessimistic and optimistic sets are dependent on contexts, we thought this would make the notational a little less convoluted.
>
> > How can this complexity measure be estimated in advance in practice?
>
> In practice, we believe the best way to address this complexity measure is to use a choice of $\kappa^*$ that ensures that $\mathcal V_{\kappa^*} < 0$ (various concrete examples of this methodology described in section 5). If one wanted to directly estimate the value of the complexity measure, one could deploy the methodology mentioned in the response to reviewer pLb5.
>
> > ... it follows that one needs to have not just bounded complexity measures $\mathcal V_t(\kappa)$, but their sum to be growing less than linearly on T
>
> Yes, this suggests we'd need the sum of $\mathcal V_t(\kappa)$ to be asymptotically growing sublinearly in T (otherwise safe learning is impossible). This is precisely our main takeaway in justifying the necessity of $\mathcal V_t(\kappa)$ as a measure of the complexity of safe learning!
>
> > ... the intuition on the notion of $\kappa$ is not clear.
>
> Thank you for the feedback. As mentioned in line 57 left column, $\mathcal V_t(\kappa)$ defined in eq 5 ... captures the per-timestep trade off between loss minimization and information gain w.r.t. unknown constraint. $\kappa$ is the parameter that controls the weight placed on information gain. While this is evident upon inspection of eq 5, we agree that it would be illuminating to mention this explicitly. We will update this paragraph to make this more clear!
>
> >  Is there a way to have a more general notion of complexity? Maybe the asymptotic sum directly, under some conditions on the feedback $x_t$?
>
> Yes, we could instead use the asymptotic sum directly as the measure of complexity instead. Getting the upper and lower bounds to match is an ongoing work, and we would like to arrive at a unified measure of complexity.
>
> > Typos in lines 260 and 275
>
> Thank you for pointing out these typos to us. We will fix them in the final version.

---

### Official Review · Reviewer_EPZo · 2025-03-17

**Overall Recommendation:** 3

**Summary:**

This paper studies the online adversarial learning problem of achieving no-regret guarantees while playing actions subject to an unknown constraint at each time-step. At each round $t$, the set of safe actions is a function of the adversarially chosen context and an unknown safety constraint $f*$; after playing an action the learner receives a loss vector and safety feedback correlated to the unknown constraint value. The goal is to achieve cumulative loss with sublinear regret with respect to the best action in hindsight that satisfies all constraints, while also playing actions that satisfy the constraints with high probability each round. The authors present a generalized algorithm that achieves this, making use of a black-box no-regret algorithm with constrained actions sets (used to make predictions that achieve low regret while simultaneously choosing only safe actions), and a black-box online regression model (used to predict the value of the unknown constraint function $f*$ at each round). They show that a simpler version of this algorithm solves the slightly-modified problem of achieving no-regret guarantees while satisfying long-term constraints (as opposed to a constraint at each round). Finally, since the algorithm is non-constructive (assuming the existence of a mapping function satisfying a particular property), they discuss several settings of action sets, safety constraints, and loss functions for which this mapping can be concretely defined, and thus the algorithm itself.

The general idea of the algorithm is to use the regression model’s predictions of the value of $f*$ at each round to determine a subset $\mathcal{F}_t$ of the constraint functions that includes the true safety function with high probability. Then the constrained no-regret algorithm can be used to predict a distribution over actions by constraining to the set of actions that could be safe (inferred from $\mathcal{F}_t$), which is mapped to a distribution $p$ over the set of actions which are definitely safe using a mapping function $M$. The authors define a complexity measure (defined in part by $M$) that at a high-level, trades off between how much worse your loss regret may be because you need to play a safe action, and how much new information you expect to learn about $f*$ using the current distribution $p$. By optimizing for $M$, one can keep the sum of this measure over time small, and the overall regret can be bound by this sum as well as the regrets of the two black-box algorithms used as subroutines.

**Claims And Evidence:**

The claims are all proved.

**Essential References Not Discussed:**

Not to my knowledge

**Experimental Designs Or Analyses:**

There are no experiments.

**Methods And Evaluation Criteria:**

Yes, the defined setting seems reasonable. The assumption of receiving (possibly noisy) feedback as a function of the safety function’s value generalizes to many safety-critical settings.

**Other Comments Or Suggestions:**

N/A

**Other Strengths And Weaknesses:**

Strengths:

Safety-critical settings often coincide with non-distributional / adversarial settings, so this is an interesting area of study and the algorithm seems novel and applicable. The complexity measure introduced which is not only sufficient (in a sense) for bounding to get low regret, but also necessary to have low for safe learning to be possible indicates that it does measure something critical to the safe learning setting. The extension of this work to the long-term constraints setting is interesting because the approach seems quite different from what I have generally seen in the area (usually involving no-regret algorithms using the Lagrangian or some modification as a loss function). I also think the paper is very well-presented, with the structure and intuition behind the ideas / proofs making it fairly easy to follow throughout, even for people not familiar with the area.

Weaknesses:

In Section 4.4, the authors say that Assumption 5.1 is necessary for safe learning in the given setting, but this seems misleading – they show this by providing a counter-example, but this doesn’t prove the necessary condition – couldn’t there be settings where Assumption 5.1 doesn’t hold and still safe learning is possible (unless the adversary is choosing the safety function, which I don’t think they are doing).

**Questions For Authors:**

1.	On line 134, is $p_{signal}$ a distribution parametrized by $f^*(a,x)$? Should the mapping be from [-1,1] to a distribution over $\mathcal{Z}$?
2.	In Section 3.3, performance is compared to some set of policies $\Pi$ which I assume need not be deterministic. Does this mean $\Pi_T$ is the set of policies $\pi$ which always only induce distributions over actions whose support is within the constrained action set, or is it comparing to specific runs of the policy that satisfy the constraints?

**Relation To Broader Scientific Literature:**

I am not very familiar with the learning with constraints literature – based on the papers cited by the authors it seems that the main difference between these results and prior ones are that other algorithms either focus on achieving strong safety guarantees without prioritizing achieving low regret, or bound the number of “unsafe” actions that will be taken over a long time, rather than consistently playing safe actions with high probability.

**Theoretical Claims:**

I looked at the proof for the main Theorem 4.2 (which looked good), though not all of the lemmas preceding it.

---

> ### Author Rebuttal · Authors · 2025-04-01
>
> We would like to thank the reviewer for their review and questions. If the reviewer wishes, we are more than happy to engage in follow-up discussions through OpenReview!
>
> > In Section 4.4, the authors say that Assumption 5.1 is necessary for safe learning in the given setting, but this seems misleading – they show this by providing a counter-example, but this doesn’t prove the necessary condition – couldn’t there be settings where Assumption 5.1 doesn’t hold and still safe learning is possible (unless the adversary is choosing the safety function, which I don’t think they are doing).
>
> We would like to point out that $f^*$ is unknown and fixed - we could indeed interpret it as being chosen by the adversary before the game starts (and hence the adversary could choose $f^*$ in the offending $\mathcal F^*$ mentioned in the counter-example). Alternatively, we could view Assumption 5.1 as being $f^*$-dependent. We could instead require that $\forall \mathcal F \subseteq \mathcal F^{FAS}$ s.t. $f^*\in \mathcal F$ ... the gap-condition in the body of Assumption 5.1 holds.
>
> > On line 134 ... is $p_{signal}$ a distribution parameterized by $f^*(a,x)$? Should the mapping be from [-1,1] to a distribution over $\mathcal Z$
>
> Yes, $p_{signal}$ is a distribution parameterized by $f^*(a,x)$. And yes! Thank you for pointing out this typo, we will update this in our final version.
>
> > In Section 3.3, performance is compared to some set of policies $\Pi$ which I assume need not be deterministic. Does this mean $\Pi_T$ is the set of policies $\pi$ which always only induce distributions over actions whose support is within the constrained action set, or is it comparing to specific runs of the policy that satisfy the constraints?
>
> $\Pi_T$ would be the set of policies which only induce distributions over intersection of constrained action sets $\cap \mathcal A_t$. This aligns with the fact that generally we compare against policies that are always safe (induce distributions over safe unknown safe set).

---

### Official Review · Reviewer_cCk6 · 2025-03-18

**Overall Recommendation:** 3

**Summary:**

This paper is on contextual online learning with unknown constraints that are stochastic and roundwise. Let $\ell$ be a given loss function, of an action $a \in \mathcal{A},$ a context $x \in \mathcal{X}$ and an "outcome" $y \in \mathcal{Y}$; and let $f_* \in \mathcal{F}$ an unknown constraint function parametrised by $a$ and $x$. It is demanded that, without knowing $f_*$ a priori, the learner must ensure that the selected action $a_t$ satisfies $f_*(a_t, x_t) \le 0$, serving as a hidden constraint. This is enabled both by the a priori knowledge of, for each possible context $x$, a safe set $\mathcal{A}_0(x) \subset \mathcal{A},$ which allows for initial selection of actions, and noisy feedback of a safety signal $z_t$ (which one can just think of as $f\_*(a\_t, x\_t) + \mathrm{noise},$ although richer structures can be accommodated). This essentially constitutes a stochastic feedback of the unknown constraint, and actions meeting these constraints are termed safe.

Of course, while meeting this constraint, the learner attempts to optimise a cost. This is captured through a regret with respect to a policy $\pi$ of the form $\sum \mathbb{E}\_{a_t \sim p_t}[ \ell(a_t, x_t,y_t)] - \ell(\pi(x_t), x_t,y_t)$. Here, this outcome signal $y_t$ is not revealed to the learner until after the action $a_t$ is selected, and $p_t$ is a law over actions that the learner uses to select $a_t$ (and this must be supported on $a : f_*(a, x_t) \le 0$). Thus, this boils down to a mildly structured version of the standard adversarial online learning setup (which is used for reasons that will be seen below).

Throughout, the set of competing policies is restricted to the elements of a known class $\Pi$ that are always safe, i.e., $\Pi_T := \{ \pi \in \Pi : \forall t, \pi(x\_t) \in \mathcal{A}\_t \},$ where $\mathcal{A}\_t = \mathcal{A}\_t(x\_t) = \\{a : f\_*(a, x_t)\le 0\\}$.

At a high level, then, this setup considers online learning with unknown unconstraints, where the constraint information follows a stochastic bandit framework, while the cost minimisation follows the adversarial online learning framework. As such, the setting mixes aspects of safe stochastic bandits, specifically the roundwise enforcement of constraint using noisy bandit observations of the same, and online learning with unknown static constraints.

The paper aims at general characterizations of when sublinear regret and roundwise-safety can be attained in this setting, along with general methods that may attain the same. Towards this, the authors begin with an algorithmic framework that generalises a "optimistic to pessimistic projection" idea of Hutchinson et al. that was proposed for the setting of safe linear bandits. At a high level, let $\mathcal{F}\_t \subset \mathcal{F}$ denote the version space of functions that agree with the constraint feedback, constructed via an online regression oracle (and naturally, under and Eluder dimension constraint on $\mathcal{F}$). This version space induces two natural sets of actions at time $t$
 - the optimistically safe actions $O\_t = \\{ a : \exists f \in \mathcal{F}\_t : f(a, x_t) \le 0 \\},$ and
-  the pessimistically safe actions $P\_t = \\{a : \forall f \in \mathcal{F}\_t : f(a,x_t) \le 0\\}$.

Note that $P\_t$ is all of the actions that certifiably are safe (with high probability et c.) given all of the safety feedback (and the known safe actions a priori). As such, to ensure safety, the learner _must_ be restricted to selecting actions only from $P\_t$. Thus, a crucial challenge of this setup is the expansion of $P\_t$ towards the true set of feasible action (this also shows up in safe bandits). On the flipside, the competing policies cannot select an action outside of $O\_t$, and so if we could pick actions from this set using a low-regret method, we would ensure good cost regret. The algorithmic approach taken is to map from a good law over actions in $O\_t$, to a law over $P\_t$.

Concretely, let $M$ be a map from laws over $O\_t$ to laws over $P\_t$ (that can further utilise the $\mathcal{F}\_t$ and the context $x_t$). The algorithmic framework proposed in the paper is thus:
 - using an online learning oracle, compute a distribution $\tilde{p}\_t$ over $O\_t$. It is assumed that this is low-regret against policies in $\Pi_T$
- Draw and play $a_t \sim p\_t = M(\tilde{p}\_t , \mathcal{F}\_t, x\_t)$.

$\newcommand{\vv}{\mathcal{V}} $ To control the regret of such play, the authors define a complexity measure associated with this map $M$, and parameterised by a number $\kappa \ge 0$,  $$  \vv\_{\kappa}^M(\tilde{p}\_t, \mathcal{F}\_t, x\_t) := \sup\_{y} \\{ \mathbb{E}\_{a \sim M(\tilde{p}\_t)}[ \ell(a, x_t, y)] - \mathbb{E}\_{a \sim\tilde{p}\_t}[ \ell(a, x_t, y)] \\} - \kappa \mathbb{E}_{a \sim M(\tilde{p}\_t)}[\Delta\_{\mathcal{F}\_t}(a, x_t)].$$

The first term above is a worst-case bound on how much extra cost is incurred due to playing according to $p_t = M(\tilde{p}\_t),$ while the second term is the expected width, wherein $\Delta\_{\mathcal{F}\_t}(a, x) = \sup_{f,f' \in \mathcal{F}\_t} |f(a,x) - f'(a,x)|$. Note that, just by using the fact that we can bound $\mathbb{E}\_{M(\tilde{p}\_t}[ \ell(a, x_t, y_t)] - \mathbb{E}\_{\tilde{p}\_t}[ \ell(a, x_t, y_t)]$ in terms of $\vee_\kappa^M + \kappa \mathrm{width}$, this gives a bound on the regret of the form $$ \mathrm{Reg}(\textrm{Online Learing}) + \kappa ( \mathrm{Reg}(\textrm{Width})) + \sum_t \mathcal{V}^M\_\kappa(\tilde{p}\_t, \mathcal{F}\_t, x_t),$$ where this second term is bounded in terms of a regression cost, the eluder dimension, and an accumulated approximation level.

The authors offer the interpretation of $\vv\_{\kappa}^M$ as a tradeoff between the cost of gaining information about the constraint function $f\_{\*}$, and between minimising regret (which in turn requires fidelity to the unsafe distribution $\tilde{p}\_t$). Naturally, for any $\kappa,$ we can define an optimal $M_*$ by minimising the objective, which yields a (trajectory-dependent!!) "measure of complexity", $\vv\_\kappa(\tilde{p}\_t, \mathcal{F}\_t, x_t).$ This is presented as an analogue of the decision-estimation coefficient for this problem.

One can also, for a fixed $M$, optimise the $\kappa$ to yield the strongest bounds, and various results of this type are stated. Naturally, there are two questions of interest: how pertinent is the measure $\vv\_{\kappa}$ to such regret problems, and if there are situations with maps $M$ that yield sublinear regret in the above setup.

To address the first, the authors show a lower bound, which essentially says that in a context-free situation with a fixed known loss that is optimised at a safe point, and with a good regression oracle and bounded Eluder dimension, there was a $\kappa$ such that $\sum \vv\_{\kappa}(\tilde{p}\_t, \mathcal{F}\_t) = \Omega(T),$ then one can find a non-expandable subset of safe actions $P\_{\*}$ that are a) all suboptmal by a constant amount, and b) non-expandable (i.e., information about $f\_{\*}$ for actions in $P\_{\*}$ does not leak information about $f\_{\*}$ for actions outside of $P\_{\*}$). This implies a strong result, that if the initial safe action set if $P\_{\*},$ then sublinear regret is impossible.

For the second point, the authors study the behaviour of linear constraints (recovering extensions of prior work of Hutchinson et al.), and show a compositionality result that, e.g., extends this to generalised linear models.

**Claims And Evidence:**

The paper is positioned as providing a general complexity measure for this problem, $\vv\_{\kappa}$ and a general methodology driven by this object. The main claims beyond this are that this methodology can yield nontrivial regret bounds in certain scenarios (which is demonstrated for linear losses and constraint), and that without sublinearity of $\sum \vv\_{\kappa}(\tilde{p}\_t, \mathcal{F}\_t),$ there may be situations where no method can obtain sublinear regret.

For me, all of the subsequent claims are fine, and the main sticking point, naturally, is whether I buy this $\vv\_{\kappa}$ as a general complexity measure. The obvious issue here is that this is strongly trajectory-dependent. Now, if the lower bound had said (and this would be very strong indeed) that no matter the situation, if $\vv\_{\kappa}$ is not summable, then no method can learn, then I would agree that nothing more can be said, but that's not quite what the lower bound shows: the setup is very particular. The authors also appeal to the DEC formulation, but note that this paper is careful to state things in terms of a static model class. A final issue, of course, is the efficiency of this method: it is not clear to me that in general this viewpoint recovers tight bounds. I do appreciate that this gets the right rates in the linear setting, but is this enough? Certainly, Assumption  In other words, I don't quite buy that $\vv\_{\kappa}$ "characterizes the difficulty of safe learning" (line 56, col 1): the evidence is too thin for such a strong claim, both from the viewpoints of upper and lower bounds.

**Essential References Not Discussed:**

I don't know if this is essential, but work on doubly-optimistic methods is omitted from the discussion of safe bandits (Chen et al., ICML 2022, "Startegies for Safe...", Gangrade et al., COLT 2024, "Safe Linear Bandits over Unknown Polytopes").

**Experimental Designs Or Analyses:**

N.A.

**Methods And Evaluation Criteria:**

This is a theoretical paper. The main methodological idea in the proposed framework is to project distributions over an optimistic estimate of the feasible set to that over a pessimistic estimate of the same, which (significantly) generalises an existing approach of Hutchinson et al. in safe bandit problems (and so makes sense).

**Other Comments Or Suggestions:**

- There are some jarring variations in the line spacing across the document (compare, e.g., line 127 onwards in columns 1 and 2). Usually people do this for space, but there's like enough whitespace floating about the document for this to not be necessary. Please fix this, it just distracts from reading.

- IMO the discussion following assumption 3.2 should simply describe, in words, that this property holds true, and move Proposition 3.3 to the appendix --- this is not per se adding much to help explain the main ideas of this paper, and this space can be more valuably spent doing that, or expanding upon the very terse treatment of section 5. I think in part I am saying this because for most online-learning algorithms I can think of, one begins with a round-wise control on the loss of the actions, and with this view it is not surprising to me that as long as the action space available to the learner contains the actions of all competing policies, the regret is controlled. There may well be issues with this (haven't thought too much), but IMO a reader would be happy to buy the assumption on the basis of this (and a general proof of achievability in the appendix).

- Please be explicit about the independence structure of the feedback (rather than telling me about its generality): in particular, my reading is that it is intended that $z_t$ is conditionally independent of the history given $x_t$ and $a_t$ (which should be stated).

**Other Strengths And Weaknesses:**

One important weakness is that the paper is that the main ideas are presented in a pretty rushed way. Take page 5, column 2, the main motivation and presentation of the underlying idea. The introduction of $M$ would benefit from some explanation as to why this projection idea could possibly get good regret, which would help introduce the idea of $\vv\_{\kappa}^M$. This would also benefit from an explanation of what the terms therein are, and why they are arising/natural. In part this is also true of section 5.

I also found the order of the presentation to be confusing, and did not really understand the idea of the paper until I went and stared at Algorithm 1, understood the methodological idea, and then went back to the general complexity measure being proposed. It somehow seems like in order to assert the generality of the result, some level of comprehensibility is sacrificed. I think this hurt the paper more than it helps it.

Note that this may well be an issue of space: the DEC paper, which the authors compare their contributions to, takes a 130+pages (with 60+ pages in the main text) to explain its ideas and convince them that they really have something concrete. If this is the reason, then the authors should, IMO, think about if a conference paper with limited page counts is really the right venue to send their work to, and if it may not instead benefit with a more comprehensive treatment sent to a journal.


In any case, right now I think I find the claims of $\vv\_{\kappa}$ being a general complexity measure that captures safe online learning to be somewhat exaggerated, and find the presentation a bit lacking, but I do think that the results are strong and interesting. The former mostly is why my recommendation below is a weak accept rather than an accept.

**Questions For Authors:**

-

**Relation To Broader Scientific Literature:**

I think the contributions are novel, and extend certain existing ideas strongly and in a insightful way. The results themselves are interesting. The observation that (although in a restricted setting) if $\vv\_{\kappa}(\tilde{p}\_t, \mathcal{F}\_t)$ is not sublinearly integrable, then _no method_ can learn nontrivially is, to me, surprisingly and insightful. The general upper bound is also interesting, especially since it can be realised well for linear models, but leaves the question of how generic it really is open. Overall, despite the gaps, I am certain that the results will be of interest to the online learning community at ICML.

**Theoretical Claims:**

I at least skimmed through most of the proofs, and they appear correct to me. I read appendices A and B more closely, and these are correct as far as I can tell.

---

> ### Author Rebuttal · Authors · 2025-04-01
>
> Thank you for the careful read and very helpful suggestions, which we will be sure to incorporate into our final version. If the reviewer wishes, we are more than happy to engage in follow-up discussions through OpenReview!
>
> > The obvious issue here is that this is strongly trajectory-dependent. Now, if the lower bound had said (and this would be very strong indeed) that no matter the situation, if $\mathcal V_\kappa$ is not summable, then no method can learn, then I would agree that nothing more can be said, but that's not quite what the lower bound shows: the setup is very particular.
>
> We believe this point to be important in the reviewer's assessment of the strength of our lower bound. We would like to ask the reviewer **what they meant by "no matter the situation"**. We would like to clarify that $\mathcal V_\kappa(\cdot) := \inf_{M} \mathcal V_\kappa^M(\cdot)$ (line 248 right column), and hence the lower bound states that if even the best mapping is not summable, no method can learn. On the other hand, if this optimal mapping makes $\mathcal V_\kappa$ summable then this mapping can be used in upper bounds. Correspondingly, our lower bound is in a setting where $\mathcal V_\kappa$ is not summable no matter what mapping is used. Furthermore, while this complexity indeed is trajectory-dependent, the lower bound correspondingly also sums across the trajectory of the best mapping. When we use a particular mapping, the trajectory is what is generated by that mapping. As to the setup being very particular, this setup was engineered to show the necessity of the $\mathcal V_\kappa(\cdot)$ terms. The other terms in the upper bound (Eluder and oracle bounds) are well justified in the literature and each one can already individually shown to be necessary. So we justify the necessity of the novel complexity term we introduced alone.
>
> >  A final issue, of course, is the efficiency of this method: it is not clear to me that in general this viewpoint recovers tight bounds. I do appreciate that this gets the right rates in the linear setting, but is this enough? Certainly, Assumption In other words, I don't quite buy that $\mathcal V_\kappa$ "characterizes the difficulty of safe learning" (line 56, col 1): the evidence is too thin for such a strong claim, both from the viewpoints of upper and lower bounds.
>
> To the best of our knowledge at the time of writing, existing works have focused solely on the linear setting. We believe our methodology is a first step towards the analysis of more general settings.
>
> > I don't know if this is essential, but work on doubly-optimistic methods is omitted from the discussion of safe bandits (Chen et al., ICML 2022, "Startegies for Safe...", Gangrade et al., COLT 2024, "Safe Linear Bandits over Unknown Polytopes").
>
> While these works do indeed have a "doubly optimistic" flavor to them, these works do not study the same setting of per-round strict safety constraints. Among "doubly optimistic" works, we believe that the referenced work of Hutchison et al., AISTATS 2024 "Directional Optimism for Safe Linear Bandits" is the most relevant as it considers a per-round strict safety constraint setting.
>
> > One important weakness is that the paper is that the main ideas are presented in a pretty rushed way... . It somehow seems like in order to assert the generality of the result, some level of comprehensibility is sacrificed.
>
> Thank you for the feedback, and we will work towards expanding the presentation in sections 4 and 5. As the reviewer later suggests, shortening section 3.3 to make space for this expansion seems like a wonderful idea. As noticed by the reviewer, a focus of our paper was to express the generality of the approach (as existing works focus solely on the linear setting) - and we will work to make this presentation more digestible.
>
> > There are jarring variations in line spacing ...
>
> We will fix this in our final version!
>
> > The discussion following assumption 3.2 should simply describe, in words, that this property holds true, and move Proposition 3.3 to the appendix ...
>
> Thank you for the feedback, we will shorten this section. The reason we provided the proposition was because while the underlying proof technique is standard, we couldn't just cite existing results and claim Sequential Rademacher Complexity upper bounds the online learning regret as we needed to satisfy the given constraints - which requires some nontrivial manipulation.
>
> > Please be explicit about the independence structure of the feedback ...
>
> We will make a note early on explicitly mentioning independence structure. We had hoped $z_t \sim p_{\text{signal}}(f^*(a_t,x_t))$ would highlight this.

---

### Official Review · Reviewer_pLb5 · 2025-03-20

**Overall Recommendation:** 3

**Summary:**

This paper studies the problem of bandit with constraints. Specifically, the forecaster wants to minimize the regret, while keeping the safe constraints satisfied with high probability. To resolve this question, the authors propose a new complexity measure, and they provide upper bounds and lower bounds to the regret
1. They proposed an algorithm with regret bounded in terms of this complexity measure.
2. If the sum of this complexity measure from step one to $T$ scales linear with $T$, then no algorithm can achieve sublinear regret.

**Claims And Evidence:**

Yes, the claims in this submission are supported by clear and convincing evidence.

**Essential References Not Discussed:**

Not that I know of.

**Experimental Designs Or Analyses:**

No experiments in this paper.

**Methods And Evaluation Criteria:**

Yes.

**Other Comments Or Suggestions:**

I don't have further comments and suggestions.

**Other Strengths And Weaknesses:**

Strengths:
1. This paper is well written. The proofs are correct.
2. The developing of this new concept $V_\kappa$ is insightful, as it provides both upper and lower bounds to the regret.

Weaknesses:
1. The upper and lower bounds do not match.

**Questions For Authors:**

I have the following questions regarding this paper:
1. The lower bound only states that when the sum of $V(F_t)$ is at least $\Omega(T)$, then no algorithm can achieve sublinear regret. Can you prove a lower bound stating that the regret is lower bounded by the sum of $V(F_t)$?
2. In the lower bound you designed, we have to solve the optimization problem to find the p to minimize $M(p)$. Is there any efficient algorithm that can achieve this?

**Relation To Broader Scientific Literature:**

Previous literature either studies safe bandit with known constraints, or linear bandit with unknown constraints, or convex optimization with unknown constraints. The setting considered in this setting of bandit with function approximation and unknown constraints, is new and not appeared in previous literature.

**Theoretical Claims:**

Yes. I checked the proofs of main theorems including Theorem 4,2, Theorem 4.3 and Theorem 4.4.

---

> ### Author Rebuttal · Authors · 2025-04-01
>
> We would like to thank the reviewer for their review and questions. If the reviewer wishes, we are more than happy to engage in follow-up discussions through OpenReview!
>
> > "The upper and lower bounds do not match" and "Can you prove a lower bound stating that the regret is lower bounded by the sum of $\mathcal V(\mathcal F_t)$?"
>
> This is true - getting exactly matching upper and lower bounds is a work in progress. The main take away of the lower bound was to show that at least asymptotically, minimizing the proposed complexity measure is necessary for safe learning.
>
> > "In the lower bound you designed, we have to solve the optimization problem to find p to minimize $M(p)$. Is there any efficient algorithm that can achieve this?
>
> We are a little confused why the reviewer is interested in the optimization problem to find p minimizing $M(p)$ in the lower bound. As for solving this optimization in the upper bound,  (line 243 right column), we notice that this is a saddle-point optimization. This can be solved up to desired accuracy (through standard techniques) by treating it as a two-player game where the min-player who chooses $p$, and the max-player chooses $f,f',y$. This can be solved by forming an $\epsilon$-net over the set of actions and while the optimization time complexity would be exponential in the dimension, it is still optimizable and gives a concrete mapping for a fixed $\kappa$. For finite, linear and generalized linear examples we show computationally efficient (but sub-optimal from point of view of optimization in line 243) mappings such as scaling etc suffice and provide upper bounds through our techniques.

---

### Decision · Program_Chairs · 2025-05-01

**Decision:**

Accept (poster)

**Comment:**

This paper studies the problem of online adversarial learning with unknown constraints, where the goal is to minimize regret while ensuring safety at each round. At every time step, the learner is presented with a context chosen adversarially, and must select an action from a set that depends on this context and an unknown safety constraint. After taking an action, the learner receives a loss vector and feedback correlated with the constraint. The authors show that their algorithm enjoys a regret bound in terms of a complexity measure. They also show that
if the sum of this complexity measure scales linear with the number of rounds $T$, then no algorithm can achieve a sublinear regret.

The proposed algorithm is non-constructive, relying on the existence of a mapping with a certain desired property. However, the authors do demonstrate that in several practical settings—defined by specific types of action sets, constraints, and loss functions—this mapping can be made explicit and hence the algorithm is implementable in those cases.

The majority of the reviewers (4 out of 5) recommend a weak accept, while one recommends a reject (whose main concern is its practical implications, which could be subjective). A potential weakness of this work might be whether the proposed algorithm can be _efficiently_ implemented in practice and if the setting is highly relevant to the real world. But a couple of reviewers believe that this can still be of interest to the online learning community at ICML. Furthermore, the majority of the reviewers agree that this paper is well written and the technical analysis are solid.  Hence, this paper is moved to a weak accept if room.